# GEPO: Group Expectation Policy Optimization for Stable Heterogeneous Reinforcement Learning

Han Zhang[1],[*] Ruibin Zheng[2],[1]*, Zexuan Yi[1], Zhuo Zhang[3],[1], Hanyang Peng[1], Hui Wang[1],
Jiayin Qi[2],[1], Binxing Fang[2], Ruifeng Xu[3],[1]†, Yue Yu[1]†
[1] Pengcheng Laboratory, [2] Guangzhou University,
[3] Harbin Institute of Technology (Shenzhen)

## Abstract

As single-center computing approaches power constraints, decentralized training becomes essential. However, traditional Reinforcement Learning (RL) methods, crucial for enhancing large model post-training, cannot adapt to decentralized distributed training due to the tight coupling between parameter learning and rollout sampling. For this, we propose **HeteroRL**, a heterogeneous RL architecture that decouples these processes, enabling stable training across geographically distributed nodes connected via the Internet. The core component is Group Expectation Policy Optimization (**GEPO**), an asynchronous RL algorithm robust to latency caused by network delays or heterogeneity in computational resources. Our study reveals that high latency significantly increases KL divergence, leading to higher variance of importance weights and training instability. GEPO mitigates this issue by using group expectation weighting to exponentially reduce the variance of importance weights, with theoretical guarantees. Experiments show GEPO achieves superior stability—only a 3% performance drop from online to 1800s latency—and reduces the best-to-last gap by 85% versus GSPO ($\Delta$=1.8 vs. 12.0) while attaining the highest scores, highlighting its effectiveness in decentralized, resource-heterogeneous environments. Code is available at `https://github.com/HanlardResearch/Hetero-RL`.

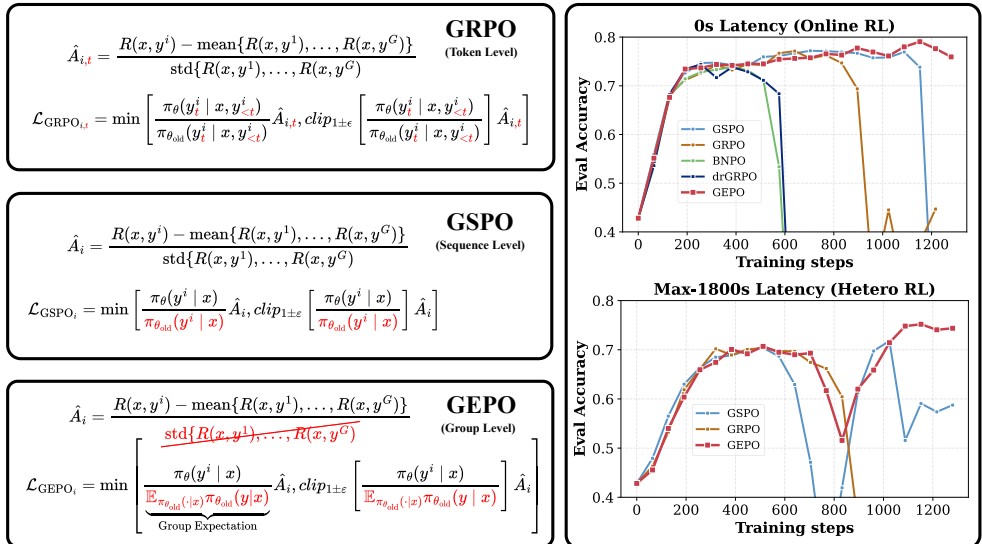

Figure 1: Left: GEPO improves upon GRPO and GSPO by employing group-level importance weights to enhance training stability. Right: In both zero-delay (online) and high-delay (up to 1800 seconds) heterogeneous reinforcement learning scenarios, GEPO demonstrates superior stability and better evaluation performance.

*    Equal contribution. Email: `HanlardResearch@gmail.com` and `rbzheng@e.gzhu.edu.cn`
†    Ruifeng Xu and Yue Yu are Corresponding authors.

# 1 INTRODUCTION

Training ever-larger AI models (Achiam et al., 2023; Dubey et al., 2024; Yang et al., 2025) is pushing the limits of single datacenters, making decentralized training across geographically distributed, heterogeneous nodes connected via the Internet an increasingly necessary pursuit (Team et al., 2025; Noukhovitch et al., 2024). Reinforcement Learning (RL), crucial for post-training LLMs on complex tasks like mathematical reasoning (Shao et al., 2024), faces a fundamental systemic challenge in this emerging paradigm: traditional RL frameworks (Guo et al., 2025; Stiennon et al., 2020; Bai et al., 2022; Wu et al., 2025; Dai et al., 2024; Fu et al., 2025) are architecturally incompatible with decentralized environments. Their tight coupling between rollout sampling and parameter learning demands strict synchronization — a requirement that becomes untenable under the high network latency and computational heterogeneity inherent in real-world distributed settings.

This architectural incompatibility manifests in two critical bottlenecks. First, synchronous frameworks force computational resources (e.g., GPUs) to idle while waiting for the slowest processes—such as generating long reasoning chains—severely constraining efficiency (Fu et al., 2025). Second, and more fundamentally, the inevitable network latency inherent in decentralized, Internet-connected environments creates a temporal gap (policy staleness) between the sampler (generating data) and the learner (updating parameters). Most existing RL algorithms, designed for homogeneous, low-latency clusters, are ill-equipped to handle this staleness. As our analysis reveals, high latency significantly inflates KL divergence, causing the variance of importance weights to explode—ultimately leading to training instability or reward collapse (Song et al., 2023). This renders conventional RL methods impractical for real-world, geographically distributed training scenarios.

To tackle these systemic bottlenecks, we introduce **HeteroRL** (Heterogeneous Reinforcement Learning), a novel RL framework explicitly architected for asynchronous, geographically distributed, and resource-heterogeneous environments. HeteroRL is designed to enable efficient and stable training of large language models for complex tasks such as mathematical reasoning, even under high network latency. At its core, HeteroRL decouples the two computationally intensive phases of the RL pipeline — rollout sampling and parameter learning — and deploys them on physically or logically independent nodes with potentially heterogeneous hardware (e.g., mixing NVIDIA and Ascend chips). The sampler nodes continuously generate reasoning trajectories without interruption, while the learner node asynchronously consumes this data to update model parameters. Critically, neither component waits for the other: communication occurs infrequently and tolerates high latency, with model checkpoints and rollout batches exchanged over the Internet.

To address the instability arising from KL divergence under high-latency conditions, we introduce **Group Expectation Policy Optimization (GEPO)**, a novel policy gradient algorithm that stabilizes asynchronous RL under high latency by replacing fragile token/sample-level importance weights with robust group-level importance weights — allowing samplers and learners to operate independently, communicating infrequently and tolerating arbitrary delays. This shift fundamentally improves the quality of gradient estimation — transforming a high-variance, unstable estimator into a low-variance, robust one, especially under large policy divergence. As we prove in Theorem 1, GEPO exponentially reduces the variance of importance weights under high KL divergence — precisely the regime where traditional methods like GRPO and GSPO collapse. Crucially, GEPO is not an ad hoc fix — it is a principled algorithmic response to the root cause of instability: variance explosion under policy divergence.

In summary, our key contributions are as follows:

**Framework:** We propose HeteroRL, an asynchronous reinforcement learning framework designed for heterogeneous compute networks, enabling decentralized training of large language models (LLMs) on mathematical reasoning tasks.

**Insight:** We identify a strong correlation between latency and the KL divergence between the rollout sampler and the learner. High latency induces high KL divergence, leading to training instability and reward collapse.

**Algorithm:** We introduce Group Expectation Policy Optimization (GEPO), which improves upon the importance sampling mechanism in GRPO (Shao et al., 2024). We theoretically show that GEPO exponentially reduces the variance of importance weights, and empirically demonstrate its superior stability and efficiency — not only under high-latency conditions, but also in the ideal zero-latency setting.

This work provides both algorithmic and system-level advancements for scalable LLM RL-training and establishes a practical foundation for large-scale distributed AI training in future heterogeneous compute network environments.

## 2 BACKGROUND

### 2.1 PROBLEM DEFINITION AND NOTATION

Consider a standard policy gradient framework. Let $\pi_\theta$ denote the language model policy (i.e., the Actor) parameterized by $\theta$, $x$ be an input prompt of a Dataset $\mathcal{D}$ (e.g., math problems), and $y$ be the output sequence generated by the model (e.g., a chain-of-thought solution). We define the following core notation:

- $\pi_{\theta_k}$ (short for $q$): the policy used by the *sampler* at time step $k$ to generate rollout trajectories.

- $\pi_{\theta_{k+\tau}}$ (short for $p$): the latest policy at the *learner* at time step $k + \tau$, used for gradient updates.

- $\tau(\geq 0)$: *policy staleness*, representing the discrepancy in policy versions between the sampler and the learner, caused by `network delays` and `computational asynchrony`.

- $y$: a trajectory sampled from the stale policy $\pi_{\theta_k}$. $y_t^i$ denotes the $t$-th token of the $i$-th response in a group.

- $r(x, y)$: the reward for response $y$ given input $x$.

- $A(x, y)$: the advantage for response $y$ to input $x$, typically defined as $A(x, y) = r(x, y) - b(x)$, where $b(x)$ is a baseline reward computed for input $x$. In this paper, we use the within-group average reward (Shao et al., 2024) as the baseline $b(x) = \frac{1}{G} \sum_{i=1}^{G} r(x, y^i)$ where $G$ is the group size.

The goal of HeteroRL is to optimize the policy $\pi_\theta$ to maximize the expected cumulative reward. To reduce gradient variance, an advantage function is used, leading to the objective:

$$\mathcal{L}(\theta) = \mathbb{E}_{x \sim \mathcal{D}, y \sim \pi_{\theta_k}(\cdot|x)} \left[ \frac{\pi_{\theta_{k+\boxed{\tau}}}(y|x)}{\pi_{\theta_k}(y|x)} \cdot A(x, y) \right], \tag{1}$$

where $\tau$ is a random variable. For online RL, $\tau \equiv 0$.

## 3 GEPO: GROUP EXPECTATION POLICY OPTIMIZATION

Our method builds upon the group-based policy optimization paradigm of GRPO and introduces the group expectation importance sampling mechanism. We emphasize a paradigm shift from *token-level* to *group-level* importance weighting, which significantly reduces the variance of importance weights and alleviates gradient instability during training.

### 3.1 GROUP EXPECTATION IMPORTANCE WEIGHTING

To enhance the stability of importance weights, we propose the `Group Expectation Importance Weight (GEIW)`, which replaces the individual proposal probability $q(y|x)$ in the standard importance weight $\frac{p(y|x)}{q(y|x)}$ with its group-wise expected value under the current prompt $x$, denoted as $\widehat{\mathbb{E}}_q[q(y|x)]$. Inspired by GRPO, for each input $x$, we generate a group of $G$ responses $\{y^1, \ldots, y^G\} \sim q(\cdot|x)$ to form a sampling group. Since $G$ is typically much smaller than the full policy space and top-$P$/top-$K$ sampling leads to $\sum_{i=1}^{G} q(y^i|x) \gg 1$,

the vector $(q(y^1|x), \dots, q(y^G|x))$ does not constitute a valid probability distribution. Simply using the arithmetic mean $\frac{1}{G} \sum_{i=1}^{G} q(y^i|x)$ would introduce bias due to ignoring the relative sampling probabilities. To obtain a more accurate estimate, we employ a weighted expectation:

$$\widehat{\mathbb{E}}_q[q(y|x)] \approx \sum_{i=1}^{G} \widehat{q(y^i|x)} \cdot q(y^i|x) = \frac{\sum_{i=1}^{G} q(y^i|x)^2}{\sum_{i=1}^{G} q(y^i|x)}, \tag{2}$$

where $\widehat{q(y^i|x)} = \frac{q(y^i|x)}{\sum_{i=1}^{G} q(y^i|x)}$ is the within-group normalized probability, serving as an empirical estimate of the sampling likelihood of each $y_i$. We define the GEIW importance weight as:

$$w_{\text{GEIW}}(y|x) = \frac{p(y|x)}{\widehat{\mathbb{E}}_q[q(y|x)]}. \tag{3}$$

The key advantages of this mechanism are as follows:

**Numerically stable and gradient-effective**: The denominator is decoupled from any single $q(y|x)$, avoiding extreme weight values when individual proposal probabilities approach zero. Although $\texttt{clip}(1 \pm \epsilon)$ can also improve numerical stability, the gradients of the clipped tensors will be set to zero, effectively skipping this data point (ineffective gradient).

**Biased yet low-variance**: By leveraging within-group statistical information, GEIW provides a more reliable scale estimate. Even under large divergence between $p$ and $q$, $\widehat{\mathbb{E}}_q[q(y|x)]$ remains well-conditioned, effectively preventing gradient explosion. Although this estimator introduces a small bias ($w_{\text{GEIW}}$ is a biased estimator), both theoretical analysis (see Theorem 1) and empirical results demonstrate that it significantly reduces variance under high KL divergence, yielding more stable gradient directions and improved training convergence.

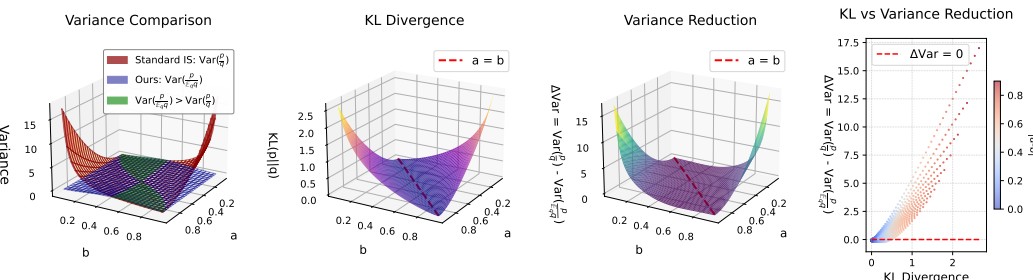

(a) Variance comparison of $\frac{p}{q}$ and $\frac{p}{\mathbb{E}_q[q]}$ under Bernoulli distributions, where $p \sim \text{Bernoulli}(a)$ and $q \sim \text{Bernoulli}(b)$.

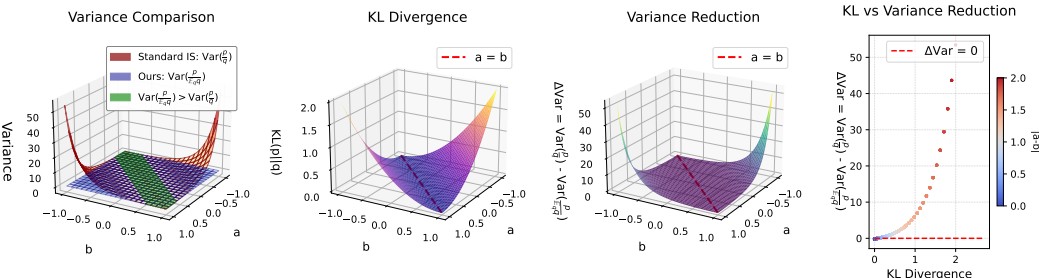

(b) Variance comparison of $\frac{p}{q}$ and $\frac{p}{\mathbb{E}_q[q]}$ under Gaussian distributions, where $p \sim \mathcal{N}(a, 1)$ and $q \sim \mathcal{N}(b, 1)$.

Figure 2: In high-KL regions, $\text{Var}\left[\frac{p(y|x)}{\widehat{\mathbb{E}}_q[q(y|x)]}\right] \ll \text{Var}\left[\frac{p(y|x)}{q(y|x)}\right]$.

**Theorem 1.** *Let $p, q$ be discrete probability distributions. Then there exists a constant $C$ such that:*

$$\mathrm{Var}\left[\frac{p(y|x)}{q(y|x)}\right] - \mathrm{Var}\left[\frac{p(y|x)}{\widehat{\mathbb{E}}_q[q(y|x)]}\right] \geq \boxed{\exp\left(D_{\mathrm{KL}}(p\|q)\right)} - C. \tag{4}$$

*In particular, when $D_{\mathrm{KL}}(p\|q) > \log C$, it holds that $\mathrm{Var}\left[\frac{p(y|x)}{q(y|x)}\right] > \mathrm{Var}\left[\frac{p(y|x)}{\widehat{\mathbb{E}}_q[q(y|x)]}\right]$.*

Theorem 1 shows that GEPO can **exponentially reduce the variance of importance weights**, making it particularly well-suited for heterogeneous RL training under high KL divergence. The full mathematical proof is provided in Appendix A. As shown in Figure 2, we visualize the relationship between KL divergence and importance weight variance when both $p$ and $q$ are Bernoulli or Gaussian distributions with varying parameters. The results indicate that in the high-KL regime, the group expectation approach significantly reduces the variance of importance weights, which benefits training stability under high network latency. Nevertheless, there exist regimes—such as the green regions in the plots—where our method incurs a slight increase in variance.

The difference across all GRPO-like algorithms lies in the computation of the importance weights, as detailed in Listing 1:

```python
if self.loss_type in ["grpo","dr_grpo","bnpo"]: # Token level
    coef_1 = learner_token_p / sampler_token_p
elif self.loss_type == "gspo":  # Sequence level
    coef_1 = learner_seq_p / sampler_seq_p
elif self.loss_type == "gepo": # Group level
    hat_q = sampler_seq_p.detach() / (sampler_seq_p.sum().detach())
    coef_1 = learner_seq_p / (hat_q * sampler_seq_p).sum()
```

Listing 1: Coefficient computation for different policy optimization methods

### 3.2 Gradient Comparison Across Tokens

**What does the GEPO update do?** For a mechanistic understanding of GEPO, it is useful to analyze the gradient of the loss function $\mathcal{L}_{GEPO}$. The equivalent gradient of each token in a group with respect to the parameters $\boldsymbol{\theta}$ of GRPO, GSPO and GEPO can be written as:

$$\frac{\partial \mathcal{L}(\boldsymbol{\theta})}{\partial \boldsymbol{\theta}} = \mathbf{A} \odot \underbrace{\begin{bmatrix} \frac{p'_{1,1}(\boldsymbol{\theta})}{q_{1,1}} & \cdots & \frac{p'_{1,T}(\boldsymbol{\theta})}{q_{1,T}} \\ \vdots & \ddots & \vdots \\ \frac{p'_{G,1}(\boldsymbol{\theta})}{q_{G,1}} & \cdots & \frac{p'_{G,T}(\boldsymbol{\theta})}{q_{G,T}} \end{bmatrix}}_{\text{GRPO}} \text{ or } \underbrace{\begin{bmatrix} \frac{p'_{1,1}(\boldsymbol{\theta})}{q_1} & \cdots & \frac{p'_{1,T}(\boldsymbol{\theta})}{q_1} \\ \vdots & \ddots & \vdots \\ \frac{p'_{G,1}(\boldsymbol{\theta})}{q_G} & \cdots & \frac{p'_{G,T}(\boldsymbol{\theta})}{q_G} \end{bmatrix}}_{\text{GSPO}} \text{ or } \underbrace{\begin{bmatrix} \frac{p'_{1,1}(\boldsymbol{\theta})}{\mathbb{E}_q q} & \cdots & \frac{p'_{1,T}(\boldsymbol{\theta})}{\mathbb{E}_q q} \\ \vdots & \ddots & \vdots \\ \frac{p'_{G,1}(\boldsymbol{\theta})}{\mathbb{E}_q q} & \cdots & \frac{p'_{G,T}(\boldsymbol{\theta})}{\mathbb{E}_q q} \end{bmatrix}}_{\text{GEPO (ours)}},$$

$$\tag{5}$$

where $\mathbf{A} \in \mathbb{R}^{G \times T}$ is token-level advantages matrix, $\odot$ denotes Hadamard product, $q_{i,t} = q(y_t^i \mid x^i, y_{<t}^i)$, $q_i = q(y^i \mid x)$, and $\mathbb{E}_q q = \widehat{\mathbb{E}}_q[q(y|x)]$. From the perspective of gradient stability, GSPO uses a shared denominator $q(y^i \mid x)$ for all tokens in sequence $i$, while GEPO further aggregates across the entire group by using a common denominator $\mathbb{E}_q q$. This progression—from token-level (GRPO) to sequence-level (GSPO) to group-level (GEPO) coefficients—demonstrates that coarser importance-weight granularity significantly reduces gradient variance. Empirically, leveraging group-level statistics enhances robustness and stabilizes training, especially under high policy divergence.

## 4 Experiments

### 4.1 Experimental Setup

**Model, Dataset and Benchmarks** We conduct reinforcement learning training and evaluation on the Qwen3-1.7B/8B model. The models are trained by strong-to-weak distillation (Yang et al., 2025), but have not been tuned with any RL. We train the model on 8,290 samples from the MATH level 3–5 dataset (Zeng et al., 2025) and evaluate it by

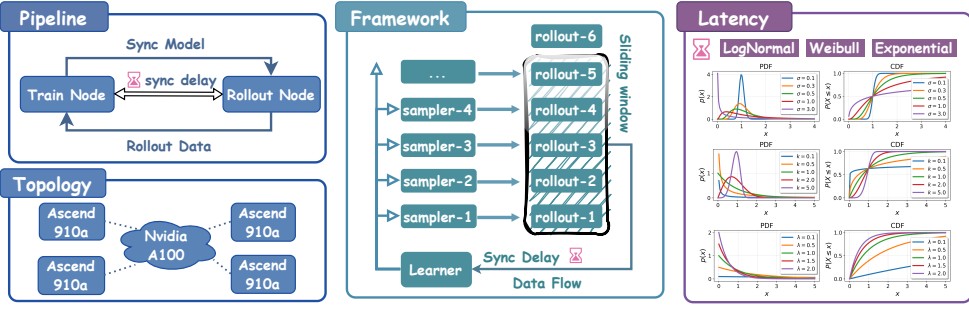

Figure 3: The Overview of HeteroRL. By decoupling sampling and training, HeteroRL enables decentralized distributed RL training of LLMs across five compute nodes: one parameter update node (learner) and four data generation nodes (sampler), forming a star-shaped network topology. Network delays between the sampler and learner nodes are explicitly modeled and can be simulated using stochastic distributions such as the log-normal or Weibull distribution.

reporting average Pass@1 over 8 sampled responses on the **MATH500** (Hendrycks et al., 2021), **AMC23** (Li et al., 2024), **AIME24** (AIME, 2024), and **AIME25** (AIME, 2025) benchmarks. To better evaluate the inherent stability of policy optimization algorithms, we remove KL divergence constraints during training under online RL scenario, and use the same KL coefficient under the heterogeneous RL scenario. We compare our method against baseline methods **GRPO** (Shao et al., 2024), **GSPO** (Zheng et al., 2025), **BNPO** (Xiao et al., 2025) and **Dr.GRPO** (Liu et al., 2025) under both zero-delay and high-delay settings. More experimental details can be found in Section B.1.

**Heterogeneous Computing Environment** As shown in Figure 3, we perform heterogeneous training across five compute nodes: one learner node and four sampler nodes, forming a star-shaped topology centered at the learner. During training, the sampler nodes generate rollout data, which is transmitted over the network to the learner node in a streaming fashion. The learner updates the model parameters and periodically broadcasts the updated weights back to the sampler nodes. The learner processes incoming rollouts in the order they arrive, operating within a fixed time window for data eligibility. Since data is transmitted in batch units—each containing text, generation probabilities, and rewards—a maximum delay of 1800 seconds is sufficient for typical network conditions. Within this window, the iteration gap (in terms of gradient updates) between the learner and samplers remains within 64 steps.

## 4.2 MAIN EXPERIMENTAL RESULTS

Table 1: Performance comparison using Qwen3-1.7B/8B under Online RL (4k limiation).

| Method | AMC2023 | | AIME2024 | | AIME2025 | | MATH500 | | Average | |
|---|---|---|---|---|---|---|---|---|---|---|
| | 1.7B | 8B | 1.7B | 8B | 1.7B | 8B | 1.7B | 8B | 1.7B | 8B |
| Qwen3-1.7/8B | 44.6 | 70.6 | 10.9 | 32.4 | 14.0 | 26.1 | 72.4 | 87.1 | 35.5 | 54.1 |
| **Max Tolerable Delay 0** | | | | | | | | | | |
| BNPO | 59.4 | 78.8 | 27.7 | **44.1** | 23.4 | 29.3 | 83.7 | 91.4 | 48.6 | 60.9 |
| Dr.GRPO | 61.6 | 77.5 | 24.6 | 41.0 | 22.7 | 27.7 | 82.9 | 91.6 | 48.0 | 59.4 |
| GRPO | 60.9 | 81.3 | 30.9 | 42.6 | 24.2 | 31.3 | 83.7 | 92.0 | 49.9 | 61.8 |
| GSPO | 60.3 | 77.8 | 28.5 | 41.8 | 25.0 | 31.3 | 83.9 | 90.9 | 49.4 | 60.5 |
| GEPO (ours) | **62.2** | **85.6** | **31.6** | **44.1** | **25.8** | **37.5** | **84.7** | **92.6** | **51.1** | **65.0** |

In this section, we compare GEPO with baselines under online RL and Hetero RL settings. The experimental results in Tables 1 and 2 demonstrate that GEPO not only achieves

superior performance but also exhibits exceptional stability across both online and Hetero RL settings. Below, we dissect these findings in depth.

Table 2: Performance comparison using Qwen3-8B under Hetero RL (4k limiation).

| Method | AMC2023 | | AIME2024 | | AIME2025 | | MATH500 | | Average | |
|---|---|---|---|---|---|---|---|---|---|---|
| | best | last | best | last | best | last | best | last | best | last |
| Qwen3-8B | 70.6 | - | 32.4 | - | 26.1 | - | 87.1 | - | 54.1 | - |
| **Max Tolerable Delay 64** | | | | | | | | | | |
| BNPO | 73.2 | 72.6 | 37.7 | 35.9 | 25.3 | 24.5 | 87.4 | 86.1 | 55.9 | 54.8 |
| Dr.GRPO | 73.1 | 72.9 | 38.1 | 37.3 | 24.8 | 22.3 | 88.1 | 86.7 | 56.0 | 54.8 |
| GRPO | 71.6 | 71.6 | 38.7 | 35.9 | 27.3 | 27.0 | 88.8 | 88.8 | 56.6 | 55.8 |
| GSPO | 76.2 | 60.0 | 37.9 | 16.4 | 28.9 | 27.3 | 90.7 | 81.9 | 58.4 | 46.4 |
| GEPO (ours) | **83.4** | **82.8** | **42.6** | **37.5** | **33.2** | **32.0** | **91.3** | **90.9** | **62.6** | **60.8** |

In the **online RL setting** (Table 1), GEPO consistently outperforms all baselines across both model sizes and all benchmarks. On Qwen3-8B, it achieves an average score of **65.0**, surpassing GRPO (61.8) and GSPO (60.5) by 3.2 and 4.1 points, respectively. The gain is most notable on AIME2025 (+6.2 points over GRPO/GSPO, 20% relative improvement). Even on the 1.7B model, GEPO sets a new SOTA, exceeding the best baseline by 1.5 points in average, confirming that group-level importance weighting improves gradient quality even without asynchrony.

In the **Hetero RL setting** (Table 2), GEPO's stability advantage becomes decisive. We find that BNPO and Dr.GRPO exhibit little performance difference compared to vanilla GRPO, and both employ token-level importance weights; therefore, we only analyze the performance differences among GRPO, GSPO, and GEPO. Both GEPO and GSPO improve over GRPO in `best` performance (+10.6% and +3.2%, respectively). However, GEPO further surpasses GSPO by 7.2% in accuracy while reducing its best-to-last degradation by **85%** versus GSPO ($\Delta$=1.8 vs. 12.0), achieving both higher performance and far greater stability. While GSPO's `last` scores collapse dramatically, GEPO maintains near-peak performance throughout training.

These results validate GEPO's core design: by replacing token or sequence-level importance weights with group-level expectations, it exponentially reduces importance weight variance under high KL divergence (Theorem 1), enabling stable, scalable decentralized RL. GEPO thus sets a new frontier in both performance and stability across ideal and real-world distributed settings. In the HeteroRL setting, the training process recorded in Figure 4 shows

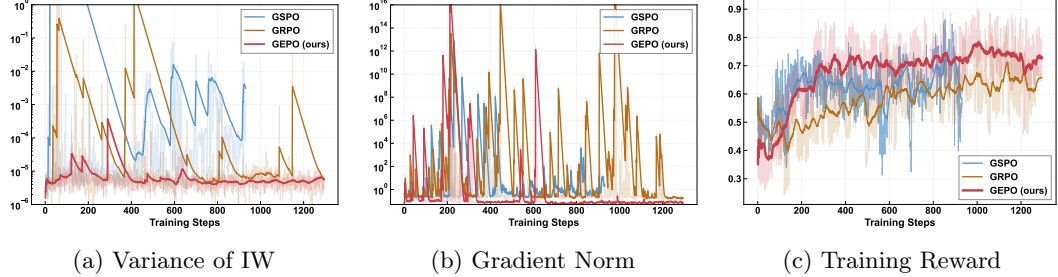

(a) Variance of IW          (b) Gradient Norm          (c) Training Reward

Figure 4: Curves of importance sampling variance, training gradient norm, and train/eval reward under max delay 64. Compared to GRPO and GSPO, GEPO maintains more stable importance weight variance, resulting in less drastic gradient changes, more stable training, and no decline in training reward.

that GRPO stably improves the reward at a slower pace, while GSPO rapidly increases the reward in the first 200 steps but becomes unstable between 500 and 700 steps. As seen in Figure 4a, GEPO exhibits significantly lower variance in importance weights compared to

GRPO and GSPO, which experience sharp spikes and fluctuations. These unstable weight variances lead to erratic gradient updates, as evidenced by the large oscillations in gradient norm (Figure 4b) for GRPO and GSPO, especially during early and mid-training phases. In contrast, GEPO's gradient norms remain relatively smooth and bounded, contributing to stable learning progress. Consequently, the training reward curve (Figure 4c) shows consistent improvement for GEPO without any noticeable decline, whereas GRPO and GSPO exhibit periods of stagnation or even degradation.

## 4.3 LATENCY ANALYSIS

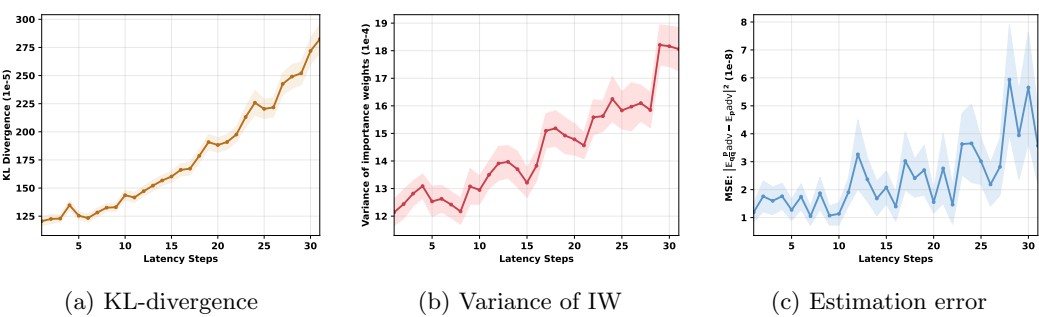

(a) KL-divergence        (b) Variance of IW        (c) Estimation error

Figure 5: KL-divergence, Variance of IW, and Estimation error are all positively correlated with the number of delay steps.

**Impact of Latency** As shown in Figure 5, we analyze the changes in KL divergence between the trainer and sampler, variance of importance weights, and estimation error of the expected value of the advantage function (optimization objective) during heterogeneous RL training as latency increases. We observe that latency leads to increased KL divergence (Figure 5a), which in turn causes an increase in the variance of importance weights (Figure 5b), ultimately resulting in increased estimation error of the expected advantage function (Figure 5c). Since the optimization objective is to maximize the estimated expectation of the advantage function, large estimation errors will cause significant fluctuations in gradients, thereby affecting training stability and performance. To show that high latency harms training stability, we compare max delays of 8 and 64 steps. As Figure 6 shows, with 64-step delay—especially near step 800—the KL divergence spikes and evaluation accuracy drops sharply, confirming that latency induces instability. Although GEPO improves stability, it still suffers a performance dip around step 900, highlighting that heterogeneous RL under high latency remains challenging.

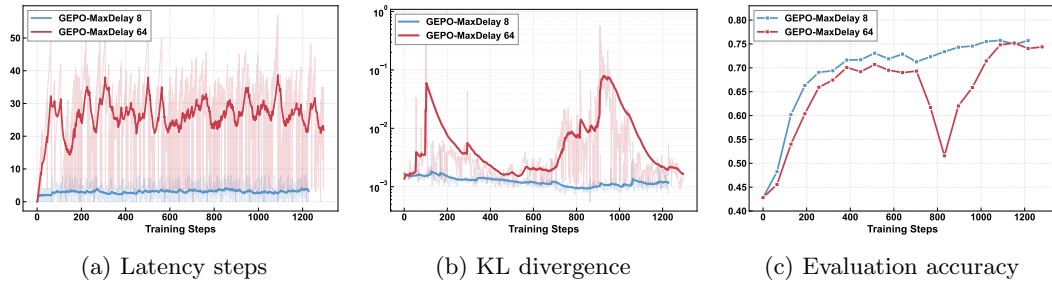

(a) Latency steps        (b) KL divergence        (c) Evaluation accuracy

Figure 6: Training processes under different latency conditions

**Correlation and Causality.** Figure 7 quantifies the pairwise correlations among KL-divergence, variance of importance weights, and estimation error of the expected advantage function. The correlation coefficients range from 0.76 to 0.96 ($\alpha = 0.05$), confirming a strong statistical association between these variables. This observation empirically supports our hypothesis (illustrated in Figure 5) that increased latency induces higher KL divergence, which in turn amplifies the variance of importance weights and the estimation error, ultimately threatening training stability. However, correlation does not imply strict causation.

While latency is a significant contributing factor to KL divergence, it is not the sole determinant — the model's internal state and the statistical properties of the sampled data also play crucial roles. This explains the observed variance in training outcomes under identical latency: sometimes collapse occurs, sometimes not. Critically, what determines survival versus collapse is not latency itself, but the algorithm's capacity to mitigate the downstream instability caused by high KL divergence. As demonstrated in Figure 4 and Table 2, GEPO's group expectation mechanism effectively suppresses the explosion of importance weight variance even when KL divergence is high. This allows GEPO to maintain stable training and avoid collapse in many scenarios where baseline methods (like GRPO and GSPO) fail — thereby establishing algorithmic robustness to policy divergence as a core contribution of this work.

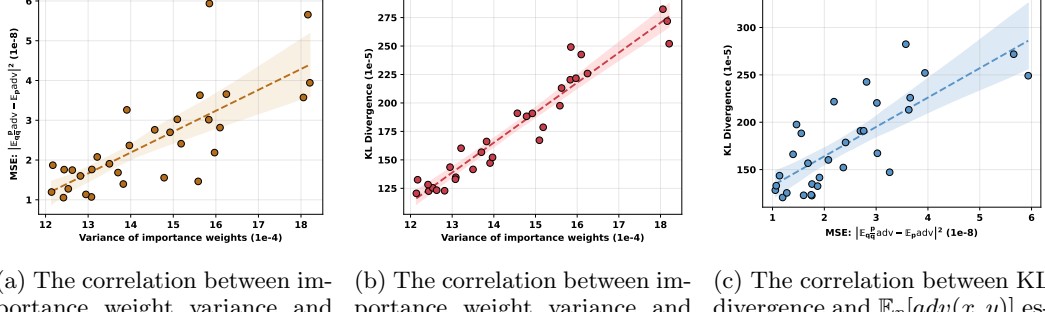

(a) The correlation between importance weight variance and estimation error of $\mathbb{E}_p[adv(x, y)]$ is **0.76**.

(b) The correlation between importance weight variance and KL divergence is **0.96**.

(c) The correlation between KL divergence and $\mathbb{E}_p[adv(x, y)]$ estimation error is **0.78**.

Figure 7: Correlation analysis (**95% CI**) of training delay steps, importance sampling variance, and estimation error of expected advantage function.

## 4.4 Hyperparameter Analysis

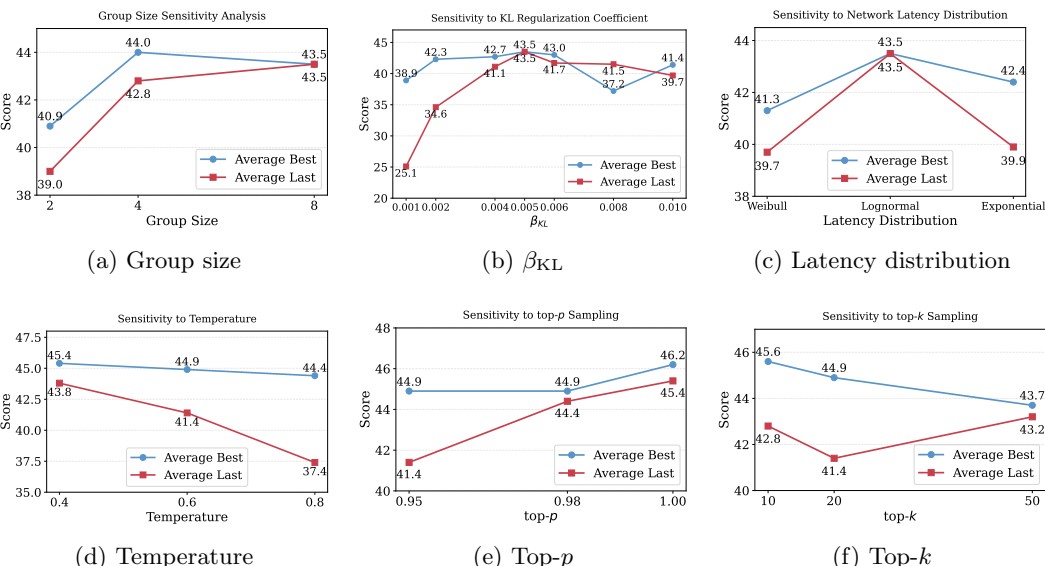

(a) Group size

(b) $\beta_{\mathrm{KL}}$

(c) Latency distribution

(d) Temperature

(e) Top-$p$

(f) Top-$k$

Figure 8: Sensitivity analysis of key hyperparameters in the Hetero/Online RL setting. All plots show the **Average Best** and **Average Last** performance across four benchmarks (AMC2023, AIME2024, AIME2025, MATH500).

We conduct comprehensive hyperparameter studies to analyze the sensitivity of GEPO to key hyperparameters under hetero RL and online RL setting, including group size $G$, sampling strategies (top-$k$, top-$p$, and temperature), the KL regularization coefficient $\beta_{\mathrm{KL}}$, and

the choice of latency simulation distribution (e.g., exponential, Weibull, or lognormal). All experiments in the hyperparameter analysis section are based on the Qwen3-1.7B model with a maximum context length of 2,048 tokens. Figure 8 plots the best and last performance curves across six hyperparameters. More detailed experimental results are provided in Appendix B.5.

In summary, our comprehensive hyperparameter analysis reveals that group size 8 ensures optimal stability in heterogeneous RL settings with consistently high final performance (Figure 8a), while a KL regularization coefficient of $\beta_{\mathrm{KL}} = 0.005$ maximizes both peak and final scores (Figure 8b). An excessively low KL coefficient reduces training stability. The Weibull latency distribution proves more challenging than lognormal or exponential alternatives (Figure 8c), and a temperature of 0.4 enhances stability by maintaining higher final performance (Figure 8d). For sampling strategies, top-$p = 1.0$ and top-$k = 10$ achieve the best trade-off between peak and final results (Figures 8e, 8f), confirming their effectiveness across diverse training scenarios. These findings validate our design choices in the main experiments and provide practical guidance for deploying GEPO in diverse distributed training environments.

## 5 Related Work

**Asynchronous Reinforcement Learning Framework.** In recent years, reinforcement learning has become central to post-training LLMs (Ziegler et al., 2019). Researchers have identified efficiency bottlenecks in traditional synchronous RL frameworks. Wu et al. (2025) first theoretically explored asynchronous RLHF, proposing to decouple generation and training across GPU clusters, enabling scalable and efficient RL fine-tuning of LLMs with provable speedup over synchronous baselines. To address practical efficiency, AREAL (Fu et al., 2025) fully decouples generation and training, using staleness thresholds and a decoupled PPO objective (Hilton et al., 2022) to handle outdated samples. Prime Intellect (Team et al., 2025) offers a decentralized, asynchronous framework for community compute, ensuring trust via verifiable inference, stability via two-sided GRPO clipping, and controllable reasoning with length-aware rewards. Nevertheless, these approaches all assume centralized, low-latency infrastructure and support only limited degrees of asynchrony, making them not suited for distributed RL in high-latency, decentralized, and heterogeneous compute environments.

**Asynchronous Reinforcement Learning Algorithm.** Major of asynchronous or off-policy RL algorithms for LLMs post-training typically adopt two primary strategies to ensure training stability: (1) **Gradient Truncation**, wherein updates are suppressed for tokens whose importance weight lie beyond a specified trust interval—Decoupled PPO (Hilton et al., 2022) exemplifies this approach; (2) **Importance Weight Truncation**, which retains contributions from all samples but mitigates variance by bounding or reshaping the importance weights—methods such as Truncated IS (Espeholt et al., 2018), CISPO (Chen et al., 2025), and TOPR (Roux et al., 2025) fall into this category. Our method diverges from prior studies in its underlying perspective: by embracing the notion of intra-group expectation, we broaden the tolerance for token importance weight ranges, thereby improving training stability.

## 6 Conclusion

We propose HeteroRL, a heterogeneous reinforcement learning framework designed for training LLMs across geographically distributed and resource-heterogeneous nodes, paired with GEPO—a novel policy optimization algorithm that stabilizes training under high latency. By decoupling rollout sampling from parameter updates, HeteroRL eliminates synchronization bottlenecks inherent in traditional RL pipelines. GEPO addresses the explosion of variance of importance weight caused by stale policies through group expectation importance weight, provably reducing variance exponentially, particularly under large KL divergence between the sampling and learning policies. This work establishes a practical foundation for scalable, delay-tolerant decentralized RL, making it well-suited for real-world LLM post-training in heterogeneous, wide-area network environments.

AUTHOR CONTRIBUTIONS

The division of labor within the research team is as follows: **Han Zhang** was responsible for the design of the GEPO algorithm, preliminary implementation of the HeteroRL training framework, and technical report writing. **Ruibin Zheng** focused on optimizing the implementation of the HeteroRL training framework, improving data transmission execution logic in simulated latency scenarios, tuning model hyperparameters and prompts, creating visualizations, conducting model performance evaluations and participating in algorithm analysis. **Zexuan Yi** was responsible for implementing data transmission in real network environments and engineering the model synchronization mechanism. **Zhuo Zhang** contributed to experimental analysis and experimental design.

ACKNOWLEDGMENTS

This work is supported by the Major Key Project of PCL (Grant No. PCL2025A09) and by the National Natural Science Foundation of China 62576120. We thank **Xiang Li** for verifying the theoretical proofs in this paper, and **Wei Li** for providing technical support on hardware infrastructure and system maintenance.

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

CONTENTS

## A    Theoretical Extension

### A.1    Theoretical Proof of Importance Sampling Variance

This appendix analyzes a newly proposed importance sampling weight $w_{\mathrm{new}}(y|x) = \dfrac{p(y|x)}{\mathbb{E}_q[q]}$, where $\mathbb{E}_q[q] = \int q(y|x)^2 dy$, and compares its variance with the standard importance sampling weight $w_{\mathrm{std}}(y|x) = \dfrac{p(y|x)}{q(y|x)}$.

**Problem Setting**    Let the target distribution be $p(y|x)$ and the proposal distribution be $q(y|x)$. We aim to estimate:

$$\mathbb{E}_p[f] = \int f(x)p(y|x)dy. \tag{6}$$

Since direct sampling from $p$ is difficult, we employ importance sampling by drawing samples from $q$.

**Standard Importance Sampling**    The standard weight is defined as:

$$w_{\mathrm{std}}(y|x) = \frac{p(y|x)}{q(y|x)}. \tag{7}$$

Its expectation under $q$ is:

$$\mathbb{E}_q\left[\frac{p(y|x)}{q(y|x)}\right] = \int \frac{p(y|x)}{q(y|x)}q(y|x)dy = \int p(y|x)dy = 1, \tag{8}$$

thus it is unbiased. Its variance is:

$$\begin{aligned}
\mathrm{Var}_q(w_{\mathrm{std}}) &= \mathbb{E}_q\left[\left(\frac{p}{q}\right)^2\right] - \left(\mathbb{E}_q\left[\frac{p}{q}\right]\right)^2 \\
&= \int \frac{p(y|x)^2}{q(y|x)}dy - 1.
\end{aligned} \tag{9}$$

Denoted as:

$$\mathrm{Var}_{\mathrm{std}} = \int \frac{p(y|x)^2}{q(y|x)}dy - 1. \tag{10}$$

**Group Expectation Importance Sampling**    The new weight is defined as:

$$w_{\mathrm{new}}(y|x) = \frac{p(y|x)}{\mathbb{E}_q[q]}, \quad \text{where} \quad \mathbb{E}_q[q] = \int q(y|x)^2 dy. \tag{11}$$

Its expectation is:

$$\mathbb{E}_q[w_{\mathrm{new}}] = \frac{1}{\mathbb{E}_q[q]} \int p(y|x)q(y|x)dy = \frac{\langle p, q \rangle}{\|q\|_2^2}, \tag{12}$$

where $\langle p, q \rangle$ denotes the inner product $\int p(y|x)q(y|x)dy$. Generally, $\langle p, q \rangle \neq \|q\|_2^2$, making this estimator biased. Its variance is:

$$\begin{aligned}
\mathrm{Var}_q(w_{\mathrm{new}}) &= \mathbb{E}_q\left[\left(\frac{p(y|x)}{\mathbb{E}_q[q]}\right)^2\right] - \left(\mathbb{E}_q\left[\frac{p(y|x)}{\mathbb{E}_q[q]}\right]\right)^2 \\
&= \frac{1}{(\mathbb{E}_q[q])^2}\left(\int p(y|x)^2 q(y|x)dy - \left(\int p(y|x)q(y|x)dy\right)^2\right).
\end{aligned} \tag{13}$$

Denoted as:

$$\mathrm{Var}_{\mathrm{new}} = \frac{1}{\left(\int q(y|x)^2 dy\right)^2}\left(\int p(y|x)^2 q(y|x)dy - \left(\int p(y|x)q(y|x)dy\right)^2\right). \tag{14}$$

**Variance Comparison**   We compare:

$$
\boxed{
\begin{aligned}
\mathrm{Var}_{\mathrm{std}} &= \int \frac{p(y|x)^2}{q(y|x)} dy - 1, \\
\mathrm{Var}_{\mathrm{new}} &= \frac{1}{\left(\int q(y|x)^2 dy\right)^2} \left( \int p(y|x)^2 q(y|x) dy - \left( \int p(y|x) q(y|x) dy \right)^2 \right).
\end{aligned}
}
\tag{15}
$$

A.1 VARIANCE COMPARISON IN DISCRETE SPACE

Since the action space of large models is discrete, this section discusses the variance difference $\Delta = \mathrm{Var}_{\mathrm{std}} - \mathrm{Var}_{\mathrm{new}}$ in discrete probability space. The integral expressions in the continuous case naturally transition to discrete summation forms:

- Replace continuous integrals $\int \cdot dy$ with discrete summations $\sum_{i=1}^n$;
- Replace probability density functions $p(y|x)$, $q(y|x)$ with probability masses $p_i$, $q_i$;
- Maintain the structural form of variance expressions unchanged.

**Notation and Setting**   Let the sample space be a finite set $\mathcal{X} = \{1, 2, \ldots, n\}$, where $n \geq 2$. Let $p = (p_1, \ldots, p_n)$ and $q = (q_1, \ldots, q_n)$ be two probability distributions satisfying:

- $p_i > 0$, $\sum_{i=1}^n p_i = 1$,
- $q_i > 0$, $\sum_{i=1}^n q_i = 1$.

Define the following four key quantities, corresponding to the integral terms in the continuous expressions:

$$
A = \left( \sum_{i=1}^n q_i^2 \right)^2, \qquad \text{(corresponding to } \left( \int q(y|x)^2 dy \right)^2 \text{)} \tag{16}
$$

$$
B = \left( \sum_{i=1}^n p_i q_i \right)^2, \qquad \text{(corresponding to } \left( \int p(y|x) q(y|x) dy \right)^2 \text{)} \tag{17}
$$

$$
I_1 = \sum_{i=1}^n \frac{p_i^2}{q_i}, \qquad \text{(corresponding to } \int \frac{p(y|x)^2}{q(y|x)} dy \text{)} \tag{18}
$$

$$
I_2 = \sum_{i=1}^n p_i^2 q_i, \qquad \text{(corresponding to } \int p(y|x)^2 q(y|x) dy \text{)} \tag{19}
$$

Accordingly, the variance difference can be written as:

$$
\Delta = I_1 + \frac{B - A - I_2}{A}. \tag{20}
$$

**Lemma 1** (Range of Quantities). *Under the above setting, we have:*

- $A \in \left[ \dfrac{1}{n^2}, \ 1 \right]$,
- $B \in [0, \ 1]$,
- $I_1 \in [1, \ \infty)$,
- $I_2 \in (0, \ 1]$.

*Proof.*
**1. Range of $A$:** By the power mean inequality, $\sum q_i^2 \geq \frac{1}{n}$, and $\sum q_i^2 \leq 1$, thus $A \in [1/n^2, 1]$.
**2. Range of $B$:** $\sum p_i q_i \in (0, 1]$, thus $B \in [0, 1]$.
**3. Range of $I_1$:** By the Cauchy-Schwarz inequality, $I_1 \geq 1$, and it can approach infinity.
**4. Range of $I_2$:** Since $p_i^2 \leq p_i$, we have $I_2 \leq 1$, and $I_2 > 0$. $\qquad \square$

**Theorem 1.** Let $p, q$ be discrete probability distributions. Then there exists a constant $C$ such that:

$$\text{Var}\left[\frac{p(y|x)}{q(y|x)}\right] - \text{Var}\left[\frac{p(y|x)}{\widehat{\mathbb{E}}_q[q(y|x)]}\right] \geq \exp\left(D_{\text{KL}}(p\|q)\right) - C. \tag{21}$$

In particular, when $D_{\text{KL}}(p\|q) > \log C$, it holds that $\text{Var}[\frac{p(y|x)}{q(y|x)}] > \text{Var}[\frac{p(y|x)}{\widehat{\mathbb{E}}_q[q(y|x)]}]$.

*Proof.* **Step 1.** From the fundamental inequality relationship between KL divergence and $\chi^2$ divergence (Pinsker's inequality):

$$D_{\text{KL}}(p\|q) \leq \log\left(1 + D_{\chi^2}(p\|q)\right), \tag{22}$$

where the chi-square divergence is defined as:

$$D_{\chi^2}(p\|q) = \sum_{i=1}^{n} \frac{(p_i - q_i)^2}{q_i} = \sum_{i=1}^{n} \frac{p_i^2}{q_i} - 1 = I_1 - 1. \tag{23}$$

Substituting yields:

$$D_{\text{KL}}(p\|q) \leq \log(I_1), \tag{24}$$

therefore:

$$I_1 \geq \exp\left(D_{\text{KL}}(p\|q)\right). \tag{25}$$

**Step 2.** From Lemma 1, we know that $A$, $B$, and $I_2$ satisfy the following bounds:

$$A \in [\frac{1}{n^2}, 1], \tag{26}$$

$$B \in [0, 1], \tag{27}$$

$$I_2 \in (0, 1]. \tag{28}$$

Consider the lower bound of the expression $\dfrac{B - A - I_2}{A}$. To obtain its minimum value, we take:

- Minimum value of $B$: $B = 0$
- Minimum value of $A$: $A = \frac{1}{n^2}$
- Maximum value of $I_2$: $I_2 = 1$

Substituting yields:

$$\frac{B - A - I_2}{A} \geq \frac{0 - 1/n^2 - 1}{1/n^2} = -(n^2 + 1). \tag{29}$$

**Step 3.** Substituting inequalities (1) and (2) into the expression for $\Delta$:

$$\Delta = I_1 + \frac{B - A - I_2}{A} \geq \exp\left(D_{\text{KL}}(p\|q)\right) - (n^2 + 1). \tag{30}$$

When $D_{\text{KL}}(p\|q) > \log(n^2 + 1)$, we have:

$$\exp\left(D_{\text{KL}}(p\|q)\right) > n^2 + 1, \tag{31}$$

thus:

$$\Delta > 0, \tag{32}$$

i.e., $\text{Var}_{\text{std}} > \text{Var}_{\text{new}}$. $\qquad\square$

**Corollary 1.** *In discrete space, if $D_{KL}(p\|q) > \log(n^2 + 1)$, then:*

$$\text{Var}_{std} > \text{Var}_{new}, \tag{33}$$

*i.e., the new estimator has smaller variance.*

**It should be noted that $A$ only attains the value $\frac{1}{n^2}$ when $q$ follows a uniform distribution.** In practice, when large models generate responses, the distribution tends to be long-tailed, so the value of $A$ is much greater than $\frac{1}{n^2}$, and the constant $C_{real} \ll \log(n^2 + 1)$. For example, we randomly generated 128 tokens using Qwen3-1.7B ($n = 151936$), and the standard variance and average of $A$ was $0.432_{\pm 0.36} \gg \frac{1}{n^2}$.

## A.2 BIAS AND VARIANCE ANALYSIS

### A.2.1 BIAS ANALYSIS

**The end-to-end bias of GEPO** We compute the following bias of GEPO and GSPO:

$$
\begin{aligned}
\text{Bias(GEPO)} &= \left| \mathbb{E}_p \left[ A(x, y) \right] - \mathbb{E}_q \left[ \frac{p(y|x)}{\mathbb{E}_q q} A(x, y) \right] \right| \\
&= \left| 0 - \mathbb{E}_q \left[ \frac{p(y|x)}{\mathbb{E}_q q} A(x, y) \right] \right| \approx \left| \frac{1}{G} \sum_{i=1}^{G} \left[ \frac{p(y^i|x)}{\mathbb{E}_q q} A(x, y^i) \right] \right|.
\end{aligned}
\tag{34}
$$

$$
\begin{aligned}
\text{Bias(GSPO)} &= \left| \mathbb{E}_p \left[ A(x, y) \right] - \mathbb{E}_q \left[ \frac{p(y|x)}{q(y|x)} A(x, y) \right] \right| \\
&= \left| 0 - \mathbb{E}_q \left[ \frac{p(y|x)}{q(y|x)} A(x, y) \right] \right| \approx \left| \frac{1}{G} \sum_{i=1}^{G} \left[ \frac{p(y^i|x)}{q(y^i|x)} A(x, y^i) \right] \right|.
\end{aligned}
\tag{35}
$$

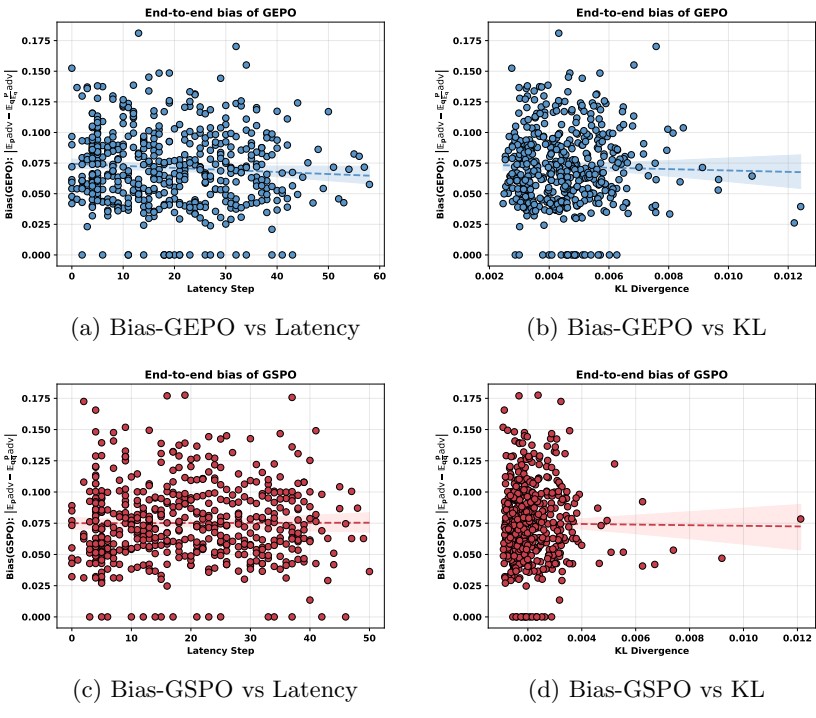

(a) Bias-GEPO vs Latency        (b) Bias-GEPO vs KL

(c) Bias-GSPO vs Latency        (d) Bias-GSPO vs KL

Figure 9: The Bias of GEPO and GSPO. Over 512 global training steps, $Bias(GEPO)$ has a mean of 0.072 and variance of 0.001, while $Bias(GSPO)$ has a mean of 0.075 and variance of 0.001.

Here, the bias of GSPO serves as a baseline for assessing the relative magnitude of the bias in GEPO. The use of the $\approx$ symbol is due to our use of Monte Carlo estimation for this expected statistic. Figure 9 plots the bias of GEPO and GSPO according to latency steps and KL-divergence.

**The theoretical bound of GEPO bias** We derive an upper bound for the **bias** introduced by GEPO's group expectation importance weighting, specifically the absolute difference between the standard importance sampling (IS) estimator and GEPO's estimator **in expectation**, under the assumption that the advantage function $A(x, y)$ is **zero-mean under the target policy** $p$ (i.e., $\mathbb{E}_p[A] = 0$).

**Theorem 2** (Bias of GEPO). *Let $p(y|x)$ and $q(y|x)$ be discrete probability distributions, and let $A(x, y)$ be a bounded advantage function. The bias of GEPO can be bounded by $\frac{\|p\|_2}{\|q\|_2}$:*

$$Bias(GEPO) = \left| \mathbb{E}_p \left[ A(x, y) \right] - \mathbb{E}_q \left[ \frac{p(y|x)}{\mathbb{E}_q[q]} A(x, y) \right] \right| < \frac{\|p\|_2}{\|q\|_2}, \tag{36}$$

*where $\frac{\|p\|_2}{\|q\|_2}$ denotes the $L^2$-norm ratio of $p$ and $q$.*

In LLM reinforcement learning, the ratio $\frac{\|p\|_2}{\|q\|_2}$—which upper-bounds the bias introduced by GEPO's group expectation weighting—remains modest in practice. Although this ratio can theoretically reach $\sqrt{n}$ when $p$ is one-hot and $q$ is uniform over $n$ actions, real-world LLM policies are highly non-uniform due to top-$k$/top-$p$ sampling and sharp output distributions. Empirical measurements in the paper (e.g., Appendix A) show that $\|q\|_2^2 \approx 0.432$ for Qwen3-1.7B, implying $\|q\|_2 \approx 0.657$, and $p$ typically exhibits similar concentration. Consequently, $\frac{\|p\|_2}{\|q\|_2} \approx 1$ in general, even under high policy divergence, ensuring that GEPO's bias remains mild and controllable—far from the catastrophic scale implied by worst-case theoretical bounds.

**The Proof Precess**

**Step 1: Define Estimators**   We adopt the notation and meanings from Section 2. Without loss of generality, we assume that $|A| \leq 1$.

We define:

- Standard IS estimator expectation:

$$\mu_1 = \mathbb{E}_q \left[ \frac{p(y|x)}{q(y|x)} \cdot A(x, y) \right] = \sum_y p(y|x) \cdot A(x, y) = \mathbb{E}_p[A] = 0. \tag{37}$$

- GEPO estimator expectation (using $\hat{q} = \mathbb{E}_q[q] = \sum_y q(y|x)^2 = \|q\|_2^2$):

$$\mu_2 = \mathbb{E}_q \left[ \frac{p(y|x)}{\mathbb{E}_q[q]} \cdot A(x, y) \right] = \frac{1}{\|q\|_2^2} \sum_y p(y|x) q(y|x) \cdot A(x, y). \tag{38}$$

Thus, the **bias** is:

$$Bias(GEPO) = |\mu_1 - \mu_2| = |\mu_2| = \left| \frac{1}{\|q\|_2^2} \sum_y p(y|x) q(y|x) \cdot A(x, y) \right|. \tag{39}$$

**Step 2: Apply Cauchy–Schwarz Inequality**   We bound the absolute value of the inner product by Cauchy–Schwarz:

$$\left| \sum_y p(y|x) q(y|x) \cdot A(x, y) \right| \leq \sqrt{\sum_y (p(y|x) \cdot A(x, y))^2} \cdot \sqrt{\sum_y q(y|x)^2}. \tag{40}$$

Since $|A(x, y)| \leq 1$, we have $A(x, y)^2 \leq 1$, so:

$$\sum_y (p(y|x) \cdot A(x, y))^2 < \sum_y p(y|x)^2 = \|p\|_2^2. \tag{41}$$

The reason we use $<$ instead of $\leq$ here is that the case "advantage $|A(x, y)| = 1$ for all possible LLM responses $y$" cannot occur.

Therefore:

$$\left| \sum_y p(y|x) q(y|x) \cdot A(x, y) \right| < \|p\|_2 \cdot \|q\|_2. \tag{42}$$

**Step 3: Combine with Denominator**  Now plug into the expression for $|\mu_2|$:

$$|\mu_2| = \left| \frac{1}{\|q\|_2^2} \sum_y p(y|x)q(y|x) \cdot A(x,y) \right| < \frac{\|p\|_2 \cdot \|q\|_2}{\|q\|_2^2} = \frac{\|p\|_2}{\|q\|_2}. \tag{43}$$

**Finally:**

$$\text{Bias(GEPO)} < \frac{\|p\|_2}{\|q\|_2}. \tag{44}$$

This upper bound quantifies the **bias** of GEPO's estimator relative to the unbiased standard IS estimator. In large language models, both $\|p\|_2$ and $\|q\|_2$ are typically on the order of $\sqrt{0.4} \approx 0.63$, so their ratio remains close to 1—indicating **mild, controllable bias**.

### A.2.2  FUTHER VARIANCE ANALYSIS

**Variance Analysis of Advantage-Weighted Importance Sampling**  We propose the extended Theorem 1 for $\text{Var}\left[ \frac{p(y|x)}{\mathbb{E}_q[q(y|x)]} \cdot A(x,y) \right]$ as follows:

**Theorem 3** (Variance of GEPO). *Let $p(y|x)$ and $q(y|x)$ be discrete probability distributions, and let $A(x,y)$ be a bounded advantage function. Assume that on the support of interest, $|A(x,y)| \geq A_{\min} > 0$, and define $A_{\max} = \max_y |A(x,y)|$. Then there exists a constant $C_{\text{adv}} = \dfrac{A_{\max}^2}{\|q\|_2^2}$ such that:*

$$\text{Var}_q\left[ A(x,y) \cdot \frac{p(y|x)}{q(y|x)} \right] - \text{Var}_q\left[ A(x,y) \cdot \frac{p(y|x)}{\mathbb{E}_q[q(y|x)]} \right] \geq \boxed{A_{\min}^2 \cdot \exp\left(D_{\text{KL}}(p\|q)\right)} - C_{\text{adv}}. \tag{45}$$

*In particular, when $D_{\text{KL}}(p\|q) > \log\left( \dfrac{C_{\text{adv}}}{A_{\min}^2} \right)$, it holds that*

$$\text{Var}_q\left[ A(x,y) \cdot \frac{p(y|x)}{q(y|x)} \right] > \text{Var}_q\left[ A(x,y) \cdot \frac{p(y|x)}{\mathbb{E}_q[q(y|x)]} \right].$$

**About $A_{min} > 0$ :**  We first note that any sample $y$ with $A(x,y) = 0$ contributes **zero gradient** to the policy update and **requires no variance control**, as it does not affect learning. Therefore, without loss of generality, we can remove the training samples for which $A(x,y) = 0$. Our variance analysis focuses only on the support where $|A(x,y)| > 0$. Since the advantage function is assumed bounded and not identically zero, we can further restrict to the subset where $A(x,y)^2 \geq A_{\min}^2 > 0$ for some constant $A_{\min}$. Within the **Group Wise Advantage** method like GRPO, $A_{\min} = 1/G$, where $G$ is the group size.

The theoretical proof is as follows:

Let $p(y|x)$, $q(y|x)$ be discrete probability distributions and $A(x,y)$ be a bounded advantage function satisfying $\mathbb{E}_p[A(x,y)] = 0$. For notational simplicity, we omit the dependence on $x$.

**Step 1: Variance Expressions**  Following the notation in Appendix A, we define: $Z_1 = A(x,y) \cdot \frac{p(y|x)}{q(y|x)}$, $Z_2 = A(x,y) \cdot \frac{p(y|x)}{\|q\|_2^2}$, where $\|q\|_2^2 = \mathbb{E}_q[q] = \int q(y|x)^2 \, dy$.

The variances are:

$$\begin{aligned}
\text{Var}_q(Z_1) &= \mathbb{E}_q\left[ \left( A(x,y) \cdot \frac{p(y|x)}{q(y|x)} \right)^2 \right] - \left( \mathbb{E}_q\left[ A(x,y) \cdot \frac{p(y|x)}{q(y|x)} \right] \right)^2 \\
&= \int A(x,y)^2 \frac{p(y|x)^2}{q(y|x)} \, dy - \left( \int A(x,y)p(y|x) \, dy \right)^2,
\end{aligned} \tag{46}$$

$$\begin{aligned}
\text{Var}_q(Z_2) &= \frac{1}{\|q\|_2^2} \left[ \mathbb{E}_q[A(x,y)^2 p(y|x)^2] - (\mathbb{E}_q[A(x,y)p(y|x)])^2 \right] \\
&= \frac{1}{\|q\|_2^2} \left[ \int A(x,y)^2 p(y|x)^2 q(y|x) \, dy - \left( \int A(x,y)p(y|x)q(y|x) \, dy \right)^2 \right].
\end{aligned} \tag{47}$$

**Step 2: Variance Difference** Define $\Delta_{\text{adv}} = \text{Var}_q(Z_1) - \text{Var}_q(Z_2)$:

$$\Delta_{\text{adv}} = \left[ \int A(x,y)^2 \frac{p(y|x)^2}{q(y|x)} \, dy - \left( \int A(x,y)p(y|x) \, dy \right)^2 \right]$$
$$- \frac{1}{\|q\|_2^2} \left[ \int A(x,y)^2 p(y|x)^2 q(y|x) \, dy - \left( \int A(x,y)p(y|x)q(y|x) \, dy \right)^2 \right]. \quad (48)$$

Rearranging terms:

$$\Delta_{\text{adv}} = \underbrace{\int A(x,y)^2 \frac{p(y|x)^2}{q(y|x)} \, dy}_{J_1} - \underbrace{\frac{1}{\|q\|_2^2} \int A(x,y)^2 p(y|x)^2 q(y|x) \, dy}_{J_2}$$
$$- \underbrace{\left( \int A(x,y)p(y|x) \, dy \right)^2}_{B_1} + \underbrace{\frac{1}{\|q\|_2^2} \left( \int A(x,y)p(y|x)q(y|x) \, dy \right)^2}_{B_2}. \quad (49)$$

For brevity, we write:

$$\Delta_{\text{adv}} = J_1 - J_2 - B_1 + B_2. \quad (50)$$

**Step 3: Lower Bound for $J_1$** Under the restriction that $A(x,y)^2 \geq A_{\min}^2 > 0$, we have:

$$J_1 = \int A(x,y)^2 \frac{p(y|x)^2}{q(y|x)} \, dy \geq A_{\min}^2 \int \frac{p(y|x)^2}{q(y|x)} \, dy = A_{\min}^2 \cdot I_1, \quad (51)$$

where $I_1 = \int \frac{p(y|x)^2}{q(y|x)} \, dy$ as defined in Appendix A.

From the relationship between KL and $\chi^2$ divergences:

$$D_{\text{KL}}(p\|q) \leq \log(I_1) \quad \Longrightarrow \quad I_1 \geq \exp(D_{\text{KL}}(p\|q)), \quad (52)$$

and therefore:

$$J_1 \geq A_{\min}^2 \cdot \exp(D_{\text{KL}}(p\|q)). \quad (53)$$

**Step 4: Bounding the Remaining Terms** For $J_2$, since $|A(x,y)| \leq A_{\max}$ (advantages are bounded in practice; for GRPO, $A_{\max} \leq 1$):

$$J_2 = \frac{1}{\|q\|_2^2} \int A(x,y)^2 p(y|x)^2 q(y|x) \, dy \leq \frac{A_{\max}^2}{\|q\|_2^2} \int p(y|x)^2 q(y|x) \, dy \leq \frac{A_{\max}^2}{\|q\|_2^2}, \quad (54)$$

where the second inequality follows from Lemma 1 on page 15 of our paper:

$$\sum_{i=1}^{n} p_i^2 q_i \leq 1.$$

For the bias terms $B_1$ and $B_2$, since $\mathbb{E}_p[A] = 0$, we have:

$$|B_1| = \left| \int A(x,y)p(y|x) \, dy \right|^2 = 0, \quad (55)$$

$$|B_2| = \left| \frac{1}{\|q\|_2^2} \left( \int A(x,y)p(y|x)q(y|x) \, dy \right)^2 \right| \geq 0. \quad (56)$$

**Step 5: Combining Bounds**   Combining all terms:

$$\Delta_{\text{adv}} = J_1 - J_2 - B_1 + B_2 \geq A_{\min}^2 \cdot \exp(D_{\text{KL}}(p\|q)) - \frac{A_{\max}^2}{\|q\|_2^2} - 0 + 0. \tag{57}$$

Note that $B_2$ appears with a positive sign in $\Delta_{\text{adv}}$, which improves the lower bound. Since $B_2$ is non-negative and bounded, we can define a constant $C_{\text{adv}}$ that absorbs all bounded subtractions:

$$C_{\text{adv}} = \frac{A_{\max}^2}{\|q\|_2^2}. \tag{58}$$

In practice, as noted in Appendix A, $\|q\|_2^2 = \mathbb{E}_q[q] \approx 0.432$ for language model distributions and $A_{\max} = \max|A| \leq 1$, making $C_{\text{adv}}$ a reasonable constant. Therefore:

$$\Delta_{\text{adv}} \geq A_{\min}^2 \cdot \exp(D_{\text{KL}}(p\|q)) - C_{\text{adv}}. \tag{59}$$

This completes the proof. The result shows that GEPO's advantage-weighted importance sampling maintains the exponential variance reduction property of Theorem 1.

## B   Supplementary experiments: Part I

### B.1   Implementation Details

All experiments use the Qwen3-1.7B/8B model with a maximum input length of 768 and output length of 2048/4096 tokens under both think and no-think mode, limited by computational constraints and low token efficiency (Shrivastava et al., 2025) (reward/length) at full context length[1]. Training follows a GRPO-like algorithm with a learning rate of $1 \times 10^{-6}$, 3% linear warmup, per-device batch size 8, and gradient accumulation of 8, with gradient checkpointing enabled for memory efficiency. To model network latency in heterogeneous environments, we introduce a log-normal delay simulator bounded between 60 and 1800 seconds (99.5% CI), with default delay at 60 seconds and policy staleness varied across 0–64 effective steps. For online training, KL divergence is not used, in order to better evaluate the training stability of the algorithms. For heterogeneous settings, CPPO-KL (Zhang et al., 2024) loss with coefficient 0.005 is applied. Another reason for using CPPO-KL is memory efficiency, as it does not require a separate reference model. Rollouts are generated using vLLM with 8 parallel responses per prompt, and each run lasts 3 epochs, with metrics logged via Weights & Biases. Due to significant discrepancies often arising between the log-probabilities computed by vLLM and those computed during the forward pass under FSDP—which can lead to training instability or divergence [2], we only use vLLM for sampling. An additional forward pass is then performed to recompute the log-probabilities.

The system prompt used in the experiments is shown in Figure 10.

> [think mode] You are a helpful AI Assistant, designed to provided well-reasoned and detailed responses. You FIRST think about the reasoning process as an internal monologue and then provide the user with the answer. Please put your final answer within \boxed{}. Also, indicate that it is the answer.
>
> [no-think mode] You are a helpful AI Assistant, designed to provided well-reasoned and detailed responses. Please put your final answer within \boxed{}. Also, indicate that it is the answer.

Figure 10: System prompt of all trainings in our experiments.

---

[1]This 2k/4k-token limit balances cost and efficiency. Longer outputs increase memory and training time, making high-latency experiments impractical. Crucially, as shown in Figure 11(c), "think mode" yields a high "Overlength Ratio"—most long outputs are truncated and wasted. **Recent studies (Shrivastava et al., 2025; Dai et al., 2025) consistently show that reasoning does not require excessively long chains of thought; redundant thinking merely wastes resources.** Thus, 2048/4096 tokens ensure fair, stable, and manageable experiments.

[2]This blog post describes the issue of training instability (or training crashes) caused by vLLM. https://fengyao.notion.site/off-policy-rl

## B.2    Baselines

The baseline methods compared in our experiments are as follows:

**On-Policy Baselines:**

- Group Relative Policy Optimization (GRPO) (Shao et al., 2024) is a reinforcement learning algorithm that enhances mathematical reasoning in LLMs by estimating advantages through group-relative reward normalization—comparing responses within a group to the same query—thereby eliminating the need for a separate value network and reducing memory overhead compared to PPO.

- Dr. GRPO (Liu et al., 2025) is a debiased variant of GRPO that removes the per-response length normalization and per-question reward standard deviation normalization, thereby eliminating optimization biases that artificially inflate response length and improving token efficiency while preserving reasoning performance.

- Beta Normalization Policy Optimization (BNPO) (Xiao et al., 2025) is a novel reinforcement learning algorithm that dynamically normalizes binary rewards using an adaptively parameterized Beta distribution to reduce gradient variance and enhance training stability for large language models.

- Group Sequence Policy Optimization (GSPO) (Zheng et al., 2025) is a novel reinforcement learning algorithm for large language models that defines importance ratios based on sequence likelihood and performs sequence-level clipping and optimization, thereby achieving superior training stability, efficiency, and performance compared to token-level methods like GRPO.

## B.3    Supplementary Results of Main Experiments

As shown in Table 3, GEPO outperforms all baselines—including GRPO and GSPO—in both best and final performance across zero-delay and high-delay settings, demonstrating superior effectiveness and training stability. A critical observation from the results is that, although GSPO's technical report claims improved stability, we only observe GSPO to be more stable than GRPO under the Online RL setting.

Table 3: Performance of GEPO and baseline methods under Online RL and Hetero RL scenarios (2k limiation).

| Method | AMC2023 | | AIME2024 | | AIME2025 | | MATH500 | | Average | |
|---|---|---|---|---|---|---|---|---|---|---|
| | Best | Last | Best | Last | Best | Last | Best | Last | Best | Last |
| Qwen3-1.7B | 25.6 | - | 1.6 | - | 3.9 | - | 54.7 | - | 21.5 | - |
| **Max Tolerable Delay 0 (Online RL)** | | | | | | | | | | |
| BNPO | 54.3 | 0.0 | 18.4 | 0.0 | 19.1 | 0.0 | 78.7 | 0.0 | 42.6 | 0.0 |
| Dr.GRPO | 53.4 | 14.3 | 19.1 | 1.6 | 18.8 | 2.0 | 78.6 | 35.9 | 42.5 | 13.5 |
| GRPO | 56.3 | 23.4 | 20.7 | 0.4 | 19.9 | 2.3 | 79.8 | 49.7 | 44.2 | 19.0 |
| GSPO | 54.1 | 27.8 | **23.8** | 3.1 | **20.7** | 4.3 | 79.9 | 62.1 | 44.6 | 24.3 |
| GEPO (ours) | **56.9** | **56.9** | 21.9 | **16.4** | 20.3 | **14.1** | **80.4** | **78.1** | **44.9** | **41.4** |
| **Max Tolerable Delay 64 (Hetero RL)** | | | | | | | | | | |
| BNPO | 45.0 | 43.1 | 12.1 | 11.3 | 12.5 | 10.1 | 71.1 | 69.3 | 35.2 | 33.5 |
| Dr.GRPO | 48.4 | 48.4 | 17.2 | 17.2 | 14.8 | 14.8 | 73.9 | 73.9 | 38.6 | 38.6 |
| GRPO | 46.6 | 46.6 | 19.1 | 14.5 | 14.8 | 14.8 | 74.9 | 74.9 | 38.9 | 37.7 |
| GSPO | **54.4** | 23.8 | 17.6 | 1.6 | 17.6 | 2.7 | 78.2 | 55.6 | 42.0 | 20.9 |
| GEPO (ours) | 53.8 | **53.8** | **21.9** | **21.9** | **18.8** | **18.8** | **79.6** | **79.6** | **43.5** | **43.5** |

## B.4    Comparison of Think and Non-think Mode

In *non-think mode*, all methods struggle with exploration, shown by consistently lower `last` scores vs. `best` — indicating late-stage instability.

Notably:

- **GEPO (ours)** dominates across all benchmarks and delays in both `best` and `last`, proving superior efficiency and stability — even without thinking steps.

- At **zero delay**, GEPO beats top baseline (GSPO) by **+3.7** (`best`) and **+20.8** (`last`), resisting early collapse.

- With **delay=64**, baselines improve (e.g., Dr.GRPO: 0.0 → 33.4), showing slack aids stability. GEPO still leads (`last` = 38.0 vs. 35.7).

- Vanilla **Qwen3-1.7B** (`best`=33.2) is outperformed by all RL methods — confirming RL's value even without reasoning.

In short: while non-think mode limits reasoning, **GEPO delivers unmatched stability and final performance**, ideal for latency-sensitive or real-time settings.

Table 4: Performance comparison using Qwen3-1.7B (non-think, 2k).

| Method | AMC2023 | | AIME2024 | | AIME2025 | | MATH500 | | Average | |
|---|---|---|---|---|---|---|---|---|---|---|
| | Best | Last | Best | Last | Best | Last | Best | Last | Best | Last |
| Qwen3-1.7B | 42.8 | - | 10.2 | - | 9.4 | - | 70.2 | - | 33.2 | - |
| **Max Tolerable Delay 0** | | | | | | | | | | |
| BNPO | 43.7 | 0.0 | 13.7 | 0.0 | 13.3 | 0.0 | 74.2 | 0.4 | 36.2 | 0.1 |
| Dr.GRPO | 45.0 | 0.0 | 14.4 | 0.0 | 11.3 | 0.0 | 73.6 | 0.0 | 36.1 | 0.0 |
| GRPO | 50.0 | 28.8 | 16.4 | 7.8 | 13.7 | 7.2 | 77.5 | 59.7 | 39.4 | 25.9 |
| GSPO | 53.1 | 23.8 | 14.4 | 2.7 | 16.0 | 0.4 | 76.5 | 55.6 | 40.0 | 20.6 |
| GEPO (ours) | **55.0** | **52.5** | **22.3** | **22.3** | **18.4** | **13.7** | **79.2** | **77.1** | **43.7** | **41.4** |
| **Max Tolerable Delay 64** | | | | | | | | | | |
| BNPO | 42.1 | 28.6 | 11.1 | 4.7 | 10.2 | 6.4 | 68.3 | 36.1 | 32.9 | 19.0 |
| Dr.GRPO | 45.0 | 41.6 | 14.1 | 9.7 | 13.3 | 10.1 | 72.5 | 72.3 | 36.2 | 33.4 |
| GRPO | 46.3 | 46.3 | 14.5 | **14.5** | 13.6 | 10.1 | 72.6 | 71.7 | 36.8 | 35.7 |
| GSPO | 47.2 | 36.6 | 13.7 | 4.3 | 13.3 | 8.9 | 75.1 | 67.9 | 37.3 | 29.4 |
| GEPO (ours) | **52.5** | **52.5** | **14.5** | 10.2 | **14.5** | **12.1** | **77.4** | **77.1** | **39.7** | **38.0** |

A comparison of RL under `think` vs. `non-think` modes reveals key trade-offs in reasoning, stability, cost, and performance.

**1) Performance and Stability (Delay = 0)** GEPO in `non-think` achieves `Best` = 43.7 and `Last` = 41.4 — near-perfect retention. In `think` mode, it reaches `Best` = 44.9 (+1.2) but identical `Last` = 41.4 — no final gain. Baselines like GRPO collapse (44.2 → 19.0), showing thinking destabilizes training without proper control.

**2) With Delay = 64** In `non-think`, methods rebound sharply: Dr.GRPO (0.0 → 33.4), GRPO (25.9 → 35.7) — minimal slack prevents collapse. In `think`, only GEPO retains peak perfectly (`Last` = `Best` = 43.5); others degrade (e.g., GSPO: 41.9 → 20.9). ⇒ Thinking introduces instability unless explicitly regularized.

**3) Exploration vs. Efficiency (Fig. 11)** `Think` produces longer rollouts — deeper reasoning — but with higher overlength ratios, risking wasted compute and divergence. `non-think` yields shorter, efficient trajectories with lower overlength — yet matches or exceeds performance when paired with stable optimizers like GEPO.

Overall, `think` offers marginal peak gains at the cost of instability and overhead. `Non-think` + GEPO is **more stable, efficient, and often equally effective** — ideal for real-time or latency-sensitive deployment. Choose based on delay tolerance, interpretability needs, and required training stability. Figure 11 compares rollout lengths in GEPO under `think` vs. `non-think` modes, with three subplots:

**1) Average Length**: The red curve (`think`) stays consistently above blue (`non-think`), showing longer rollouts — likely due to added reasoning or policy deliberation.

**2) Terminated Length**: `think` again produces significantly longer *successful* rollouts, suggesting it enables deeper, valid exploration.

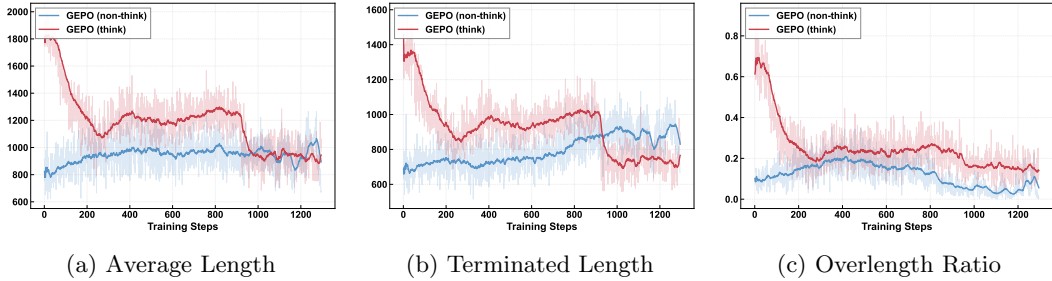

(a) Average Length       (b) Terminated Length       (c) Overlength Ratio

Figure 11: Comparison of rollout (cliped by 2k tokens) lengths under think and non-think modes.

**3) Overlength Ratio**: `think` incurs a much higher overlength rate, especially early in training — indicating that while it boosts exploration, it also risks inefficient, overly long sequences.

**Takeaway**: `think` extends useful exploration and improves completion quality, but at the cost of higher computational overhead. In practice, this trade-off can be managed via length limits or dynamic rollout control to preserve efficiency without sacrificing performance.

### B.5 Details of Hyperparameter Analysis

### B.5.1 Heterogeneous RL Sensitivity Analysis

The default parameters for our main experiments are: group size 8, KL regularization coefficient ($\beta_{KL}$) of 0.005, and lognormal delay distribution.

Table 5: Sensitivity to group size under Hetero RL setting (max delay 64 steps).

| Group Size | AMC2023 | | AIME2024 | | AIME2025 | | MATH500 | | Average | |
|:---:|:---:|:---:|:---:|:---:|:---:|:---:|:---:|:---:|:---:|:---:|
| | Best | Last | Best | Last | Best | Last | Best | Last | Best | Last |
| **2** | 51.3 | 48.4 | 17.2 | 15.6 | 17.2 | 16.4 | 77.8 | 75.4 | 40.9 | 39.0 |
| **4** | **56.8** | **56.8** | 20.3 | 15.6 | **21.1** | **21.1** | 77.7 | 77.7 | **44.0** | 42.8 |
| **8** | 53.8 | 53.8 | **21.9** | **21.9** | 18.8 | 18.8 | **79.6** | **79.6** | 43.5 | **43.5** |

**Group size analysis**. Table 5 shows the performance impact of varying group sizes. We observe that a group size of 4 achieves the highest best performance (44.0), while a group size of 8 yields the best final performance (43.5), demonstrating superior training stability. This validates that larger group sizes provide better statistical regularization for the hetero-RL process.

Table 6: Sensitivity to KL regularization coefficient under Hetero RL setting (max delay 64 steps).

| $\beta_{KL}$ | AMC2023 | | AIME2024 | | AIME2025 | | MATH500 | | Average | |
|:---:|:---:|:---:|:---:|:---:|:---:|:---:|:---:|:---:|:---:|:---:|
| | Best | Last | Best | Last | Best | Last | Best | Last | Best | Last |
| **0.001** | 48.8 | 33.4 | 14.1 | 2.0 | 17.6 | 5.5 | 75.3 | 59.5 | 38.9 | 25.1 |
| **0.002** | 52.5 | 40.3 | 18.0 | 11.3 | **21.1** | 15.2 | 77.6 | 71.4 | 42.3 | 34.6 |
| **0.004** | **55.0** | 53.8 | 16.8 | 15.2 | 19.9 | 16.4 | 79.0 | 79.0 | 42.7 | 41.1 |
| **0.005** | 53.8 | **53.8** | 21.9 | **21.9** | 18.8 | 18.8 | **79.6** | **79.6** | **43.5** | **43.5** |
| **0.006** | 53.4 | 52.5 | 20.7 | 18.0 | 19.1 | **19.1** | 78.5 | 77.1 | 43.0 | 41.7 |
| **0.008** | 47.5 | 49.7 | 12.5 | 21.5 | 14.5 | 17.6 | 74.4 | 77.4 | 37.2 | 41.5 |
| **0.010** | 54.1 | 49.7 | 17.6 | 17.2 | 17.2 | 15.6 | 76.9 | 76.3 | 41.4 | 39.7 |

**KL regularization analysis**. Table 6 demonstrates that the KL divergence coefficient is critical for stability in heterogeneous RL settings. With $\beta_{KL} = 0.005$, GEPO maintains

perfect stability (no performance degradation from best to last). When $\beta_{KL}$ is too small (0.001), the algorithm collapses dramatically (AIME2024 drops from 14.1 to 2.0). When $\beta_{KL}$ is too large (0.010), the model becomes overly conservative, limiting its peak performance.

Table 7: Sensitivity to network latency distribution (95% CI / 60-1920s) under Hetero RL setting (max delay 64 steps).

| Distribution | AMC2023 | | AIME2024 | | AIME2025 | | MATH500 | | Average | |
|---|---|---|---|---|---|---|---|---|---|---|
| | Best | Last | Best | Last | Best | Last | Best | Last | Best | Last |
| **Weibull** | 51.6 | 49.1 | 18.4 | 14.8 | 18.0 | 17.6 | 77.4 | 77.4 | 41.3 | 39.7 |
| **Lognormal** | **53.8** | **53.8** | **21.9** | **21.9** | **18.8** | **18.8** | **79.6** | **79.6** | **43.5** | **43.5** |
| **Exponential** | 53.4 | 47.8 | 19.9 | 17.2 | 16.8 | 15.2 | 79.5 | 79.4 | 42.4 | 39.9 |

**Latency distribution analysis**. Table 7 compares different network latency simulation distributions. The lognormal distribution yields the best performance and stability. The Weibull and exponential distributions show slightly degraded performance, particularly on AIME2024 and AMC2023 benchmarks.

### B.5.2 ONLINE RL SENSITIVITY ANALYSIS

The default parameters for our online RL experiments are: top-$p$=0.95, top-$k$=20, and temperature=0.6.

Table 8: Sensitivity to top-$p$ sampling under Online RL setting (topK=20, temp=0.6).

| top-$p$ | AMC2023 | | AIME2024 | | AIME2025 | | MATH500 | | Average | |
|---|---|---|---|---|---|---|---|---|---|---|
| | Best | Last | Best | Last | Best | Last | Best | Last | Best | Last |
| **0.95** | 56.9 | 56.9 | 21.9 | 16.4 | 20.3 | 14.1 | 80.4 | 78.1 | 44.9 | 41.4 |
| **0.98** | 55.9 | 55.9 | **23.8** | **23.8** | 19.5 | 17.6 | 80.3 | 80.3 | 44.9 | 44.4 |
| **1.00** | **59.7** | **59.7** | 22.3 | 22.3 | **20.7** | **17.6** | **82.0** | **82.0** | **46.2** | **45.4** |

**Top-$p$ sampling analysis**. Table 8 reveals that top-$p$=1.00 achieves the highest best performance (46.2) and final performance (45.4). It suggests that total probability provides a better balance between exploration and stability.

Table 9: Sensitivity to temperature under Online RL setting (topP=0.95, topK=20).

| Temp | AMC2023 | | AIME2024 | | AIME2025 | | MATH500 | | Average | |
|---|---|---|---|---|---|---|---|---|---|---|
| | Best | Last | Best | Last | Best | Last | Best | Last | Best | Last |
| **0.4** | 55.9 | 53.4 | **24.6** | **21.9** | 20.3 | **19.5** | **80.9** | **80.3** | **45.4** | **43.8** |
| **0.6** | **56.9** | **56.9** | 21.9 | 16.4 | 20.3 | 14.1 | 80.4 | 78.1 | 44.9 | 41.4 |
| **0.8** | 53.4 | 52.5 | 22.3 | 7.8 | **21.1** | 15.6 | 80.6 | 73.5 | 44.4 | 37.4 |

**Temperature analysis**. Table 9 shows that a lower temperature (0.4) sharpens the sampling distribution, favoring high-confidence samples and reducing noise. This yields better final performance (43.8 vs. 41.4), demonstrating improved convergence stability. A higher temperature (0.8) increases exploration but compromises stability, particularly on AIME2024 (drops from 22.3 to 7.8).

Table 10: Sensitivity to top-$k$ sampling under Online RL setting (topP=0.95, temp=0.6).

| top-$k$ | AMC2023 | | AIME2024 | | AIME2025 | | MATH500 | | Average | |
|---|---|---|---|---|---|---|---|---|---|---|
| | Best | Last | Best | Last | Best | Last | Best | Last | Best | Last |
| **10** | **58.1** | **58.1** | **22.3** | 18.4 | **21.1** | 15.6 | **80.9** | 78.9 | **45.6** | 42.8 |
| **20** | 56.9 | 56.9 | 21.9 | 16.4 | 20.3 | 14.1 | 80.4 | 78.1 | 44.9 | 41.4 |
| **50** | 55.6 | 55.6 | 20.3 | **18.4** | 18.4 | **18.4** | 80.6 | 80.6 | 43.7 | **43.2** |

**Top-$k$ sampling analysis**. Table 10 demonstrates that GEPO is relatively insensitive to the top-$k$ parameter. While top-$k$=20 achieves the highest best performance (44.9), top-$k$=50 yields the best final performance (43.2), suggesting that a larger candidate set enhances training stability without significantly compromising peak performance.

In summary, our comprehensive hyperparameter analysis reveals that:

1. Group size is crucial for stability in heterogeneous RL settings, with size 8 providing optimal stability.

2. KL regularization coefficient ($\beta_{KL}$=0.005) is critical for maintaining stability under policy divergence.

3. The Weibull distribution is more challenging compared to other distributions.

4. Lower temperature (0.4) improves stability in online RL by reducing sampling noise.

5. Top-$p$ and top-$k$ have a moderate impact on performance, with top-$p$=1.0 and top-$k$=10, providing a good balance.

These findings validate our design choices in the main experiments and provide practical guidance for deploying GEPO in diverse distributed training environments.

## C  SUPPLEMENTARY EXPERIMENTS: PART II

### C.1  COMPARISON WITH ASYNCHRONOUS RL BASELINES

In this section, we compare GEPO with asynchronous RL baselines, including Truncated IS (IMPALA) (Espeholt et al., 2018), TOPR (Roux et al., 2025) and CISPO (Chen et al., 2025). The mathematical distinctions among these three methods are shown in Table 11:

Table 11: Objective of Asynchronous RL Baselines, $sg$ denotes the stop gradient operation.

| Method | Objective |
|---|---|
| Truncated IS | $\mathcal{J}^{\text{TIS}}(\pi_\theta) = \mathbb{E}_{\tau \sim \pi_{\text{old}}} \left[ \text{sg} \left( \frac{\pi_\theta(\tau)}{\pi_{\text{old}}(\tau)} \right)_0^1 R(\tau) \log \pi_\theta(\tau) \right]$ |
| CISPO | $\mathcal{J}^{\text{CISPO}}(\pi_\theta) = \mathbb{E}_{\tau \sim \pi_{\text{old}}} \left[ \text{sg} \left( \frac{\pi_\theta(\tau)}{\pi_{\text{old}}(\tau)} \right)_{1-\epsilon_{\text{low}}^{\text{IS}}}^{1+\epsilon_{\text{high}}^{\text{IS}}} R(\tau) \log \pi_\theta(\tau) M(\tau) \right]$ |
| TOPR | $\mathcal{J}^{\text{TOPR}}(\pi_\theta) = \mathbb{E}_{\tau \sim \pi_{\text{old}}} \left[ \left( \mathbf{1}_{\{\tau \in T^+\}} + \mathbf{1}_{\{\tau \in T^-\}} \text{sg} \left( \frac{\pi_\theta(\tau)}{\pi_{\text{old}}(\tau)} \right)_0^1 \right) R(\tau) \log \pi_\theta(\tau) \right]$ |

**Off-Policy Baselines:**

- **Truncated IS** IMPALA (V-trace) (Espeholt et al., 2018) relies on temporal-difference learning with step-level rewards, whereas LLM reasoning tasks typically employ policy-gradient-based optimization with sparse, episode-level rewards—causing V-trace to degenerate and rendering IMPALA an unsuitable direct baseline for comparison. However, Truncated IS is the key technique of IMPALA, we direcrly compare this technique with GEPO.

- **TOPR** (Roux et al., 2025) is a stable and efficient off-policy reinforcement learning algorithm for fine-tuning large language models that uses asymmetric importance sampling—applying no truncation to positive examples to accelerate learning and lower-truncating negative examples at zero to ensure stability—without requiring KL regularization.

- **CISPO** (Chen et al., 2025) is a stable and efficient off-policy reinforcement learning algorithm for large language models that clips the importance sampling weights—rather than token-level policy updates—to preserve gradient contributions from all tokens while reducing training variance and avoiding premature entropy collapse.

We adopt the experimental settings of CISPO and TOPR as described in their original papers, both of which claim that an explicit KL penalty is unnecessary for achieving stable training. However, in our empirical evaluation, we find that both methods exhibit significant fragility under heterogeneous and high-latency conditions. To ensure a fair and robust comparison, we therefore introduce augmented variants—CISPO+KL and TOPR+KL—as additional baselines, where a KL regularization term is incorporated.

Table 12: Performance of GEPO and asynchronous baselines under Hetero RL scenarios (2k limiation).

| Method | AMC2023 | | AIME2024 | | AIME2025 | | MATH500 | | Average | |
|---|---|---|---|---|---|---|---|---|---|---|
| | Best | Last | Best | Last | Best | Last | Best | Last | Best | Last |
| Qwen3-1.7B | 25.6 | - | 1.6 | - | 3.9 | - | 54.7 | - | 21.5 | - |
| **Max Tolerable Delay 64 (Hetero RL)** | | | | | | | | | | |
| Truncated IS + KL | 47.5 | 0.0 | 15.2 | 0.0 | 15.6 | 0.0 | 74.0 | 0.0 | 38.1 | 0.0 |
| TOPR w/o KL | 51.3 | 0.0 | 17.2 | 0.0 | 15.6 | 0.0 | 76.4 | 0.0 | 40.1 | 0.0 |
| TOPR + KL | 51.9 | 30.0 | 17.2 | 1.2 | 17.2 | 7.0 | 76.9 | 55.1 | 40.8 | 23.3 |
| CISPO w/o KL | 44.1 | 20.0 | 10.2 | 0.8 | 12.1 | 0.8 | 70.7 | 43.4 | 34.3 | 16.3 |
| CISPO + KL | 44.1 | 21.2 | 9.8 | 2.3 | 12.5 | 2.0 | 71.8 | 41.9 | 34.5 | 16.9 |
| GEPO (ours) | **53.8** | **53.8** | **21.9** | **21.9** | **18.8** | **18.8** | **79.6** | **79.6** | **43.5** | **43.5** |

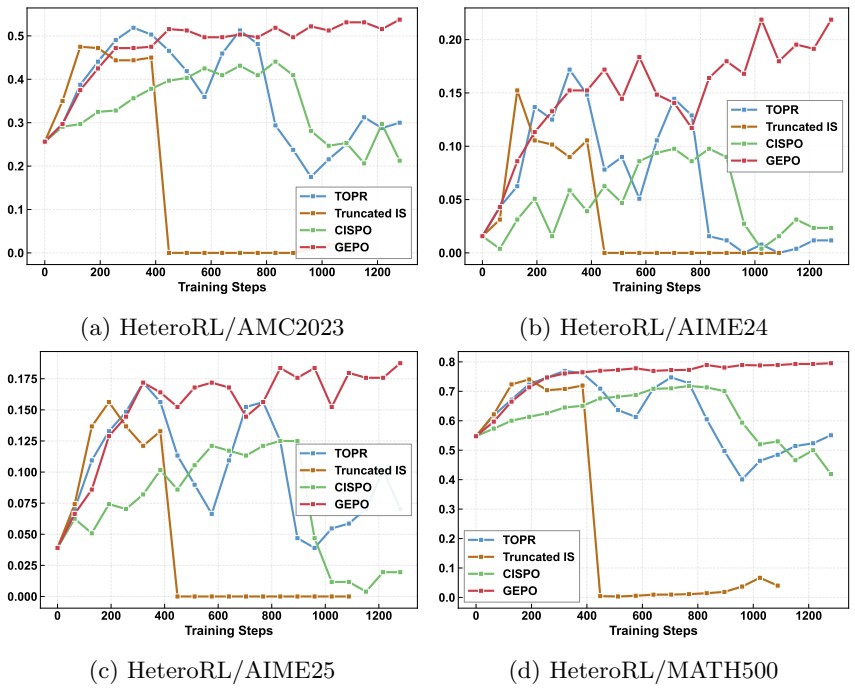

(a) HeteroRL/AMC2023     (b) HeteroRL/AIME24

(c) HeteroRL/AIME25     (d) HeteroRL/MATH500

Figure 12: Eval accuracy curves of GEPO and asynchronous baselines ($\beta_{KL} = 0.005$).

Table 12 presents the best and final performance of GEPO alongside various asynchronous baselines, while Figure 12 shows their evaluation curves across the four benchmarks. The

asynchronous baselines exhibit severe instability—or even collapse—under high-latency conditions, indicating that merely clipping the importance weights is insufficient to ensure training stability.

## D FROM TOKEN-LEVEL TO GROUP-LEVEL IMPORTANCE WEIGHT

**Token-Level Importance Weight**   In traditional policy optimization methods, such as GRPO (Shao et al., 2024) or PPO (Schulman et al., 2017), importance sampling is typically performed at the `token-level`. Specifically, for a generated sequence $y = (y_1, \ldots, y_T)$, the importance weight is computed token by token:

$$w_{token} = \texttt{clip}\left(\frac{p(y_t \mid x, y_{<t})}{q(y_t \mid x, y_{<t})}, 1 - \epsilon, 1 + \epsilon\right), \tag{60}$$

and used to compute the policy gradient at each time step. However, this per-token reweighting scheme suffers from two key limitations:

**1) Inconsistency between optimization and reward granularity:** Concurrent work such as GSPO (Zheng et al., 2025) argues that since the reward is assigned at the sequence level—i.e., to the full response $y$—the importance weighting for policy updates should also operate at the same granularity, rather than at the token level.

**2) High variance in token-level probabilities:** Beyond this alignment issue, we provide an explanation from the perspective of importance weight variance: because the final reward depends on the entire sample, large local changes in token probabilities can lead to extreme values in the ratio $\frac{p(y_t|x,y_{<t})}{q(y_t|x,y_{<t})}$. This inflates the variance of token-level importance weights, causing them to frequently fall outside the clipping range. Once clipped, gradients for these tokens are effectively zeroed out due to the stop-gradient behavior of `torch.clamp`, preventing meaningful updates. As a result, tokens that require large corrections may be ignored, leading to inefficient optimization.

**Sample-Level Importance Weight**   The `sample-level` importance weighting treats the entire response as a sampling unit and computing the weight based on its full conditional probability. Given a prompt $x$, both the target policy $p(y|x)$ and the behavior policy $q(y|x)$ are defined as the joint probability of the sequence:

$$p(y|x) = \left|\prod_{t=1}^{T} p(y_t \mid x, y_{<t})\right|^{\frac{1}{T}}, \quad q(y|x) = \left|\prod_{t=1}^{T} q(y_t \mid x, y_{<t})\right|^{\frac{1}{T}}. \tag{61}$$

The sample-level importance weight is then:

$$w_{\text{sequence}}(y|x) = \texttt{clip}\left(\frac{p(y|x)}{q(y|x)}, 1 - \epsilon, 1 + \epsilon\right). \tag{62}$$

This formulation computes the full sequence-level ratio before clipping, thereby avoiding premature truncation caused by high-variance individual tokens. By preserving gradient flow across the entire sequence, sample-level weighting enables more stable and effective policy updates, especially under high policy divergence induced by network latency.

**Group-Level Importance Weight**   `Group-level` importance weighting represents a paradigm shift in reinforcement learning optimization by recognizing that policy updates should not only align with the granularity of reward assignment but also leverage higher-order statistical relationships among multiple samples. The key insight is that individual samples should not be treated in isolation; instead, their collective behavior under the same prompt provides crucial information for stable policy updates. By considering the expected value of proposal probabilities within a group of responses, we effectively smooth out the erratic fluctuations that plague token-level and even sequence-level weighting schemes. This approach acknowledges an important reality of distributed training: in heterogeneous environments with network latency, policy divergence is inevitable, and our optimization methods must be designed to gracefully handle—not merely tolerate—this divergence. The group-level perspective transforms what was previously seen as a limitation (policy staleness due to latency) into an opportunity for more robust learning through statistical regularization of importance weights.

## D.1 Ablation Study

Since the removal of the variance divisor term in the advantage function has already been extensively validated in prior work (Liu et al., 2025), we focus solely on ablating the Group Expectation component. Table 13 compares three importance weights (IW) in Listing 1, namely token-level (IW of GRPO), sequence-level (IW of GSPO), and group-level (IW of GEPO), under a max tolerable delay of 64 steps. Table 13 presents the results, the sequence-level IW proposed by GSPO does not bring significant stability improvement. Although it outperforms token-level weighting in terms of best performance, it suffers from severe performance degradation by the end of training. We also conducted an ablation study on advantage normalization, and the results show that omitting normalization yields slightly better performance.

Table 13: Ablation study of different importance weights (Hetero RL mode).

| Ablation | AMC2023 | | AIME2024 | | AIME2025 | | MATH500 | | Average | |
|---|---|---|---|---|---|---|---|---|---|---|
| | Best | Last | Best | Last | Best | Last | Best | Last | Best | Last |
| group-lv | 53.8 | **53.8** | **21.9** | **21.9** | **18.8** | **18.8** | **79.6** | **79.6** | **43.5** | **43.5** |
| token-lv | 46.1 | 43.9 | 18.7 | 14.2 | 15.3 | 14.3 | 74.3 | 74.9 | 38.6 | 36.8 |
| seq-lv | **55.2** | 24.1 | 17.2 | 1.9 | 18.1 | 2.1 | 77.3 | 56.9 | 42.0 | 21.3 |
| w/o adv norm | 53.8 | **53.8** | **21.9** | **21.9** | **18.8** | **18.8** | **79.6** | **79.6** | **43.5** | **43.5** |
| wt adv norm | **54.7** | 49.7 | 21.5 | 21.5 | 18.0 | 17.6 | 77.8 | 77.4 | 43.0 | 41.5 |

# E  Network Latency

We test the performance limits of GEPO and baseline methods in simulated network delay scenarios and train the model on heterogeneous computing resources connected through a real-world internet connection.

## E.1 Simulated Network Latency Configuration

Our goal is to simulate RL training of large models over internet-connected heterogeneous compute clusters, where network delays are inherently uncertain. To evaluate algorithmic performance and training stability under extreme and variable latency conditions, we employ three widely used delay distributions $P_d$: log-normal, Weibull, and exponential. We set a high maximum delay threshold of 1800 seconds, which is sufficient to cover typical model and data transmission times. In our simulation, model parameter synchronization and rollout data transfer are implemented as follows:

- **Learner saves model:** The learner periodically saves model checkpoints to a shared model file path `Model_Sync_Path` via `torch.save_pretrained()`.

- **Sampler loads model:** The sampler generates data using the current model until a delay $D_M$, drawn from $P_d$, elapses. It then loads the latest model from `Model_Sync_Path`. The sampler remains active throughout—no idling occurs.

- **Sampler sends data:** Generated rollout batches are saved to `Rollout_Sync_Path` with timestamp $T_{sync}$. To simplify implementation, data transmission is assumed instantaneous; its latency is effectively merged into the model sync delay $D_M$, without affecting simulation validity.

- **Learner uses data:** The learner trains on rollout data from `Rollout_Sync_Path` that falls within a recent time window (e.g., no older than 1800 seconds).

## E.2 Real-World Network Scenarios

To evaluate the performance of heterogeneous reinforcement learning algorithms in realistic network environments, we develop a communication toolkit based on ZeroMQ, supporting TCP/IP-based transmission of inference trajectories from samplers to learners and synchronized model parameter updates. As shown in Figure 13, the toolkit enables multi-node communication over wide-area networks (WANs), with the following core communication logic:

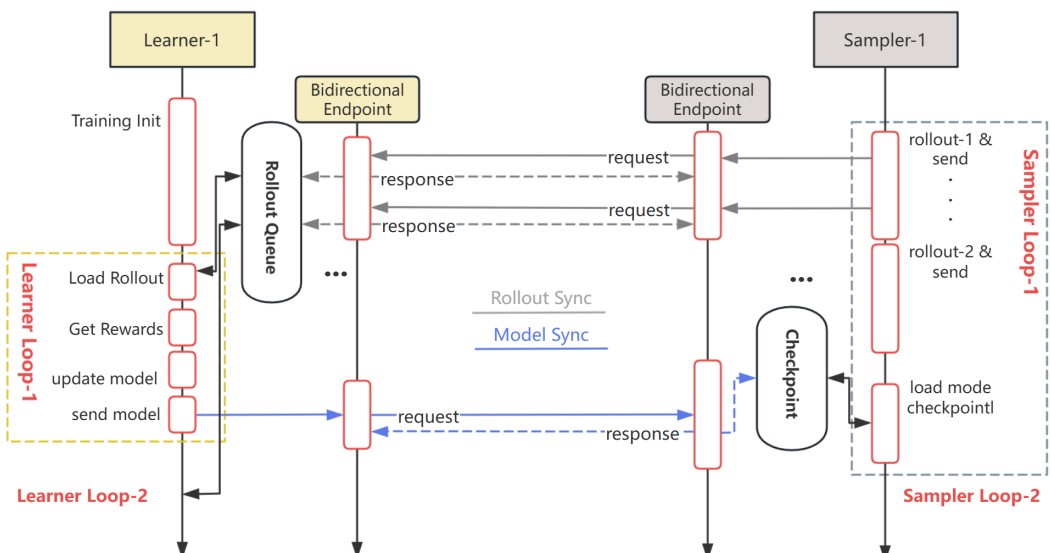

Figure 13: Communication flow between a learner and sampler in a distributed reinforcement learning system, showing trajectory collection and model synchronization over a bidirectional endpoint.

- The learner continuously listens for and buffers inference trajectory messages from samplers. It automatically updates its trajectory buffer upon message arrival and broadcasts the latest model parameters to all connected samplers once a predefined synchronization interval is reached.

- After generating an inference trajectory, the sampler sends it to the learner and continuously listens for parameter update messages. Upon receiving new model parameters, the sampler updates its local parameters and resumes sampling before the next sampling step.

## F  Engineering Optimization: Localized Reward Computation for Reduced Communication

In large-scale heterogeneous reinforcement learning (HeteroRL), where sampler and learner nodes are geographically distributed, network communication overhead becomes a critical bottleneck. A major source of this overhead lies in the *reward aggregation phase*, where traditional implementations require an `all_gather` operation across all processes to collect rewards for group-wise normalization (e.g., computing mean and standard deviation per group of generations). This global synchronization introduces significant latency, especially under high network delay or when using large numbers of distributed nodes. To address this, we introduce a key engineering optimization: **Localized Reward Computation**. Instead of gathering rewards from all processes, we ensure that each group of generations (e.g., $G$ responses per prompt) is entirely generated and scored *within the same process or node*. This design guarantees that all samples belonging to a single group reside locally, enabling the calculation of group statistics (mean, std) without any cross-process communication.

This optimization is particularly synergistic with GEPO's design philosophy. GEPO mitigates training instability under high latency by reducing the variance of importance weights through group-level expectation. Our engineering improvement complements this by *reducing the system's sensitivity to communication latency itself*, creating a virtuous cycle: the algorithm is robust to policy staleness, and the system is optimized to minimize the communication that causes staleness.

The core change is implemented by removing the global `gather` operation in the reward calculation function. This seemingly minor change yields substantial performance benefits. The following table summarizes the key improvements:

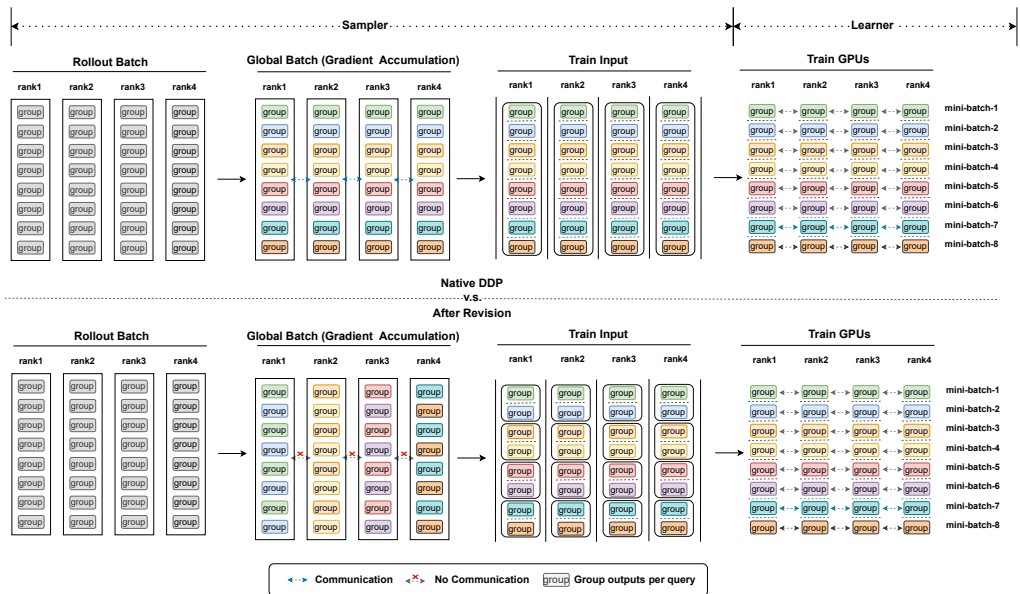

Figure 14: Native Distributed Data Parallel and Revision.

Table 14: Comparison of Communication Overhead: Before vs. Optimized (Localized Reward)

| Component | Before Optimization | After Optimization (Ours) |
|---|---|---|
| Reward Aggregation | Requires `all_gather` for every batch | **No communication required** |
| Group Statistics | Computed globally across all GPUs | Computed **locally** per GPU |
| Communication Frequency | High (per batch) | **None** for reward calculation |
| Latency Sensitivity | High (blocked by slowest node) | **Low** (fully asynchronous) |
| Scalability | Limited by network bandwidth | **Highly scalable** with node count |

By eliminating this frequent and costly synchronization point, our system achieves higher throughput and better resource utilization. This is especially crucial in the HeteroRL setting, where network conditions are unpredictable. The optimization allows sampler nodes to operate more independently, reducing their idle time waiting for global synchronization and making the overall training pipeline more resilient to network fluctuations. This engineering refinement, combined with the GEPO algorithm, provides a holistic solution for stable and efficient training in truly heterogeneous, high-latency environments.

# G  THE COMPARISON OF DIFFERENT REINFORCEMENT LEARNING PARADIGMS

## G.1  CORE CHALLENGE: OFF-POLICY LEARNING AND IMPORTANCE SAMPLING

This formulation highlights the central challenge in asynchronous RL: the mismatch between the behavior policy ($\pi_{\theta_k}$) and the target policy ($\pi_{\theta_{k+\tau}}$), which grows with $\tau$ and introduces bias and variance into the learning process. Addressing this mismatch under high and uncertain latency is the primary focus of our work.

Table 15: Core Characteristics of Online, Offline, and Heterogeneous Reinforcement Learning Paradigms

| Aspect | Online RL | Offline RL | Heterogeneous RL |
|---|---|---|---|
| **Data Generation** | Online interaction: Data generated instantly by the **current policy**. | Fixed, pre-collected dataset: Uses **static data** collected by some (unknown) behavior policy. | Delayed interaction: Data generated by **historical policy versions** (due to **unpredictable network delay**). |
| **Policy Version during Data Generation** | **Always current**: Requires strict synchronization with the learning policy. | **Fixed**: The behavior policy is fixed and inherent to the dataset. | **Dynamically stale**: Staleness determined by **unpredictable network delay** *(core characteristic)*. |
| **System Architecture** | Tightly-coupled: **Actor (environment interaction)** and **Learner (parameter update)** typically co-located or on a low-latency network. | Single-machine or simple distributed: **No real-time interaction** required; training is data-driven. | Geographically decoupled: **Actor and Learner separated** by a **high-latency network**, tolerant to delays. |
| **Handling of System Latency** | Treated as failure: Requires synchronization; latency causes **resource idling** and training stalls. | Not applicable: Training process has **no real-time interaction**. | **Algorithmic compensation**: Employs **corrective techniques** (e.g., importance sampling) to mitigate latency effects. |
| **Core Challenge** | **Exploration-exploitation** trade-off during learning. *Resource utilization* under sequential tasks. | **Distributional shift** and **limited data coverage**. | **Algorithmic stability under dynamic delays**: High/variable latency causes **importance weight variance explosion**. |

The HeteroRL framework decouples rollout sampling from parameter learning, leading to a setting where the learner updates the new policy $\pi_{\theta_{k+\tau}}$ using data generated by an older policy $\pi_{\theta_k}$—a canonical off-policy learning scenario. To correct for the distributional shift between the behavior and target policies, importance sampling (IS) is commonly employed. Under the GRPO framework, this weight is used to scale the advantage during policy gradient updates. However, when the delay $\tau$ is large, causing significant divergence between the policies (i.e., high KL divergence $D_{\mathrm{KL}}(\pi_{\theta_{k+\tau}} \| \pi_{\theta_k})$), we observe in practice that the variance of the importance weights increases rapidly with $\tau$, and the estimation error of the expected reward also grows. As a result, the variance of $w_{\mathrm{std}}$ explodes, introducing high noise into gradient estimates and ultimately leading to training instability or collapse. The central challenge of this work is thus to design a robust algorithm that mitigates the training instability caused by the explosion of importance sampling variance under high network latency.

## H  FUTURE WORK

### H.1  DEFENSIVE SAMPLING AND SMOOTH DENOMINATOR MECHANISM

As a promising direction for future work, we propose to explore a defensive sampling strategy that adaptively blends the target policy probability into the importance weight denominator. This approach aims to mitigate bias introduced by approximating the denominator while improving training stability—particularly in asynchronous or heterogeneous learning settings. By introducing a variance-aware smoothing coefficient, the method could dynamically interpolate between standard importance weighting and the policy gradient objective, thereby reducing sensitivity to high-variance estimates. We hypothesize that such a smooth,

adaptive weighting scheme would lead to more robust and stable policy updates, and plan to investigate its theoretical properties and empirical effectiveness in future studies.

## I   CASE STUDY

### A case of MATH level 3-5: GRPO v/s GSPO v/s GEPO

**User:**

Find the greatest integer less than $(\sqrt{7} + \sqrt{5})^6$. (Do not use a calculator!).

**GRPO**

`<think>` Okay, so I need to find the greatest integer less than $(\sqrt(7) + \sqrt(5))^6$. Hmm, that seems pretty complicated at first glance. Let me think about how to approach this.
Wait, maybe ...(👎 **Omitted numerous ineffective thoughts and reflections**)
Alternatively, maybe I can find a recurrence relation for $S_n$. Let me try to find a recurrence. Let me note that $x$ and $y$ are roots of the equation $t^2 - 2\sqrt{7}\,t + 2 = 0$. Therefore, for any $n \geq 2$, we have
$$x^n = 2\sqrt{7}\,x^{n-1} - 2x^{n-2}.$$
Similarly for $y^n$. Therefore,
$$S_n = x^n + y^n = 2\sqrt{7}(x^{n-1} + y^{n-1}) - 2(x^{n-2} + y^{n-2}) = 2\sqrt{7}\,S_{n-1} - 2S_{n-2}.$$
(👍 **This is the key step to solving this problem**) ...
But this seems to involve $\sqrt(7)$, which complicates things. However, if I consider $S_n$ + something, maybe I can eliminate the $\sqrt(7)$ term? Wait, but maybe I need another approach.
Alternatively, maybe I can compute $S_n$ for $n = 0, 1, 2, ...$ and find a pattern. Let me try that.
First, let's compute $S_0 = x^0 + y^0 = 1 + 1 = 2$.
...(👎 **Omitted numerous ineffective thoughts and reflections**)
Wait, but $S_6$ is $x^6 + y^6$. However, since $x = \sqrt(7) + \sqrt(5)$ and y $= \sqrt(7) - \sqrt(5)$, and since $x > 0$ and $y$ is positive? Wait, $\sqrt(7)$ is about 2.645, $\sqrt(5)$ is about 2.236. So $\sqrt(7) + \sqrt(5)$ is about 4.881, and $\sqrt(7) - \sqrt(5)$ is about 0.409. So $y$ is positive, but small. Therefore, $x^6$ is a large number, and $y^6$ is a small number. Therefore, $S_6 = x^6 + y^6$ is approximately $x^6$, and the integer part of $x^6$ would be $S_6 - 1$, since $y^6$ is positive but less than 1. Therefore, the greatest integer less than $x^6$ is $S_6 - 1$.
But wait, let me check if $y^6$ is less than 1. Since $y = \sqrt(7) - \sqrt(5) \approx 0.409$, then $y^6$ is $(0.409)^6$. Let me compute that: $0.409^2 \approx 0.167$, then $0.167^3 \approx 0.004$
(👍 **Over length here, although close to arriving at the correct answer**)

**GSPO**

`<think>` Okay, so I need to find the greatest integer less than $(\sqrt(7) + \sqrt(5))^6$. Hmm, that seems a bit intimidating at first, but I'll try to break it down step by step. First, I remember that expressions like $(a + b)^n$ often have a pattern when expanded, especially when a and b are irrational numbers. Since the question is about the greatest integer less than this value, I might need to find an expression that's just below the actual value.
Let me think. If I consider $(\sqrt(7) + \sqrt(5))^6$ and $(\sqrt(7) - \sqrt(5))^6$, their sum might be an integer because the irrational parts cancel out. That's a common trick with conjugates. So, the sum of these two would be $(sqrt(7) + sqrt(5))^6 + (sqrt(7) - sqrt(5))^6$, which is an integer. Therefore, the integer part of the original expression would be that integer minus the $(sqrt(7) - sqrt(5))^6$, which is a small positive number.
So, the answer would be boxed as the boxed integer boxedboxedboxedboxedboxed. Wait, no, the boxed answer should be the boxed boxed answer boxed. So, boxed $\boxed{\text{boxed boxed}}$ but boxed in the final answer boxed.
Therefore, the boxed answer is the boxed boxed boxed boxed boxed boxed boxed answer boxed. So boxed boxed boxed boxed boxed boxed boxed boxed boxed boxed boxed boxed...
(👍 **Numerous repetitions occur here.**)

---

**GEPO**

`<think>` Okay, so I need to find the greatest integer less than $(\sqrt{(7)} + \sqrt{(5)})^6$. Hmm, that seems pretty big. I don't have a calculator, so I have to figure this out without calculating the actual value. Let me think.
...

But maybe I can find a recurrence formula. Let's see:
We know that $x$ and $y$ are roots of the quadratic equation $t^2 - (x + y)t + xy = 0$. So $t^2 - 2\sqrt{(7)}t + 2 = 0$. Therefore, the recurrence relation for $S_n$ is $S_n = 2\sqrt{(7)}S_{n-1} - 2*S_{n-2}$.
(👉 **This is the key step to solving this problem**) ...
Therefore, the answer is $\boxed{13535}$.
`</think>`
To find the greatest integer less than $(\sqrt{7} + \sqrt{5})^6$, we use the identity that for any expression of the form $(\sqrt{a} + \sqrt{b})^n + (\sqrt{a} - \sqrt{b})^n$, the result is an integer. This is because the irrational parts cancel out when the two expressions are added together.
Let: - $x = \sqrt{7} + \sqrt{5}$ - $y = \sqrt{7} - \sqrt{5}$
We compute:
$$x^6 + y^6$$
First, calculate $x^2$:
$$x^2 = (\sqrt{7} + \sqrt{5})^2 = 7 + 2\sqrt{35} + 5 = 12 + 2\sqrt{35}$$
Then $x^4 = (x^2)^2 = (12 + 2\sqrt{35})^2 = 144 + 48\sqrt{35} + 4 \cdot 35 = 284 + 48\sqrt{35}$
Next, $x^6 = x^4 \cdot x^2 = (284 + 48\sqrt{35})(12 + 2\sqrt{35})$. Expanding this:
$$284 \cdot 12 + 284 \cdot 2\sqrt{35} + 48\sqrt{35} \cdot 12 + 48\sqrt{35} \cdot 2\sqrt{35}$$
$$= 3408 + 568\sqrt{35} + 576\sqrt{35} + 96 \cdot 35 = 3408 + 1144\sqrt{35} + 3360$$
$$= 6768 + 1144\sqrt{35}$$
Similarly, $y^6 = 6768 - 1144\sqrt{35}$
Adding $x^6$ and $y^6$:
$$x^6 + y^6 = 2 \cdot 6768 = 13536$$
Since $x^6 = 13536 - y^6$, and $y^6$ is a small positive number (as $\sqrt{7} - \sqrt{5} \approx 0.4097$), we have:
$$x^6 \approx 13536 - 0.004 \approx 13535.996$$
Thus, the greatest integer less than $x^6$ is:
$$\boxed{13535}$$

**Comparison:**

GRPO exhibits a large number of ineffective reflections, wasting many valuable tokens; statistics show that GRPO uses words such as "wait" and "however" more than 22 times. Due to reward collapse, GSPO begins generating extensive repetitions after brief reasoning, rendering the response unreadable. GEPO arrives at the correct answer during the thinking phase through sound reasoning and reflection. When formally responding, its thought process is clear and steps are concise.

## J   LARGE LANGUAGE MODEL USAGE

The use of LLMs in this article is limited to text polishing and code generation for plotting.

