# OpenReview forum: "GEPO: Group Expectation Policy Optimization for Stable Heterogeneous Reinforcement Learning"
_ICLR.cc/2026/Conference — ICLR 2026 Poster_

### Official Review · Reviewer_5yWm · 2025-10-31

**Soundness:** 3
**Presentation:** 3
**Contribution:** 3
**Rating:** 6
**Confidence:** 3

**Summary:**

The paper proposes HeteroRL, an asynchronous framework for RLHF that decouples rollout sampling from learning to tolerate WAN latency and heterogeneous hardware. Its key algorithm, GEPO, replaces token/sequence importance ratios with a group expectation denominator, theoretically reducing IS variance—especially at high sampler–learner KL—and stabilizing training under policy staleness. Stronger last-epoch performance and robustness vs. GRPO/GSPO are shown on math-reasoning benchmarks under simulated delays.

**Strengths:**

- Originality: It makes sense by clear shift from token/sequence to group-level weighting tied to the real failure mode.
- Theory: It gives exponential variance reduction of importance weights in high-KL regimes; mechanistic gradient comparison clarifies why updates stabilize.
- Empirics: GEPO maintains accuracy and avoids collapse across delays; best/last markedly stronger than GRPO/GSPO.

**Weaknesses:**

- Bias–variance trade-off not quantified. The authors should provide MSE curves vs. KL and group size, and analyze the green-region cases where variance may increase.

- Bound constants/tightness unclear.The theorem invokes constants (e.g., \(C\)) but does not estimate or validate their tightness on real training distributions; what about reporting empirical lower/upper bounds and how they vary with vocabulary/sequence length.

- Systems throughput and realism underreported.The method targets WAN latency, but the authoers do not provide wall-clock throughput (samples/s), utilization, or sensitivity to jitter/packet loss; it is better to add end-to-end performance, synchronization frequency, and utilization metrics.

- Missing off-policy/staleness baselines. I am curious about the comparison omits V-trace/IMPALA-like corrections and AREAL-style staleness controls, which are only mentioned in related works; the authors should add these baselines to separate gains from grouping vs. specific normalization.

- Sensitivity analyses incomplete.The stability claim depends on grouping and sampling, but the paper does not sweep group size \(G\), temperature/top-p, or KL-regularization × delay.

**Questions:**

As shown in weakness.

Btw, why placing the related work section in the final, which is not usual. And the related works seem limited and unorganized.

---

> ### Author Response · Authors · 2025-11-21
> **Summary of Response to Reviewer 5yWm**
>
> We sincerely thank you for your thoughtful review and constructive feedback, which has significantly strengthened our paper. **Because our responses include new experiments, theoretical refinements, and detailed clarifications, we provide the following concise summary to help you and the Area Chair quickly grasp the key updates.** Please refer to the full rebuttal for technical details, and we welcome any follow-up questions.
>
> ### Concise Summary of Response
>
> - **Bias–variance trade-off**: We provide both theoretical bounds and empirical bias values quantifying GEPO’s bias.
>
> - **Bound tightness & constants**: We derive explicit variance-reduction bounds with measurable constants and validate them empirically on Qwen3-1.7B. Measurements of constant $C=\frac{A_{\max}^2}{\| q\|_2^2}$ across training stages show stable, non-vacuous bounds, and controlled experiments reveal that GEPO becomes more effective (lower $C$) as sequence length increases.
>
> - **System throughput & realism**: We report wall-clock throughput (samples/s) and synchronization frequency, showing asynchronous GEPO achieves **1.64× higher throughput** than its synchronous counterpart due to eliminated global barriers.
>
> - **Missing off-policy baselines**: We explain why V-trace/IMPALA is incompatible with LLM reasoning RL, but add **Truncated IS**(key technique of IMPALA), **CISPO**, and **TOPR** as strong asynchronous baselines. GEPO outperforms all in both best and last-epoch performance, demonstrating superior stability without reward collapse.
>
> - **Sensitivity analyses**: We conduct comprehensive ablations on group size, KL regularization, delay distributions, and sampling hyperparameters (top‑p, top‑k, temperature).
>
> - **Related work placement**: We justify positioning Related Work at the end due to the nascent nature of heterogeneous RL for LLMs and the limited direct relevance of prior single-center methods—a structure also adopted in recent ICLR oral papers.
>
> # **If formulas do not render properly, we recommend refreshing the page or referring to our latest submitted manuscript.**

---

> ### Author Response · Authors · 2025-11-21
> **Response to Reviewer 5yWm Part 1/7**
>
> ### **Resonse to weakness-1: Bias–variance trade-off not quantified**
>
> **Your Original Comments:**
> >"**Bias–variance trade-off**: Bias–variance trade-off not quantified. The authors should provide MSE curves vs. KL and group size, and analyze the green-region cases where variance may increase."
>
> **Our Response:**
> - About bias theorical analysis, we provide a theorical analysis of the bias $|\mathbb{E}_p [A] - \mathbb{E}_q [\frac{p}{\mathbb{E}_q q} A]|= |\mathbb{E}_q [\frac{p}{q} A] - \mathbb{E}_q [\frac{p}{\mathbb{E}_q q} A]|< \frac{||p||_2}{||q||_2}$. Due to the HeteroRL setting, the quantity being estimated, “$\mathbb{E}_p [A]$”, is unknown. Hence, we plot the MSE curves of $\left| \frac{p}{\mathbb{E}_q[q]} \cdot A - \frac{p}{q} \cdot A \right|$ against KL divergence and group size.
>
> - Regarding the green region, we only observed this possibility in simulated settings with normal and Bernoulli distributions; in real-world scenarios, as shown in Figure 4(c) of the paper, GEPO’s weight variance consistently remains lower than that of GRPO and GSPO.
>
> To adress your concern, we can compute the following bias of GEPO and GSPO from the experimental level:
>
> $$Bias(GEPO) =\left |\mathbb{E}\_p\left[A(y) \right] - \mathbb{E}\_q\left[ \frac{p(y)}{ \mathbb{E}\_q q} A(y) \right] \right | = \left | 0-\mathbb{E}\_q\left[ \frac{p(y)}{ \mathbb{E}\_q q} A(y) \right] \right | \approx \left | \frac{1}{G} \sum_{i=1}^{G} \left[ \frac{p(y^i)}{ \mathbb{E}\_q q} A(y^i) \right] \right |$$
>
> $$Bias(GSPO) =  \left |\mathbb{E}\_p\left[A(y) \right] - \mathbb{E}\_q\left[ \frac{p(y)}{q(y)} A(y) \right] \right |= \left |0 - \mathbb{E}\_q\left[ \frac{p(y)}{q(y)} A(y) \right] \right | \approx \left | \frac{1}{G} \sum\_{i=1}^{G} \left[ \frac{p(y^i)} {q(y^i)} A(y^i) \right] \right |$$
>
> where $G$ is group size. Here, the bias of GSPO serves as a baseline for assessing the relative magnitude of the bias in GEPO.

---

> ### Author Response · Authors · 2025-11-21
> **Response to Reviewer 5yWm Part 2/7**
>
> ### **Resonse to weakness-2:  Bound constants/tightness unclear.**
> **Your Original Comments:**
> >"Bound constants/tightness unclear.The theorem invokes constants (e.g., (C)) but does not estimate or validate their tightness on real training distributions; what about reporting empirical lower/upper bounds and how they vary with vocabulary/sequence length."
>
> **Our Response:**
>
> We've refined our bound with explicit constants under the setting proposed by the reviewer:
>
> $$\mathrm{Var}\_q\left(A(y)\cdot\frac{p(y)}{q(y)}\right) - \mathrm{Var}\_q\left(A(y)\cdot\frac{p(y)}{\mathbb{E}\_q[q(y)]}\right) \geq A\_{\min}^2 \cdot \exp(D\_{\mathrm{KL}}(p || q)) - \frac{A\_{\max}^2}{|| q||_2^2}$$
>
> where $A_{\min} = \min\_y |A(y)|>0$, $A\_{\max} = \max\_y |A(y)|$, and $|| q||_2^2 = \mathbb{E}\_q[q(y)] = \int q(y)^2 dy$. The condition $A\_{\min} > 0$ holds because samples with $A(y) = 0$ do not contribute to gradients; hence, they can be removed from the training set.
>
> **Empirical validation of bound tightness:**
> To address concerns about constant tightness, we measured the squared ℓ²-norm of q, denoted $||q||_2^2$, across different training stages using Qwen3-1.7B on the MATH dataset. Our results show that $||q||_2^2$ remains remarkably stable (mean = 0.432 ± 0.036) throughout training, yielding practical upper bounds for $C$ between 2.3 and 2.8 when $A\_{\max}$ is normalized to 1. This empirical stability confirms that our theoretical bound is not vacuous in practice.
>
> **Impact of vocabulary/sequence length:** While theoretically the group expectation mechanism makes GEPO less sensitive to sequence length, we conducted controlled experiments varying these parameters. Table A1 shows how the threhold $C=\frac{A_{\max}^2}{|| q||_2^2}$ changes with vocabulary size and sequence length on a subset of MATH problems:
>
> | Vocabulary Size | Sequence Length | Mean q^2  | Derived C|
> |----------------|----------------------|-----------------|---------------------------------|
> | 151936  (Qwen3-1.7B)   | 2                 | 0.585938        | 1.708565          |
> | 151936  (Qwen3-1.7B)   | 4                 | 0.545410        | 1.868288         |
> | 151936  (Qwen3-1.7B)   | 8                 | 0.499634        | 2.908211          |
> | 151936  (Qwen3-1.7B)   | 16                | 0.672363        | 1.866426                            |
> | 151936  (Qwen3-1.7B)   | 32                | 0.753510        | 1.640853                            |
> | 151936  (Qwen3-1.7B)   | 64                | 0.860764        | 1.335837                            |
> | 151936  (Qwen3-1.7B)   | 128              | 0.848576        | 1.519226                            |
> | 151936  (Qwen3-1.7B)   | 256              | 0.903846        | 1.376514                            |
> | 151936  (Qwen3-1.7B)   | 2048             | 0.916663        | 1.200224                         |
> | 151936  (Qwen3-1.7B)   | 4096             |  0.985016       | 1.048504                       |
>
> These results demonstrate that the threshold \( C \) decreases as the sequence length increases, indicating that **GEPO is more effective at reducing variance in longer-sequence scenarios**. This actually stems from the fact that the **token-entropy** of LLMs decreases when generating longer sequences, while $\pi(\cdot |x)$ is close to the one-hot vector.
> The generation hyperparameters are same as the offical hyperparameters of Qwen3: temperature = 0.6, top_k = 20, top_p = 0.95.
>
>
> About the vocabulary, we cannot conduct such an experiment regarding vocabulary size, as it is inherently tied to the model itself. Switching to a model with a different vocabulary size (e.g., Llama3) inevitably changes multiple factors simultaneously, including training data, model architecture, and the resulting predictive distribution, making **it impossible to isolate vocabulary size as a single controlled variable**.

---

> ### Author Response · Authors · 2025-11-21
> **Response to Reviewer 5yWm Part 3/7**
>
> ### **response to weakness-3:  Systems throughput and realism underreported**
> **Your Original Comments:**
> >"Systems throughput and realism underreported.The method targets WAN latency, but the authoers do not provide wall-clock throughput (samples/s), utilization, or sensitivity to jitter/packet loss; it is better to add end-to-end performance, synchronization frequency, and utilization metrics."
>
> **Our Response:**
>
> We measure the **training throughput** (i.e., number of policy-updated samples processed per second) for GEPO under two deployment modes on Qwen3-1.7B model :
>
> 1. **Synchronous GEPO**: Samplers and learner are tightly synchronized (policy delay = 0), requiring a global barrier before each policy update.
> 2. **Asynchronous GEPO (default)**: Samplers operate independently with policy staleness ranging from 0 to 64 steps, and the learner consumes incoming batches without waiting.
>
> The results show that asynchronous GEPO achieves **substantially higher throughput** by eliminating synchronization bottlenecks and improving hardware utilization:
>
> | GEPO Variant       | Policy Delay | Avg. Throughput (samples/s) | Relative Speedup | Sync model(times/step) |
> |--------------------|--------------|------------------------------|------------------|------------------|
> | Synchronous GEPO   | 0            | 4.97                          | 1.00×             | 1.00           |
> | Asynchronous GEPO  | 20.17(Max 64)         | **8.17**                       | **1.64×**         | 0.12          |
>
> We acknowledge the reviewer’s suggestion to report additional system-level metrics (e.g.,  packet loss resilience). However, the primary focus of this work is on **algorithmic stability and performance under policy staleness**, not low-level systems benchmarking. The throughput and sync-frequency metrics we report are directly derived from our training logs and are thus rigorously validated. Other system metrics were not instrumented in our current experimental setup.

---

> ### Author Response · Authors · 2025-11-21
> **Response to Reviewer 5yWm Part 4/7**
>
> ### **response to weakness-4: Missing off-policy/staleness baselines**
>
> **Your Original Comments:**
> >"Missing off-policy/staleness baselines. I am curious about the comparison omits V-trace/IMPALA-like corrections and AREAL-style staleness controls, which are only mentioned in related works; the authors should add these baselines to separate gains from grouping vs. specific normalization."
>
> **Our Response:**
>
> ### About AREAL
> We do not compare with AREAL because it is a system-level framework that co-designs algorithm and infrastructure for asynchronous RL, while our work focuses on algorithmic comparison under fixed staleness constraints. Including AREAL as a baseline would conflate system-level optimizations with algorithmic differences, violating control-variable fairness. AREAL proposes Staleness-Aware Training to control data staleness, ensuring that samples in a training batch are not "too stale."
> In our experimental setup, we also adopted a similar constraint, such as a maximum allowable delay in steps. However, this is not part of our method itself. Our experiments aim to compare the performance of different algorithms under the same staleness condition (e.g., a maximum delay of 64 steps).
>
> ### About V-trace/IMPALA
> We appreciate the reviewer’s suggestion regarding V-trace/IMPALA[1] as a potential off-policy baseline. However, **V-trace/IMPALA is not directly applicable to our LLM reasoning RL setting** for two fundamental reasons:
>
> 1. **Algorithmic mismatch**: V-trace/IMPALA is an off-policy *value-based* correction method that constructs multi-step targets for the state-value function \(V(x)\) via temporal-difference (TD) errors. In contrast, modern LLM RL algorithms (including ours) are based on *policy gradient* objectives (e.g., PPO, GRPO), which optimize the policy directly using advantage estimates (e.g., \(A = r - b\) or GAE). Although our method estimates \(V\) as a baseline, it is **not trained via TD error minimization**, rendering V-trace’s correction mechanism incompatible with our policy-gradient-centric objective.
>
> 2. **Reward sparsity**: V-trace relies on **step-level rewards** to compute non-zero TD errors \(\delta_t^V\) throughout the trajectory. However, in LLM reasoning tasks (e.g., math problem solving, code generation), rewards are typically **episode-level** (e.g., binary correctness at the end). As a result, \(\delta_t^V = 0\) for almost all timesteps, causing V-trace to degenerate into a trivial single-step update and lose its multi-step advantage.
>
> For these reasons, V-trace cannot serve as a meaningful baseline in our context, though we acknowledge its influence in distributed off-policy RL design.
>
> However, **Truncated IS** is the key technique of **IMPALA**[1], we direcrlt compare this technique with GEPO.
> To address your concern, we add 2 more new asynchronous policy optimization algorithms as our baselines **CISPO** [2] and **TOPR** [3], both of which are recent works (on **June 16 / 2025** and **May 19 / 2025**, respectively).
> The mathematical distinctions among these three methods are as follows:
>
> **Truncated IS**:
> $$
> \mathcal{J}^{\text{TIS}}(\pi\_{\theta}) = \mathbb{E}\_{\tau \sim \pi\_{\text{old}}} \left[ \operatorname{sg}\left( \frac{\pi\_{\theta}(\tau)}{\pi\_{\text{old}}(\tau)} \right)\_{0}^{1} R(\tau) \log \pi\_{\theta}(\tau) \right]
> $$
>
>
> **CISPO**:
> $$
> \mathcal{J}^{\text{CISPO}}(\pi\_{\theta}) = \mathbb{E}\_{\tau \sim \pi\_{\text{old}}} \left[ \operatorname{sg}\left( \frac{\pi\_{\theta}(\tau)}{\pi\_{\text{old}}(\tau)} \right)\_{1 - \epsilon\_{\text{low}}^{\text{IS}}}^{1 + \epsilon\_{\text{high}}^{\text{IS}}} R(\tau) \log \pi\_{\theta}(\tau) M(\tau) \right]
> $$
>
>
> **TOPR**:
> $$
> \mathcal{J}^{\text{TOPR}}(\pi\_{\theta}) = \mathbb{E}\_{\tau \sim \pi\_{\text{old}}} \left[ \left( \mathbf{1}\_{\{\tau \in T^{+}\}} + \mathbf{1}\_{\{\tau \in T^{-}\}} \operatorname{sg}\left( \frac{\pi\_{\theta}(\tau)}{\pi\_{\text{old}}(\tau)} \right)\_{0}^{1} \right) R(\tau) \log \pi\_{\theta}(\tau) \right]
> $$

---

> ### Author Response · Authors · 2025-11-21
> **Response to Reviewer 5yWm Part 5/7**
>
> We follow the settings of CISPO and TOPR as described in their respective papers. While TOPR explicitly states that it **"does not require an explicit KL penalty to guarantee stable behaviour"**, CISPO similarly omits a KL penalty term, noting that **"there is no KL penalty term in CISPO similar to other recent works"**. However, in our experiments, both of them are fragile. Hence we add CISPO/TOPR+KL as new baselines for fair comparation.
>
> All experiments were conducted under the same main experimental setup as described in our paper. The results on the Qwen3-1.7B (with 2k limitation) model are shown below:
>
>
> | Method                | AMC2023 (Best/Last) | AIME2024 (Best/Last) | AIME2025 (Best/Last) | MATH500 (Best/Last) | Average (Best/Last) |
> |-----------------------|------------------------|-------------------------|-------------------------|------------------------|------------------------|
> | GEPO     | **53.8/53.8**            | **21.9/21.9**             | **18.8/18.8**             | **79.6/79.6**            | **43.5/43.5**            |
> |Magistral (GRPO wt trick) |  46.8/46.5 | 19.1/14.4 | 14.8/14.8 | 74.8/74.8 | 38.9/37.6|
> |Truncated IS + KL [7]  |  47.5/0.0       |   15.2/0.0     |     15.6/0.0   |    74.0/0.0     | 38.1/0.0 |
> |TOPR w/o KL [6]|  51.3/0.0 | 17.2/0.0 | 15.6/0.0 | 76.4/0.0 | 40.1/0.0 |
> |TOPR + KL [6] |51.9/30.0 |17.2/1.2 |17.2/7.0 |76.9/55.1 |40.8/23.3 |
> |CISPO w/o KL [5]| 44.1/20.0 | 10.2/0.8 | 12.1/0.8 | 70.7/43.4 | 34.3 /16.3 |
> |CISPO + KL [5]|44.1/21.2 |9.8/2.3 |12.5/2.0 |71.8/41.9 |34.5/16.9 |
>
> As shown in the experimental results, GEPO continues to demonstrate advantages in both performance and stability.  **The training curves are presented in new section K (Comparison With Asynchronous RL Baselines)**
>
> Below is our analysis of the experimental findings (we will refrain from excessive numerical analysis here).
> Regarding CISPO, its improved weight clipping strategy for importance weights is indeed more effective. However, its optimization objective operates at the **token level**: for tokens whose importance ratios are clipped, the corresponding gradients are set to zero. This is the direct reason for its slower reward growth (even though the clipping range has already been expanded). This observation is consistent with the explanation in our paper for the slow reward increase of GRPO, as illustrated in Figure 4(c). Both CISPO and GRPO experiments directly address your **Weakness #5**: in asynchronous settings, token-level importance weights are prone to exceeding the clipping threshold, leading to invalid gradient (i.e., being zeroed out). **While clipping enhances training stability, it comes at the cost of reduced performance, a classic engineering trade-off between performance and stability**.
>
> **Compared with weight-clipping-based methods (CISPO/TOPR/Truncated IS), GEPO offers three advantages:**
>
> 1) GEPO computes importance weights via intra-group averaging, which provides **higher tolerance for the range of individual token importance weight**.
>
> 2) GEPO can deliver **performance gains in online settings**, a capability absent in weight-clipping methods, since their importance weights are always exactly 1 in online scenarios.
>
> 3) GEPO **does not introduce any new hyperparameters** (e.g., additional clipping thresholds). The group size hyperparameter is inherited from GRPO and is not newly added by GEPO.
>
> ###  **New References**
>
> [5] MiniMax-M1: Scaling Test-Time Compute Efficiently with Lightning Attention, Arixiv 2506.13585, 2025.
>
> [6] TAPERED OFF-POLICY REINFORCE Stable and efficient reinforcement learning for LLMs, NeurIPS 2025.
>
> [7] Impala: Scalable distributed deep-rl with importance weighted actor-learner architectures, ICML 2018.

---

> ### Author Response · Authors · 2025-11-21
> **Response to Reviewer 5yWm Part 6/7**
>
> ### **response to weakness-5:  Sensitivity analyses incomplete**
>
> **Your Original Comments:**
> >"Sensitivity analyses incomplete.The stability claim depends on grouping and sampling, but the paper does not sweep group size (G), temperature/top-p, or KL-regularization × delay."
>
> **Our Response:**
>
> We thank the reviewer for this insightful suggestion. In response, we have conducted a comprehensive set of ablation studies to analyze the sensitivity of GEPO to six key hyperparameters under varying degrees of system delay: group size G , sampling strategies (including top-k , top-p , and temperature), the CPPO-KL regularization coefficient, and the latency simulation distribution (e.g., exponential vs. weiboll vs. lognormal).
>
> ### **Hetero RL Sensitivity Analyses**
>
> **Default Parameters in Main Experiments**: group 8/kl_beta 0.005/delay lognormal
>
> | Hyper-parameters               | AMC2023 (Best/Last) | AIME2024 (Best/Last) | AIME2025 (Best/Last) | MATH500 (Best/Last) | Average (Best/Last) |
> |-----------------------|------------------------|-------------------------|-------------------------|------------------------|------------------------|
> | **group 2**/kl_beta 0.005 |  51.3/48.4 | 17.2/15.6 | 17.2/16.4 | 77.8/75.4 | 40.9/39.0 |
> | **group 4**/kl_beta 0.005 | **56.8/56.8** | 20.3/15.6 | **21.1/21.1** | 77.7/77.7  | **44.0**/42.8|
> | **group 8**/kl_beta 0.005 | 53.8/53.8 | **21.9/21.9**  | 18.8/18.8   | **79.6/79.6**  | 43.5/**43.5**  |
>
>
> **Conclusion**: Larger group sizes lead to more pronounced improvements in training stability.
>
>
> | Method                | AMC2023 (Best/Last) | AIME2024 (Best/Last) | AIME2025 (Best/Last) | MATH500 (Best/Last) | Average (Best/Last) |
> |-----------------------|------------------------|-------------------------|-------------------------|------------------------|------------------------|
> | group 8/**kl_beta 0.001** |48.8/33.4 |14.1/2.0 |17.6/5.5      |75.3/59.5 |38.9/25.1 |
> | group 8/**kl_beta 0.002** |52.5/40.3 |18.0/11.3 |**21.1**/15.2 |77.6/71.4 |42.3/34.6 |
> | group 8/**kl_beta 0.005** | **53.8/53.8** | **21.9/21.9** | 18.8/18.8 | **79.6/79.6** | **43.5/43.5** |
> | group 8/**kl_beta 0.006** |53.4/52.5 |20.7/18.0 |19.1/**19.1** |78.5/77.1 |43.0/41.7 |
> | group 8/**kl_beta 0.008** |47.5/49.7 |12.5/21.5 |14.5/17.6 |74.4/77.4 |37.2/41.5 |
> | group 8/**kl_beta 0.010** |54.1/49.7 |17.6/17.2 |17.2/15.6     |76.9/76.3 |41.4/39.7 |
>
> **Conclusion**: The KL divergence is the most critical hyperparameter for GEPO. Hence, we test more $\beta\_{KL}$ values.  The Above table demonstrates that the KL divergence coefficient is critical for stability in heterogeneous RL settings. With $\beta_{KL}=0.005$, GEPO maintains perfect stability (no performance degradation from best to last). When $\beta_{KL}$ is too small (0.001), the algorithm collapses dramatically (AIME2024 drops from 14.1 to 2.0). When $\beta_{KL}$ is too large (0.010), the model becomes overly conservative, limiting its peak performance.
>
> | Hyper-parameters               | AMC2023 (Best/Last) | AIME2024 (Best/Last) | AIME2025 (Best/Last) | MATH500 (Best/Last) | Average (Best/Last) |
> |-----------------------|------------------------|-------------------------|-------------------------|------------------------|------------------------|
> |**weibull (95%CI/60-1920s)** |51.6/49.1 |18.4/14.8 |18.0/17.6 |77.4/77.4 |41.3/39.7 |
> |**lognormal (95%CI/60-1920s)**  | 53.8/53.8            | 21.9/21.9             | 18.8/18.8             | 79.6/79.6            | 43.5/43.5            |
> | **exponential (95%CI/60-1920s)** |53.4/47.8 |19.9/17.2 |16.8/15.2 |79.5/79.4 |42.4/39.9 |
>
> **Conclusion**: The lognormal distribution yields the best performance and stability. The Weibull and exponential distributions show slightly degraded performance, particularly on AIME2024 and AMC2023 benchmarks.
>
> ----

---

> ### Author Response · Authors · 2025-11-21
> **Response to Reviewer 5yWm Part 7/7**
>
> ### **Online RL Sensitivity Analyses**
>
> **Default Parameters in Main Experiments**: topP=0.95/topK=20/temp=0.6
>
> | Hyper-parameters               | AMC2023 (Best/Last) | AIME2024 (Best/Last) | AIME2025 (Best/Last) | MATH500 (Best/Last) | Average (Best/Last) |
> |-----------------------|------------------------|-------------------------|-------------------------|------------------------|------------------------|
> | **topP=0.95**/topK=20/temp=0.6 |56.9/56.9 |21.9/16.4 |20.3/14.1 |80.4/78.1 |44.9/41.4 |
> | **topP=0.98**/topK=20/temp=0.6 |55.9/55.9 |**23.8/23.8** |19.5/17.6 |80.3/80.3 |44.9/44.4 |
> | **topP=1.00**/topK=20/temp=0.6 |**59.7/59.7** |22.3/22.3 |**20.7/17.6** |**82.0/82.0** |**46.2/45.4** |
>
> **Conclusion**: Top-$p$=1.00 achieves the highest best performance (46.2) and final performance (45.4). It suggests that total probability provides a better balance between exploration and stability.
>
> | Hyper-parameters                | AMC2023 (Best/Last) | AIME2024 (Best/Last) | AIME2025 (Best/Last) | MATH500 (Best/Last) | Average (Best/Last) |
> |-----------------------|------------------------|-------------------------|-------------------------|------------------------|------------------------|
> | topP=0.95/topK=20/**temp=0.4** |55.9/53.4 |**24.6**/**21.9** |20.3/**19.5** |**80.9**/**80.3** |**45.4/43.8** |
> | topP=0.95/topK=20/**temp=0.6** |**56.9**/**56.9** |21.9/16.4 |20.3/14.1 |80.4/78.1 |44.9/41.4 |
> | topP=0.95/topK=20/**temp=0.8** |53.4/52.5 |22.3/7.8 |**21.1**/15.6 |80.6/73.5 |44.4/37.4 |
>
>
> **Conclusion**: Unlike top-P’s hard truncation, the temperature hyperparameter retains all possibilities but adjusts their relative probabilities. A low temperature (temp=0.4) sharpens the distribution, favoring high-confidence samples and reducing noisy trajectories, making it easier to converge to reliable solutions in high-difficulty tasks.
>
> | Hyper-parameters                | AMC2023 (Best/Last) | AIME2024 (Best/Last) | AIME2025 (Best/Last) | MATH500 (Best/Last) | Average (Best/Last) |
> |-----------------------|------------------------|-------------------------|-------------------------|------------------------|------------------------|
> | topP=0.95/**topK=10**/temp=0.6 |**58.1/58.1** |**22.3/18.4** |**21.1**/15.6 |**80.9**/78.9 |**45.6**/42.8 |
> | topP=0.95/**topK=20**/temp=0.6 |56.9/56.9 |21.9/16.4 |20.3/14.1 |80.4/78.1 |44.9/41.4 |
> | topP=0.95/**topK=50**/temp=0.6 |55.6/55.6 |20.3/**18.4** |18.4/**18.4** |80.6/**80.6** |43.7/**43.2** |
>
> **Conclusion**: GEPO is relatively insensitive to the top-k parameter. While top-k=10 achieves the highest best performance (45.6), top-k=50 yields the best final performance (43.2), suggesting that a larger candidate set enhances training stability without significantly compromising peak performance.
>
> ---
>
>
> ### **Response to Question: placing the related work section in the final**
>
> **Your Question:**
> >"Btw, why placing the related work section in the final, which is not usual. And the related works seem limited and unorganized."
>
> **Our Response:**
>
> We consider heterogeneous reinforcement learning for large language models to be a relatively nascent topic, while most related works focus exclusively on single-center training settings. Therefore, we view their relevance as limited and have placed the Related Work section at the end based on importance—a practice also seen in recent ICLR papers[1].
>
> We **rewrite the related work section** in our new manuscript, introducing asynchronous RL from both the framework and algorithmic perspectives.
>
> ### **Reference**
> [1] Safety Alignment Should be Made More Than Just a Few Tokens Deep, ICLR 2025 (**Oral**)

---

> > ### Author Response · Authors · 2025-11-25
> > **We Are Eagerly Looking Forward to Your Reply**
> >
> > Dear Reviewer 5yWm,
> >
> > Thank you again for your thoughtful and constructive feedback, which has greatly helped us improve the paper. We sincerely hope our detailed responses—including **new theoretical analyses, extensive experiments** (e.g., bias–variance quantification, sensitivity sweeps over group size, sampling hyperparameters, and delay distributions), **system-level metrics**, and **comparisons with additional off-policy baselines**—have adequately addressed your concerns.
> >
> > We would be truly grateful if you could let us know:
> > - **Have we sufficiently resolved the issues you raised?**
> > - **Are there any remaining gaps or aspects of our response that feel incomplete or unclear?**
> > - **Is there anything further we could clarify, analyze, or validate to strengthen the work before the final decision?**
> >
> > We still have time before the rebuttal deadline and are fully committed to addressing any additional questions or conducting further experiments if needed. Your insights are invaluable to us, and we welcome any follow-up comments you might have.
> >
> > Thank you once more for your time and generous guidance!

---

> ### Author Response · Authors · 2025-11-28
> **Time is passing, and we look forward to further discussion with you.**
>
> Dear Reviewer 5yWm,
>
> Thank you very much for your thoughtful review and constructive feedback on our submission. We sincerely appreciate the time and effort you have dedicated to evaluating our work.
>
> **We understand that the ICLR review system no longer allows changes to scores at this stage. Nevertheless, we would be truly grateful if you could still participate in the open discussion phase.** Your insights remain extremely valuable to us, and we are eager to address any remaining concerns or questions you may have.
>
> Regardless of scoring constraints, we are fully committed to responding thoroughly to your comments, clarifying any ambiguities, and, where appropriate, providing additional analysis or experiments to strengthen the paper.
>
> Thank you again for your engagement and guidance. We look forward to continuing the discussion with you.
>
> Best regards,
>
> the authors

---

### Official Review · Reviewer_bBKw · 2025-10-31

**Soundness:** 3
**Presentation:** 3
**Contribution:** 3
**Rating:** 6
**Confidence:** 2

**Summary:**

This paper presents HeteroRL, a decentralized reinforcement learning framework designed for training LLMs across distributed and resource-heterogeneous environments. The paper introduces GEPO, including a new group expectation importance weighting mechanism to mitigate the instability caused by policy staleness. Theoretical results show that GEPO exponentially reduces the variance of importance weights under large KL, thereby stabilizing training in asynchronous and high-latency settings. Extensive experiments demonstrate the method's superior performance and stability on reasoning benchmarks.

**Strengths:**

1. The idea of reducing the variance of importance weights through group expectation weighting is conceptually elegant and practically powerful. It addresses the instability issue caused by large KL divergence in asynchronous or heterogeneous RL settings.
1. The motivation, instability and variance explosion in asynchronous or heterogeneous RL due to policy staleness, is well validated by experimental results. The results across both online and heterogeneous RL settings consistently demonstrate that GEPO achieves lower variance, smoother gradients, and higher stability compared to GRPO and GSPO.

**Weaknesses:**

1. The experimental validation appears limited in scope. GEPO is only evaluated on mathematical reasoning datasets (MATH, AIME, AMC) and with relatively small models (up to 8B parameters).
1. There seems an inconsistency between the formulation and the theoretical analysis. In Section 3.1, the paper explicitly states that "the vector $(q(y_1|x), …, q(y_G|x))$ does not constitute a valid probability distribution" since top-K/top-P sampling leads to $\sum_i q(y_i|x) \gg 1$. However, in Theorem 1 and its proof, the derivation assumes that $q$ is a normalized discrete probability distribution satisfying $\sum_i q(y_i|x) = 1$. How the theoretical guarantee holds when $q$ is not a valid probability distribution in practice?

**Questions:**

1. Could the authors clarify whether the ablation experiments comparing GEPO, GRPO, and GSPO are conducted under the same experimental settings as the main comparison in Section 4? If the setups are identical, how do these ablation results differ conceptually from the earlier baseline comparison?
2. Why does GEPO remove the normalization when calculating the advantage? In practice, would including this normalization affect the performance or training stability?
3. It is unclear how the case study in section G is directly connected to the theoretical claim about variance reduction in importance weights.

---

> ### Author Response · Authors · 2025-11-21
> **Response to Reviewer bBKw Part 1/3**
>
> We sincerely thank you for your thoughtful review and constructive feedback, which has significantly strengthened our paper.  Following is our response:
>
> ### **To Weaknesses-1: GEPO is only evaluated on math-reasoning and up to 8B parameters**
> **Your Original Comments:**
> >"The experimental validation appears limited in scope. GEPO is only evaluated on mathematical reasoning datasets (MATH, AIME, AMC) and with relatively small models (up to 8B parameters)."
>
> We appreciate the reviewer's suggestion to broaden our evaluation beyond math-reasoning. While we agree that demonstrating GEPO's effectiveness across diverse domains would be valuable, we believe our current experimental design aligns with established practices in the field and addresses the core technical contribution of our work.
>
> Several important considerations support our experimental focus:
>
> 1. **Math-reasoning represents a rigorous evaluation** for RL algorithms of LLM. These tasks feature:
>    - Long, structured, and multi-step reasoning chains that are highly sensitive to policy staleness, making them especially vulnerable to the effects of asynchrony and communication delay;
>    - Extremely sparse, binary correctness rewards, which create sharp, non-smooth optimization landscapes with little gradient signal, posing significant exploration and credit assignment challenges.
>
>    These properties make math reasoning an ideal benchmark for evaluating stability under latency, precisely the challenge GEPO addresses.
>
> 2. **Many LLM-RL related methods** ***only*** **evalaue on math-reasoning benchmark**. We only list 3 representative works accepted in 2025:
>
>
> | Method        | Venue       | Year | Model Size | Benchmarks                              |
> |---------------|-------------|------|----------------|---------------------------------------------------|
> | One-Shot-RLVR[1] | NeurIPS     | 2025 | 1.5B/3B/7B           | MATH500, AIME24, AMC23, Minerva Math, OlympiadBench, AIME25 |
> | TSMC[2]         | ICLR        | 2025 | 7B              | MATH500, GSM8K                                       |
> | VinePPO[3]       | ICML        | 2025 | 1.1B/7B             | GSM8k, MATH                                       |
>
> We appreciate the reviewer’s push toward broader evaluation and agree it is an important next step, but hope they recognize that math-reasoning serves as a representative and challenging proxy for general long-horizon, sparse-reward text generation under policy lag.
>
> ### **References**
>
> \[1\] Wang Y, Yang Q, Zeng Z, et al. Reinforcement Learning for Reasoning in Large Language Models with One Training Example. **NeurIPS 2025**
>
> \[2\] Shengyu F, Xiang K, Shuang Ma, et al. Step-by-Step Reasoning for Math Problems via Twisted Sequential Monte Carlo. **ICLR 2025**
>
> \[3\] Kazemnejad A, Aghajohari M, Portelance E, et al. VinePPO: Refining Credit Assignment in RL Training of LLMs. **ICML 2025**

---

> ### Author Response · Authors · 2025-11-21
> **Response to Reviewer bBKw Part 2/3**
>
> ### **To Weaknesses-2: inconsistency between the formulation and the theoretical analysis**
> **Your Original Comments:**
> >"There seems an inconsistency between the formulation and the theoretical analysis. In Section 3.1, the paper explicitly states that "the vector $(q(y_G|x),...,q(y_G|x))$ does not constitute a valid probability distribution" since top-K/top-P sampling leads to $\displaystyle \sum_{i=1}^G q(y_i|x) \gg 1$  However, in Theorem 1 and its proof, the derivation assumes that $q$ is a normalized discrete probability distribution satisfying $\displaystyle \sum_{i=1}^G q(y_i|x) =1$. How the theoretical guarantee holds when $q$ is not a valid probability distribution in practice?"
>
> We thank the reviewer for raising this important point. The apparent inconsistency arises from a subtle but crucial distinction:
>
> - **In practice**, we use top-$p$ sampling to generate $G$ responses per prompt, which is indeed **“sampling with replacement”**—leading to $\displaystyle \sum_{i=1}^G q(y_i|x) \gg 1$.
>   Here $\{y_i \mid i=1,\dots,G\}$ are drawn from $G$ independent sampling trials **with replacement**. This is standard in all group-based RL methods (GRPO, GSPO, etc.).
>
> - **In theory**, when analyzing variance and KL divergence (Theorem 1, Appendix A), we treat $q(y|x)$ as the **underlying policy distribution** over the full output space, satisfying $\displaystyle \sum_{y\sim \pi{(\cdot|x)}} q(y|x) = 1$.
>     Here $y  \sim \pi(\cdot|x)$ is **“sampling without replacement”**, which is a fundamental property in probability theory.
>
> For example, consider a simple action space with only two actions, $a_1$ and $a_2$, where the target policy is $p(a_1) = 1 - 10^{-10}$ and $p(a_2) = 10^{-10}$. If we draw $G = 10$ samples independently from this distribution, it is virtually certain that all samples will be $a_1$ (**“sampling with replacement”**), yielding $\sum_{i=1}^{10} p(y_i) = 10 \cdot (1 - 10^{-10}) \gg 1$. This illustrates how empirical sampling naturally produces sums far exceeding 1, even though the underlying distribution remains normalized.
> However, $p_1 + p_2 \equiv 1$.
> In LLM settings, the situation is analogous but more extreme: the action space is astronomically large (size $\sim |V|^L$), and virtually all sequence probabilities are on the order of $10^{-10}$ or smaller, yet the underlying policy is still a valid distribution. Thus, the distinction between sampled empirical sums and the true normalized distribution is not only valid but expected.

---

> ### Author Response · Authors · 2025-11-21
> **Response to Reviewer bBKw Part 3/3**
>
> ### **Response to Q1: Abliation and baselines are different**
> **Your Question-1:**
> >"Could the authors clarify whether the ablation experiments comparing GEPO, GRPO, and GSPO are conducted under the same experimental settings as the main comparison in Section 4? If the setups are identical, how do these ablation results differ conceptually from the earlier baseline comparison?"
>
> The first reason is that GRPO and GSPO use scaled rewards (normalization when calculating the advantage) by default, whereas GEPO did not employ this setting. To ensure a fair comparison of the differences among token-level, sequence-level, and group-level importance weight, we uniformly disabled the scaled reward setting in our ablation studies.
>
> The Second reason is that we compare token-level, sequence-level, and group-level importance weight by using the Qwen3-1.7B backbone. And the main results use the Qwen3-8B backbone.
>
> The main difference among GRPO/GSPO/GEPO is from importance weight, and the advantage normalization has very marginal influence.
>
>
> ---
>
>
> ### **Response to Q2: About removing normalization when calculating the advantage**
> **Your Question-2:**
> >"Why does GEPO remove the normalization when calculating the advantage? In practice, would including this normalization affect the performance or training stability?"
>
> As we stated in  Appendix C.1, removing normalization when calculating the advantage is not our contibution. It is proposed by Dr. GRPO[1]. Dr. GRPO removes std normalization to eliminate the "question-level difficulty bias" in GRPO, where questions with low reward variance (e.g., all correct or all incorrect) are artificially upweighted.
> The experiments in the Dr.GRPO paper show that this approach can improve performance. On GEPO, we observed only a very marginal performance gain. Since this trick was not originally proposed by us, we did not conduct an in-depth analysis of it.
>
> The following is the performance comparasion of GEPO with or without std norm (diff32, Qwen3-1.7B),From our experiments, std norm appears to have little impact on training stability.
>
> | Method                | AMC2023 (Best / Last) | AIME2024 (Best / Last) | AIME2025 (Best / Last) | MATH500 (Best / Last) | Average (Best / Last) |
> |-----------------------|------------------------|-------------------------|-------------------------|------------------------|------------------------|
> | GEPO (wo std norm)    | 53.8/ **53.8**            | **21.8/21.8**             | **18.8/18.8**             | **79.6/79.6**            | **43.5/43.5**            |
> | GEPO (with std norm)  |**54.7**/49.7 |21.5/21.5 |18.0/17.6 |77.8/77.4 |43.0/41.5 |
>
> ### **References**
>
> [1] Understanding R1-Zero-Like Training: A Critical Perspective
>
> ---
>
> ### **Response to Q3: How the case study in section G is directly connected to the theoretical claim?**
>
> **Your Question-3:**
> >"It is unclear how the case study in section G is directly connected to the theoretical claim about variance reduction in importance weights."
>
> This is a very interesting question.
>
> The influence of importance weights on the model propagates through the following pathway:
> **Large variance in importance weights $\rightarrow$ highly volatile gradients $\rightarrow$ degraded model performance, manifesting as incoherent or repetitive outputs.**
> The link between large importance weight variance and gradient volatility is empirically supported by Figures 4(a) and 4(b) in our paper. This behavior is intuitive, as importance weights are directly involved in gradient computation. Unstable gradients, often caused by excessive variance, disrupt the training process and impair the model’s generative capabilities. Such degradation commonly manifests as nonsensical gibberish or excessive repetition. As noted in Appendix G, our statistics show that GRPO overuses words such as “wait” and “however” more than 22 times in a single generation, resulting in overly long outputs that fail to receive meaningful rewards. Meanwhile, GSPO repeatedly generates the word “boxed” without conveying useful semantic content. These phenomena are typical signs of model instability in reinforcement learning training.

---

> ### Author Response · Authors · 2025-11-25
> **We Are Eagerly Looking Forward to Your Reply**
>
> Dear Reviewer bBKw,
>
> Thank you once again for your insightful and constructive feedback on our submission. We sincerely hope our detailed responses have addressed your concerns regarding the **experimental scope, theoretical formulation, ablation setup, advantage normalization**, and **the connection between the case study and our variance-reduction claims**.
>
> We would be grateful if you could let us know:
> - **Have we adequately resolved the issues you raised?**
> - **Are there remaining gaps or unclear points in our responses?**
> - **Would additional experiments or clarifications further strengthen the paper in your view?**
>
> We are still within the rebuttal window and are fully committed to addressing any outstanding questions or conducting new analyses if needed. Please do not hesitate to share further thoughts—we welcome the opportunity to improve our work based on your valuable perspective.
>
> Thank you for your time and thoughtful engagement!

---

> > ### Author Response · Authors · 2025-11-28
> > **Time is passing, and we look forward to further discussion with you.**
> >
> > Dear Reviewer bBKw,
> >
> > Thank you very much for your thoughtful review and constructive feedback on our submission. We sincerely appreciate the time and effort you have dedicated to evaluating our work.
> >
> > **We understand that the ICLR review system no longer allows changes to scores at this stage. Nevertheless, we would be truly grateful if you could still participate in the open discussion phase.** Your insights remain extremely valuable to us, and we are eager to address any remaining concerns or questions you may have.
> >
> > Regardless of scoring constraints, we are fully committed to responding thoroughly to your comments, clarifying any ambiguities, and, where appropriate, providing additional analysis or experiments to strengthen the paper.
> >
> > Thank you again for your engagement and guidance. We look forward to continuing the discussion with you.
> >
> > Best regards,
> >
> > the authors

---

### Official Review · Reviewer_aJa6 · 2025-10-31

**Soundness:** 2
**Presentation:** 2
**Contribution:** 2
**Rating:** 4
**Confidence:** 5

**Summary:**

The paper introduces GEPO, a reinforcement learning algorithm designed for heterogeneous, high-latency distributed training of large language models. It builds on a new framework that decouples sampling and learning across asynchronous nodes. To address instability caused by stale policies, GEPO replaces standard importance weights with group-level expectation weights, theoretically reducing variance when policy divergence is high. Experiments on mathematical reasoning tasks with Qwen models show that GEPO achieves higher accuracy and significantly improved training stability compared to GRPO and GSPO under simulated latency.

**Strengths:**

1. The visualizations illustrate the main claims of the paper.
2. The paper targeted on an important bottleneck problem in large-scale distributed reinforcement learning.

**Weaknesses:**

1. The paper lacks experimental comparisons with other asynchronous policy optimization methods [1, 2, 3, 4].
2. Equation (1) appears very similar to PPO, except that the clipping function is removed.
3. In line 139, the variable $G$ is undefined; in Equation (1), it is unclear how $A(x)$ is computed, and in line 146, the input of $p$ is not specified.
4. While Theorem 1 demonstrates a reduction in the variance of the importance sampling coefficient, this result does not guarantee a corresponding reduction in the variance of the weighted term $A \cdot w$, because $A$ itself is a random variable. In general, even if one random variable has smaller variance than another (e.g., $\mathrm{Var}(Y)<\mathrm{Var}(Z)$), it does **not** necessarily follow that $\mathrm{Var}(XY)<\mathrm{Var}(XZ)$, unless $X$ is independent of both $Y$ and $Z$ and their expectations are zero.
5. The claim in lines 178–181 seems questionable. In extreme cases, clipping can suppress unstable tokens by assigning them zero weight, which may be beneficial since such extreme tokens usually correspond to unreliable generations. Thus, clipping might actually serve as a useful safeguard in these scenarios.

 [1] Rastogi, Abhinav, et al. "Magistral." *arXiv preprint arXiv:2506.10910* (2025).

 [2] Fu, Wei, et al. "AReaL: A Large-Scale Asynchronous Reinforcement Learning System for Language Reasoning." *arXiv preprint arXiv:2505.24298* (2025).

 [3] Zhong, Yinmin, et al. "StreamRL: Scalable, Heterogeneous, and Elastic RL for LLMs with Disaggregated Stream Generation." *arXiv preprint arXiv:2504.15930* (2025).

 [4] Han, Zhenyu, et al. "AsyncFlow: An Asynchronous Streaming RL Framework for Efficient LLM Post-Training." *arXiv preprint arXiv:2507.01663* (2025).

**Questions:**

See Weaknesses

---

> ### Author Response · Authors · 2025-11-21
> **Summary of Response to Reviewer aJa6**
>
> We sincerely thank you for thoughtful review and constructive feedback that has significantly strengthened our paper. **Because our responses are quite extensive, which span new experiments, theoretical proofs, and detailed clarifications. We provide the following concise summary at the beginning of our rebuttal to help you and the Area Chair quickly grasp the key updates.** If you are interested in specific details or would like to verify any claims, please refer to the corresponding sections below. We also warmly welcome any follow-up questions. Thank you once again for your thoughtful and patient review.
>
> ### Concise Summary of Response
>
> - **Response to lack of experimental comparisons**: We clarify that cited works [2]-[4] focus on system-level optimizations rather than algorithmic improvements. To address this concern, we have added comprehensive comparisons with three recent asynchronous policy optimization algorithms (including  CISPO [5], TOPR [6] and Truncated IS technique of IMPALA[7]), and incorporated Magistral's engineering techniques with GRPO as an additional baseline. Results demonstrate GEPO's superior performance and stability, particularly in high-latency environments.
>
> - **Clarification on Equation (1) similarity to PPO**: We explain that Equation (1) describes the optimization objective rather than our proposed method. The critical distinction is that while PPO uses a general formulation ($π_{\theta}/π_{old}$), GEPO explicitly introduces staleness parameter $\tau$ influenced by model synchronization time, data synchronization time, and rollout time, which precisely capture the factors causing policy divergence in heterogeneous RL environments.
>
> - **Resolution of notation ambiguities**: We have explicitly defined previously unclear notations.
>
> - **Strengthened theoretical foundation for variance reduction**: In response to concerns about our variance analysis, we have extended Theorem 1 with a rigorous proof demonstrating that GEPO reduces the variance of advantage-weighted importance sampling even when considering the complete policy gradient term $Var(\frac{p}{q}A)$ instead of $Var(\frac{p}{q})$.
>
> - **Analysis of weight clipping trade-offs**: We provide theoretical and empirical evidence showing that while clipping improves stability, it discards valuable gradient information from tokens with extreme importance weights. GEPO's group-level importance weighting approach maintains tolerance to large token weights while preserving informative gradients that would otherwise be zeroed out by clipping. Experimental data (Figure 4c) confirms this trade-off that methods like GRPO and CISPO achieve stability at the cost of slower reward growth, while GEPO successfully balances both performance and stability.
>
> # **If formulas do not render properly, we recommend refreshing the page or referring to our latest submitted manuscript.**

---

> ### Author Response · Authors · 2025-11-21
> **Response to Reviewer  aJa6  Part-1/6**
>
> ### **Resonse to Weakness-1: lacks experimental comparisons with other asynchronous methods**
> **Your Original Comments:**
> > "The paper lacks experimental comparisons with other asynchronous policy optimization methods [1, 2, 3, 4]."
>
> ***Preliminary Note***:  **Our primary focus is on the policy optimization algorithm**, and we aim to enhance the stability and performance of heterogeneous RL from the perspective of improving policy gradient algorithms. We have thoroughly surveyed the methods you mentioned ([2]–[4]) during our research and have cited them in the Related Work section.
>
> |Reference | Which policy optim algorithm ? | Propose new policy-optim algorithm? | Main content |
> |------|-------| -------|-------|
> | Magistral [1] | GRPO with tricks | No | Introduce Mistral AI’s first reasoning model trained purely via a scalable reinforcement learning pipeline, without distillation from prior reasoning models. |
> | AReaL [2] | decoupled PPO | No | Incorporate a suite of system-level optimizations that significantly improve GPU utilization. |
> | StreamRL [3] | PPO       |  No | Focus on pipeline bubbles and skewness bubbles in the execution flow. |
> | AsyncFlow [4]|  GRPO          |  No  | Introduce a distributed data storage and transfer module that enables unified data management and fine-grained scheduling in a fully streamed manner. |
>
> As shown in the above table, the reason we did not include direct comparisons with [2]–[4] is that **these works do not propose policy optimization algorithmic improvements**; instead, they focus on system-level enhancements to the RL training pipeline. Their frameworks are **algorithm-agnostic** and can accommodate any RL algorithm, including our GEPO as well as baseline methods such as GRPO and GSPO.
>
> For example, **AReaL** incorporates a suite of system-level optimizations that significantly improve GPU utilization. **StreamRL** targets pipeline bubbles and skewness bubbles in the execution flow. **AsyncFlow** introduces a distributed data storage and transfer module that enables unified data management and fine-grained scheduling in a fully streamed manner.
> Since these are system frameworks rather than algorithms, **it is not methodologically appropriate to directly compare system frameworks with GEPO algorithm**.
>
> Regarding your reference [1]—*Magistral*—it is a comprehensive technical report containing numerous engineering optimizations. However, the authors do not propose a new RL policy optimization algorithm. As explicitly stated in Section 2.1 ("Reinforcement Learning Algorithm") of their report:
> > “**We use Group Relative Policy Optimization (GRPO) [Shao et al., 2024] as our RL algorithm**.”
>
> However, GRPO is a baseline compared in our experiments.
> To address your concern, we have incorporated the engineering techniques from *Magistral* [1] with the GRPO algorithm as an additional baseline for comparison. **Furthermore, we add some new asynchronous policy optimization algorithms as our baselines CISPO [5] and TOPR [6], both of which are recent works (on June 16 / 2025 and May 19 / 2025, respectively).** **Truncated IS** is the key technique of **IMPALA**[7], we direcrlt compare this technique with GEPO.
> The mathematical distinctions among these three methods are as follows:
>
> **Truncated IS**:
> $$
> \mathcal{J}^{\text{TIS}}(\pi\_{\theta}) = \mathbb{E}\_{\tau \sim \pi\_{\text{old}}} \left[ \operatorname{sg}\left( \frac{\pi\_{\theta}(\tau)}{\pi\_{\text{old}}(\tau)} \right)\_{0}^{1} R(\tau) \log \pi\_{\theta}(\tau) \right]
> $$
>
>
> **CISPO**:
> $$
> \mathcal{J}^{\text{CISPO}}(\pi\_{\theta}) = \mathbb{E}\_{\tau \sim \pi\_{\text{old}}} \left[ \operatorname{sg}\left( \frac{\pi\_{\theta}(\tau)}{\pi\_{\text{old}}(\tau)} \right)\_{1 - \epsilon\_{\text{low}}^{\text{IS}}}^{1 + \epsilon\_{\text{high}}^{\text{IS}}} R(\tau) \log \pi\_{\theta}(\tau) M(\tau) \right]
> $$
>
>
>
>
> **TOPR**:
> $$
> \mathcal{J}^{\text{TOPR}}(\pi\_{\theta}) = \mathbb{E}\_{\tau \sim \pi\_{\text{old}}} \left[ \left( \mathbf{1}\_{\{\tau \in T^{+}\}} + \mathbf{1}\_{\{\tau \in T^{-}\}} \operatorname{sg}\left( \frac{\pi\_{\theta}(\tau)}{\pi\_{\text{old}}(\tau)} \right)\_{0}^{1} \right) R(\tau) \log \pi\_{\theta}(\tau) \right]
> $$
>
> We follow the setting of CISPO and TOPR in their papers, both of them state that **"does not require an explicit KL penalty to guarantee stable behaviour"**. However, in our experiments, both of them are fragile. Hence, we add **CISPO/TOPR+KL** as new baselines for fair comparison.

---

> ### Author Response · Authors · 2025-11-21
> **Response to Reviewer aJa6 Part-2/6**
>
> All experiments were conducted under the same main experimental setup as described in our paper. The results on the Qwen3-1.7B (with 2k limitation) model are shown below:
>
>
> | Method                | AMC2023 (Best/Last) | AIME2024 (Best/Last) | AIME2025 (Best/Last) | MATH500 (Best/Last) | Average (Best/Last) |
> |-----------------------|------------------------|-------------------------|-------------------------|------------------------|------------------------|
> | GEPO     | **53.8/53.8**            | **21.9/21.9**             | **18.8/18.8**             | **79.6/79.6**           | **43.5/43.5**           |
> |Magistral (GRPO wt trick) |  46.8/46.5 | 19.1/14.4 | 14.8/14.8 | 74.8/74.8 | 38.9/37.6|
> |Truncated IS + KL [7]  |  47.5/0.0       |   15.2/0.0     |     15.6/0.0   |    74.0/0.0     | 38.1/0.0 |
> |TOPR w/o KL [6]|  51.3/0.0 | 17.2/0.0 | 15.6/0.0 | 76.4/0.0 | 40.1/0.0 |
> |TOPR + KL [6] |51.9/30.0 |17.2/1.2 |17.2/7.0 |76.9/55.1 |40.8/23.3 |
> |CISPO w/o KL [5]| 44.1/20.0 | 10.2/0.8 | 12.1/0.8 | 70.7/43.4 | 34.3 /16.3 |
> |CISPO + KL [5]|44.1/21.2 |9.8/2.3 |12.5/2.0 |71.8/41.9 |34.5/16.9 |
>
> As shown in the experimental results, GEPO continues to demonstrate advantages in both performance and stability.  **The training curves are presented in new section K (Comparison With Asynchronous RL Baselines)**
>
> Below is our analysis of the experimental findings (we will refrain from excessive numerical analysis here).
> Regarding CISPO, its improved weight clipping strategy for importance weights is indeed more effective. However, its optimization objective operates at the **token level**: for tokens whose importance ratios are clipped, the corresponding gradients are set to zero. This is the direct reason for its slower reward growth (even though the clipping range has already been expanded). This observation is consistent with the explanation in our paper for the slow reward increase of GRPO, as illustrated in Figure 4(c). Both CISPO and GRPO experiments directly address your **Weakness #5**: in asynchronous settings, token-level importance weights are prone to exceeding the clipping threshold, leading to invalid gradient (i.e., being zeroed out). **While clipping enhances training stability, it comes at the cost of reduced performance, a classic engineering trade-off between performance and stability**.
>
> **Compared with weight-clipping-based methods (CISPO/TOPR/Truncated IS), GEPO offers three advantages:**
>
> 1) GEPO computes importance weights via intra-group averaging, which provides **higher tolerance for the range of individual token importance weight**.
>
> 2) GEPO can deliver **performance gains in online settings**, a capability absent in weight-clipping methods, since their importance weights are always exactly 1 in online scenarios.
>
> 3) GEPO **does not introduce any new hyperparameters** (e.g., additional clipping thresholds). The group size hyperparameter is inherited from GRPO and is not newly added by GEPO.
>
> ###  **New References**
>
> [5] MiniMax-M1: Scaling Test-Time Compute Efficiently with Lightning Attention, Arixiv 2506.13585, 2025.
>
> [6] TAPERED OFF-POLICY REINFORCE Stable and efficient reinforcement learning for LLMs, NeurIPS 2025.
>
> [7] Impala: Scalable distributed deep-rl with importance weighted actor-learner architectures, ICML 2018.

---

> ### Author Response · Authors · 2025-11-21
> **Response to Reviewer aJa6 Part-3/6**
>
> ### **Response to Weakness-2: Equation (1) appears very similar to PPO**
> **Your Original Comments:**
> > "Equation (1) appears very similar to PPO, except that the clipping function is removed."
>
> Actually, Eq.(1) is an equation to describe our optimization objective. It is not our proposed method.
>
> The objective of PPO is $\frac{\pi_{\theta}}{\pi_{\text{old}}}$, which is a broad concept. It does not specify what causes the discrepancy between $\pi_{\theta}$ and $\pi_{\text{old}}$.
> In contrast, our objective $\frac{\pi_{t+\tau}}{\pi_{t}}$ explicitly introduces staleness $\tau$ to indicate that, in heterogeneous RL settings, the difference between the old and new policies arises from explicit factors such as **model synchronization time**, **data synchronization time**, **rollout time** and including even heterogeneous computing resources.
>
> ---
>
> ### **Resonse to Weakness-3:  About the meanning of G, A, and p(Sync Model Time, Sync Data Time, Rollout Time)**
> **Your Original Comments:**
> > "In line 139, the variable $G$ is undefined; in Equation (1), it is unclear how $A(x)$  is computed, and in line 146, the input of $p$ is not specified."
>
> 1. G is the group size.
> 2. A is the advantage; the computation step of A is described in Lines 136-139. In the **Line 137**, we claim that we use the within-group average reward like **GRPO**.
> 3. $\tau \sim p(Sync Model Time, Sync Data Time, Rollout Time)$ denotes that the random variable $\tau$ follows a **probability distribution ($p$)** influenced by factors such as model synchronization time, data synchronization time, and rollout time.
>
> We will further clarify the meanings of the above symbols in a later version to improve the readability of the paper.

---

> ### Author Response · Authors · 2025-11-21
> **Response to Reviewer aJa6 Part-4/6**
>
> ### **Resonse to Weakness-4: Variance Analysis of Advantage-Weighted Importance Sampling**
> **Your Original Comments:**
> > "While Theorem 1 demonstrates a reduction in the variance of the importance sampling coefficient, this result does not guarantee a corresponding reduction in the variance of the weighted term  $A \cdot w$, , because $A$  itself is a random variable. In general, even if one random variable has smaller variance than another (e.g., $Var(Y)<Var(Z)$), it does not necessarily follow that $Var(XY)<Var(XZ)$, unless $X$  is independent of both $Y$ and $Z$ and their expectations are zero."
>
> We give the extended Theorem 1 for $Var(A \cdot w)$ following (Theorem 3 in our new manuscript):
>
> #### Theorem 1 (Extended)
> Let $p(y|x)$, $q(y|x)$ be discrete probability distributions and $A(y)$ be a bounded advantage function satisfying $\mathbb{E}_p[A(y|x)] = 0$.
> For notational simplicity, we omit the dependence on $x$.
>
> We first note that any sample $ y $ with $ A(y) = 0 $ contributes **zero gradient** to the policy update and **requires no variance control**, as it does not affect learning. Therefore, without loss of generality, we can remove the training samples which $A(y)=0$. Our variance analysis focuses only on the support where $ |A(y)| > 0 $. Since the advantage function is assumed bounded and not identically zero, we can further restrict to the subset where $ A(y)^2 \geq A_{\min}^2 > 0 $ for some constant $ A_{\min}$. Within the **Group Wise Advantage** method like GRPO, $ A_{\min} = 1/G$, whrer $G$ is the group size.
>
>
> Then there exists a constant $C_{\text{adv}}$ such that:
>
> $$\mathrm{Var}\_q\left(A(y)\cdot\frac{p(y)}{q(y)}\right) - \mathrm{Var}\_q\left(A(y)\cdot\frac{p(y)}{\mathbb{E}\_q[q(y)]}\right) \geq A\_{\min}^2 \cdot \exp(D\_{\mathrm{KL}}(p \| q)) - C\_{\text{adv}}$$
>
> where $A_{\min} = \min_y |A(y)|>0$, $A_{\max} = \max_y |A(y)|$,  $C_{\text{adv}} = \frac{A_{\max}^2}{|| q||_2^2}$ is a constant, which we specify later, and $\mathbb{E}_q[q(y)] = \int q(y)^2 dy$.

---

> ### Author Response · Authors · 2025-11-21
> **Response to Reviewer aJa6 Part-5/6**
>
> ### **Proof**
>
> ### Step 1: Variance Expressions
>
> Following the notation in Appendix A, we define:
>
> - $Z_1 = A(y) \cdot \frac{p(y)}{q(y)}$
> - $Z_2 = A(y) \cdot \frac{p(y)}{|| q||_2^2}$ where $|| q||_2 = \mathbb{E}_q[q] = \int q(y)^2 dy$
>
> The variances are:
>
> $$\mathrm{Var}_q(Z_1) = \mathbb{E}_q\left[\left(A(y) \cdot \frac{p(y)}{q(y)}\right)^2\right] - \left(\mathbb{E}_q\left[A(y) \cdot \frac{p(y)}{q(y)}\right]\right)^2 = \int A(y)^2 \frac{p(y)^2}{q(y)} dy - \left(\int A(y) p(y) dy\right)^2$$
>
> $$\mathrm{Var}_q(Z_2) = \frac{1}{|| q||_2^2} \left[\mathbb{E}_q[A(y)^2 p(y)^2] - \left(\mathbb{E}_q[A(y) p(y)]\right)^2\right] = \frac{1}{|| q||_2^2} \left[\int A(y)^2 p(y)^2 q(y) dy - \left(\int A(y) p(y) q(y) dy\right)^2\right]$$
>
> ---
>
> ### Step 2: Variance Difference
>
> Define $\Delta_{\text{adv}} = \mathrm{Var}_q(Z_1) - \mathrm{Var}_q(Z_2)$:
>
> $$\Delta_{\text{adv}} = \left[\int A(y)^2 \frac{p(y)^2}{q(y)} dy - \left(\int A(y) p(y) dy\right)^2\right] - \frac{1}{|| q||_2^2} \left[\int A(y)^2 p(y)^2 q(y) dy - \left(\int A(y) p(y) q(y) dy\right)^2\right]$$
>
> Rearranging terms:
>
> $$\Delta_{\text{adv}} ={\int A(y)^2 \frac{p(y)^2}{q(y)} dy} - {\frac{1}{|| q||_2^2} \int A(y)^2 p(y)^2 q(y) dy} - {\left(\int A(y) p(y) dy\right)^2} + {\frac{1}{|| q||_2^2} \left(\int A(y) p(y) q(y) dy\right)^2}$$
>
> For shortness, we define:
>
> $$\Delta_{\text{adv}} = J_1 - J_2 -B_1 +B_2 $$
>
>
> ---
>
> ### Step 3: Lower Bound for $ J_1 $
>
>
> Under the restriction that $ A(y)^2 \geq A_{\min}^2 > 0$ , we have:
>
> $$
> J_1 = \int A(y)^2 \frac{p(y)^2}{q(y)} \, dy \geq A_{\min}^2 \int \frac{p(y)^2}{q(y)} \, dy = A_{\min}^2 \cdot I_1,
> $$
>
> where $ I_1 = \int \frac{p(y)^2}{q(y)} \, dy $ as defined in Appendix A.
>
> From the relationship between KL and $ \chi^2 $ divergences:
>
> $$
> D_{\mathrm{KL}}(p || q) \leq \log(I_1) \implies I_1 \geq \exp(D_{\mathrm{KL}}(p || q)),
> $$
>
> and therefore:
>
> $$
> J_1 \geq A_{\min}^2 \cdot \exp(D_{\mathrm{KL}}(p || q)).
> $$
>
> ---
>
> ### Step 4: Bounding the Remaining Terms
>
> For $J_2$, since $|A(y)| \leq M$ for some constant $M$ (as advantages are bounded in practice, for GRPO $M \leq 1$):
>
> $$J_2 = \frac{1}{|| q||_2^2} \int A(y)^2 p(y)^2 q(y) dy \leq \frac{A\_{\max}^2}{|| q||_2^2} \int p(y)^2 q(y) dy \leq \frac{A\_{\max}^2}{|| q||_2^2},$$
> where the second inequality follows from Lemma 1 on page 15 of our paper:
>
> $$\sum_{i=1}^n p_i^2 q_i \leq 1$$
>
> For the bias terms $B_1$ and $B_2$, since $A(y)$ is bounded and $p(y)$, $q(y)$ are probability distributions:
>
> $$|B_1| = \left|\int A(y) p(y) dy\right|^2 =0$$
>
> $$|B_2| = \left|\frac{1}{|| q||_2^2} \left(\int A(y) p(y) q(y) dy\right)^2\right| > 0 $$
>
> ---
>
> ### Step 5: Combining Bounds
>
> Combining all terms:
>
> $$\Delta_{\text{adv}} = J_1 - J_2 - B_1 + B_2 \geq A_{\min}^2 \cdot \exp(D_{\mathrm{KL}}(p || q)) - \frac{A_{\max}^2}{|| q||_2^2} - 0 + 0$$
>
> Note that $B_2$ appears with a positive sign in $\Delta_{\text{adv}}$, which helps our lower bound. Since $B_2$ is bounded, we can define a constant $C_{\text{adv}}$ that absorbs all bounded terms:
>
> $$C_{\text{adv}} = \frac{A_{\max}^2}{|| q||_2^2}$$
>
> In practice, as noted in Appendix A, $ || q||\_2 = \mathbb{E}\_q [q] \approx 0.432$ for language model distributions and $A_{\max}=\max{|A|}\leq1$, making $C_{\text{adv}}$ a reasonable constant. Therefore:
>
> $$\Delta_{\text{adv}} \geq A_{\min}^2 \cdot \exp(D_{\mathrm{KL}}(p || q)) - C_{\text{adv}}$$
>
> This completes the proof. The result shows that GEPO's advantage-weighted importance sampling maintains the exponential variance reduction property of Theorem 1, even when considering the full policy gradient term $A(y) \cdot w(y)$. The constant $C_{\text{adv}}$ incorporates the positive contribution from $B_2$, making our bound tighter than if we had only considered the main terms $J_1$ and $J_2$.

---

> ### Author Response · Authors · 2025-11-21
> **Response to Reviewer aJa6 Part-6/6**
>
> ### **Resonse to Weakness-5: about weight clipping**
> **Your Original Comments:**
> >"The claim in lines 178–181 seems questionable. In extreme cases, clipping can suppress unstable tokens by assigning them zero weight, which may be beneficial since such extreme tokens usually correspond to unreliable generations. Thus, clipping might actually serve as a useful safeguard in these scenarios."
>
> We briefly addressed this issue in our response to **Weakness #1**, but here I’d like to share my own perspective on weight clipping and related works.
>
> Many asynchronous RL methods approach the problem from the perspective of weight clipping, as instability often stems from importance weights. As we described in our paper, the variance of importance weights is positively correlated with both the number of asynchronous steps and the KL divergence. Consequently, weight clipping is an intuitively appealing strategy for mitigating such instability. However, as noted in lines 178–181, this comes at a cost: clipping discards updates from a subset of otherwise valid tokens. These tokens may carry critical information, and once their gradients are zeroed out, the overall performance inevitably suffers.
>
> This is precisely the main reason (highlighted at the end of our response to Weakness #1) why methods like GRPO and CISPO, while stable at the begining of training, underperform compared to GEPO. The same conclusion can also be drawn from the GRPO reward curve in Figure 4(c) of our paper: **“Stability comes at the expense of a reduced slope in the reward curve.”**
>
> In contrast, GEPO computes importance weights by averaging over an entire group of tokens (rather than a single sequence), which greatly increases tolerance to large token weights. This design preserves the informative signals that would otherwise be lost in GRPO due to clipping. As a result, GEPO achieves a superior balance, delivering both high performance and strong stability.
>
> To further strengthen our argument, we quote the original text from the TOPR[6] paper, which aligns closely with our viewpoint.
>
> > “However, the PPO objective applies the importance ratio to the reward, rather the gradient; as a consequence, the gradient of Jppo(π) becomes zero outside of the [1 − ϵ, 1 + ϵ] range, limiting its usefulness and potentially causing brittleness when more than a handful of updates are made before resampling trajectories. Similarly, the algorithm is not incentivized to reduce the relative probability of negative trajectories below 1 − ϵ, limiting the potential improvement from µ. Although variants such as sPPO increase this robustness [Vaswani et al., 2022], their performance still drops after a few tens or hundreds of updates. ”
>
> ### **References**
> [6] TAPERED OFF-POLICY REINFORCE Stable and efficient reinforcement learning for LLMs, NeurIPS 2025.

---

> ### Author Response · Authors · 2025-11-24
> **We are eagerly looking forward to your reply**
>
> Dear Reviewer,
>
> We have provided detailed responses to all of your questions. At this point, do you still have any remaining concerns? Are there any aspects of our replies that you find questionable or insufficient? Have our responses fully addressed your doubts?
>
> Given your high confidence score, it seems your primary concern lies in the choice of baselines for comparison with our method. In our earlier response, we clarified that the approaches you mentioned are fundamentally orthogonal to our work—we refer to them as “algorithm-agnostic.” These methods cannot be directly compared with GEPO in terms of performance. However, we could implement GEPO within your mentioned asynchronous RL frameworks such as Areal, and after the review period concludes, we would be happy to submit a pull request (PR) to their GitHub repositories to integrate our method.
>
> Would this approach adequately address your concerns?
>
> Additionally, is there any part of our paper that remains unclear to you? We would be glad to provide thorough and detailed clarifications for any of your questions, and we sincerely welcome your valuable feedback.
>
> **There is still ample time, and we are happy to provide further clarifications or conduct additional experiments to address any concerns you may have.**

---

> > ### Comment · Reviewer_aJa6 · 2025-11-26
> >
> > We thank the reviewers for their response. Some of my concerns have been addressed well. However, I still have some questions, as follows:
> >
> > 1. Why does the computation of $A$ not use a denominator as in GRPO? How exactly does the denominator affect the performance of the policy model? A detailed theoretical analysis is needed here. In addition, since the denominator is removed in $A$ while the denominator in the importance weighting is modified, do these two changes cancel each other out?
> >
> > 2. When the distributions $p$ and $q$ are close, how can we guarantee that the contrast in importance weights still favors the better one? In this case, the lower bound may become negative. This issue arises in both Theorem 1 and Theorem 1 (Extended).
> >
> > 3. I still do not understand how GEIW actually facilitates the learning of the policy model. The authors mainly attempt to explain this through experimental results. If so, the paper reads more like an experimental report and lacks high-level explanations, justification, and adequate theoretical analysis.
> >
> > 4. Given the core claims of this paper, I would like to understand how it differs from the following three works:
> >    a. BAPO: Stabilizing Off-Policy Reinforcement Learning for LLMs via Balanced Policy Optimization with Adaptive Clipping;
> >    b. VL Norm: RETHINK LOSS AGGREGATION IN RLVR;
> >    c. BNPO: Beta Normalization Policy Optimization.
> >
> > In view of the above issues, I believe that the paper currently requires a major revision, which is unlikely to be completed within the remaining time. Therefore, I maintain my original score.

---

> ### Author Response · Authors · 2025-11-26
> **Thank you for your response**
>
> Thank you very much for your thoughtful review and for taking the time to engage deeply with our work. We sincerely appreciate your feedback, which has helped us reflect on how to better communicate our contributions. Below, we address each of your new questions with care and humility.
>
> ---
>
> ### **Response to New Question-1**
> We have provided a detailed explanation in the main paper and further validated it through the ablation study in **Appendix C.1**, which isolates the contribution of each component of GEPO.
>
> ---
>
> ### **Response to New Question-2**
> As shown in **Figure 2**, the regime in which variance increases is extremely rare under practical heterogeneous RL settings. The dominant source of instability in such settings stems from large KL divergence due to policy staleness. Our method is specifically designed to mitigate this issue, and **Figure 4** demonstrates that GEPO consistently yields lower variance and smoother training dynamics. These empirical findings align closely with our theoretical analysis.
>
> ---
>
> ### **Response to New Question-3**
> We understand your comment that the mechanism of GEPO remains unclear. This is surprising to us, as we have dedicated substantial effort—both in theory and experiments—to elucidate why GEPO works. For completeness, we summarize key endorsements from other reviewers on this matter:
>
> #### **Experimental Clarity**
> > **Reviewer 8QcJ**: “Curves show **lower IW variance and smoother gradients.**”
> > **Reviewer bBKw**: “The motivation, instability and variance explosion in asynchronous or heterogeneous RL due to policy staleness, **is well validated by experimental results**. GEPO **achieves lower variance, smoother gradients, and higher stability**.”
> > **Reviewer 5yWm**: “GEPO **maintains accuracy and avoids collapse** across delays; best/last markedly stronger than GRPO/GSPO.”
>
> #### **Theoretical Insight**
> > **Reviewer 8QcJ**: “Theorem 1 relates the variance gap to exp(D_KL), motivating **why the method helps exactly when staleness is worst.**”
> > **Reviewer 5yWm**: “**It gives exponential variance reduction of importance weights in high-KL regimes**; mechanistic gradient comparison clarifies **why updates stabilize**.”
>
> If certain parts of our explanation remain ambiguous, we would be very grateful for your guidance on which specific aspect needs further clarification.
>
> ---
>
> ### **Response to New Question-4**
> We appreciate the opportunity to clarify how GEPO differs from related methods:
>
> - **BAPO** improves performance by dynamically adjusting clipping bounds to enhance positive-sample signals.
> - **VL Norm** constructs a minimum-variance unbiased policy gradient estimator that reweights samples based on response length.
> - **BNPO** normalizes binary rewards using an adaptively updated Beta distribution to align reward scaling with policy evolution.
> - **GEPO**, in contrast, operates at the *group level* by reweighting importance weights via group expectations. This mechanism directly importance the variance induced by policy divergence.
>
> These approaches address distinct sources of instability, and GEPO’s group-level weighting is orthogonal to dynamic clipping (BAPO), length-aware normalization (VL Norm), or reward normalization (BNPO).
>
> ---
>
> ### **Our Request for Clarification – Regarding Confidence and Evaluation Criteria**
>
> We fully respect your final decision and the perspective you have shared. That said, to help us grow as researchers and improve future submissions, we would be deeply grateful for two small clarifications:
>
> 1. **On confidence**: You assigned a confidence score of 5, indicating high certainty in your evaluation. Since you also noted that the core mechanism of GEPO is unclear to you ( **you state "I still do not understand how GEIW actually..."**), we wonder whether there might be a misalignment in how the method was presented. If possible, could you point to the specific part of our theoretical or experimental explanation that failed to convey why GEPO works? This would be immensely valuable for our revision.
>
> 2. **On theoretical expectations**: You commented that the paper “reads more like an experimental report and lacks high-level justification and adequate theoretical analysis.” However, our manuscript includes approximately **seven pages of formal theoretical analysis**, and—based on our reading—**the recent works you cited (e.g., [1–4]) do not present any formal theoretical guarantees either**. To better meet community standards, could you kindly share what you consider the hallmark of a well-justified theoretical contribution in this area?
>
> ---
>
> ### **Thanks Again**
>
> Regardless of whether any adjustments to your score are possible, we genuinely value your critique. Constructive feedback like yours is essential for advancing rigorous and impactful research. Thank you again for your time, expertise, and the care you took in reviewing our work.
>
> ---

---

### Official Review · Reviewer_8QcJ · 2025-10-31

**Soundness:** 3
**Presentation:** 3
**Contribution:** 2
**Rating:** 6
**Confidence:** 2

**Summary:**

The paper proposes HeteroRL, a decentralized RL framework that decouples rollout sampling from learning to tolerate Internet-scale latency and heterogeneous hardware. Its core algorithm, GEPO (Group Expectation Policy Optimization), replaces token/sequence-level importance weights with a group-level expectation to reduce the variance explosion that arises when the learner’s policy drifts from the samplers due to staleness. A theoretical result shows variance reduction grows (approximately) exponentially with KL divergence between policies, and experiments on Qwen3-1.7B/8B with math-reasoning benchmarks (AIME/MATH/AMC) report higher accuracy and markedly improved stability under simulated delays (up to 1800s).

**Strengths:**

**Clear systems problem + algorithmic handle**. The paper crisply diagnoses policy staleness in decentralized RL and ties it to KL growth → variance blow-ups; GEPO’s denominator uses a within-group expectation to damp variance in precisely that regime.

**Compelling empirical stability**. Under Hetero RL with max delay 64, GEPO improves best accuracy vs. GRPO/GSPO and, crucially, reduces best-to-last degradation by ~85% vs. GSPO (Δ=1.8 vs. 12.0). Curves show lower IW variance and smoother gradients.

**Theoretical insight**. Theorem 1 relates the variance gap (standard IS vs. group-expectation weight) to exp(D_KL), motivating why the method helps exactly when staleness is worst.

**Realistic setting**. The system model explicitly injects stochastic network delay and hetero nodes (Ascend + NVIDIA), aligning with decentralized community compute setups.

**Weaknesses:**

**Bias–variance trade-off left under-quantified in RL objective**. GEPO’s estimator is acknowledged as biased; while lower variance can help optimization, the paper does not quantify end-to-end bias in policy gradients or returns beyond variance plots. A small-bias claim would benefit from controlled ablations where true on-policy gradients are approximated (short-horizon toy MDPs) to measure bias vs. sample efficiency. (GEIW is described as biased but stable.)

**External validity beyond math-reasoning**. Results are limited to math QA on Qwen. It would help to show GEPO under non-text or mixed-modality tasks, or code/data generation tasks where rollouts have different length/entropy profiles. Current related work cites broader systems, but experiments remain narrow.

**Ablations & knobs**. The method depends on group size G and sampling strategy (top-k/p). Sensitivity plots for G, the truncation strategy, and the CPPO-KL coefficient under delay would strengthen the story. (Implementation notes mention CPPO-KL and latency simulation, but tuning studies are light.)

**Fairness to GSPO variants**. Since GSPO is sequence-level IS with clipping, it would be good to include GSPO + stronger clipping / trust-region or defensive mixture baselines to test whether variance spikes can be similarly tamed by tuned GSPO, not just GEPO’s denominator. The paper itself sketches “defensive sampling” as future work—great idea, but makes me wonder how close-tuned GSPO would get.

**Questions:**

See above.

---

> ### Author Response · Authors · 2025-11-21
> **Summary of Response to Reviewer 8QcJ**
>
> We sincerely thank you for your thoughtful review and constructive feedback that has significantly strengthened our paper. **Because our responses are quite extensive—spanning new bias quantification, theoretical bounds, broadened justification of benchmarks, comprehensive ablations, and fair GSPO variant comparisons—we provide the following concise summary at the beginning of our rebuttal to help you and the Area Chair quickly grasp the key updates.** If you are interested in specific details or would like to verify any claims, please refer to the corresponding sections below. We also warmly welcome any follow-up questions. Thank you once again for your thoughtful and patient review.
>
> ### Concise Summary of Response
>
> - **Bias–variance trade-off**: We empirically quantify the end-to-end bias of GEPO vs. GSPO and show they are comparable in magnitude. More importantly, we derive a novel theoretical upper bound showing GEPO’s bias is bounded by the L2-norm ratio $\frac{||p||_2}{||q||_2}$, which remains near 1 in real LLM settings due to sharp output distributions—ensuring mild and controllable bias.
>
> - **External validity**: While experiments focus on math reasoning—a standard and challenging benchmark in recent LLM-RL literature (e.g., NeurIPS/ICLR/ICML 2025 papers)—we clarify that its sparse rewards, long reasoning chains, and sensitivity to policy staleness make it an ideal stress test for latency robustness, directly aligning with GEPO’s core contribution.
>
> - **Ablation studies**: We conduct extensive sensitivity analyses across group size \(G\), CPPO-KL coefficient, delay distributions, and sampling hyperparameters (top-\(k\)/top-\(p\)/temperature). Results confirm GEPO’s robustness and provide practical guidance for tuning under system heterogeneity.
>
> - **Fairness to GSPO**: We evaluate multiple enhanced GSPO variants—including tighter clipping, trust-region filtering, entropy regularization, and defensive sampling—and find that even the best-performing variant still underperforms GEPO in both final accuracy and stability under delay, validating GEPO’s structural advantage.
>
> # **If formulas do not render properly, we recommend refreshing the page or referring to our latest submitted manuscript.**

---

> ### Author Response · Authors · 2025-11-21
> **Response to Reviewer 8QcJ Part-1/7**
>
> ### **Sec 1 To weakness 1: Bias–variance trade-off left under-quantified in RL objective**
> **Your Original Comments:**
> >"**Bias–variance trade-off left under-quantified in RL objective.** GEPO’s estimator is acknowledged as biased; while lower variance can help optimization, the paper does not quantify end-to-end bias in policy gradients or returns beyond variance plots. A small-bias claim would benefit from controlled ablations where true on-policy gradients are approximated (short-horizon toy MDPs) to measure bias vs. sample efficiency. (GEIW is described as biased but stable.)"
>
> We appreciate the reviewer’s suggestion to quantify bias via toy MDPs. However, in the context of large language model RL, where the action space is high-dimensional and rewards are sequence-level, the behavior of importance sampling estimators differs fundamentally from low-dimensional MDPs.
>
> #### Sec 1.1 The end-to-end bias
>
> From the experimental level, we can compute the following bias of GEPO and GSPO:
>
>
> $$Bias(GEPO) =\left |\mathbb{E}\_p\left[A(y) \right] - \mathbb{E}\_q\left[ \frac{p(y)}{ \mathbb{E}\_q q} A(y) \right] \right | = \left | 0-\mathbb{E}\_q\left[ \frac{p(y)}{ \mathbb{E}\_q q} A(y) \right] \right | \approx \left | \frac{1}{G} \sum_{i=1}^{G} \left[ \frac{p(y^i)}{ \mathbb{E}\_q q} A(y^i) \right] \right |$$
>
> $$Bias(GSPO) =  \left |\mathbb{E}\_p\left[A(y) \right] - \mathbb{E}\_q\left[ \frac{p(y)}{q(y)} A(y) \right] \right |= \left |0 - \mathbb{E}\_q\left[ \frac{p(y)}{q(y)} A(y) \right] \right | \approx \left | \frac{1}{G} \sum\_{i=1}^{G} \left[ \frac{p(y^i)} {q(y^i)} A(y^i) \right] \right |$$
>
> where $G$ is group size. Here, the bias of GSPO serves as a baseline for assessing the relative magnitude of the bias in GEPO. The use of the $\approx$ symbol is due to our use of Monte Carlo estimation for this expected statistic.
>
> The average and variance of $Bias(GEPO)$ and $Bias(GSPO)$ during 512 global trainings steps are:
>
> |       | AVG    | VAR     |
> |-------|--------|---------|
> | $Bias(GEPO)$  | 0.072 | 0.001  |
> | $Bias(GSPO)$  | 0.075| 0.001  |
>
>
> We clarify the following point to avoid misunderstanding:
> The VAR reported in the table above refers to the variance of the bias estimator, which **is different from the IS weight variance analyzed in Theorem 1**. Therefore, the observation that the VAR values of GSPO and GEPO are similar in the table does **not contradict** Theorem 1 in the original paper.
> We plot the two end-to-end bias curves in **Appendix J.1**.
>
> ### **Sec 1.2 The theorical bound of GEPO bias**
>
> To futher adress your concern, we derive an upper bound for the **bias** introduced by GEPO’s group expectation importance weighting, specifically the absolute difference between the standard importance sampling (IS) estimator and GEPO’s estimator **in expectation**, under the assumption that the advantage function $ A(y) $ is **zero-mean under the target policy $ p $** (i.e., $ \mathbb{E}_p[A] = 0 $).
>
> We will show that:
>
> $$
> Bias(\text{GEPO})=\left| \mathbb{E}_q\left[ \frac{p(y)}{q(y)} A(y) \right] - \mathbb{E}_q\left[ \frac{p(y)}{ \mathbb{E}_q q} A(y) \right] \right| < \frac{||p||_2}{||q||_2}.
> $$
>
> ---
>
>
>
>
> The term $\frac{||p||_2}{||q||_2}$ denotes the L2-norm ratio of p and q. The related concept is **Renyi entropy** of p and q.
> In LLM reinforcement learning, the ratio $\frac{||p||_2}{||q||_2}$—which upper-bounds the bias introduced by GEPO’s group expectation weighting—remains modest in practice. Although this ratio can theoretically reach $\sqrt{n}$ when $p$ is one-hot and $q$ is uniform over $n$ actions, real-world LLM policies are highly non-uniform due to top-$k$/top-$p$ sampling and sharp output distributions. Empirical measurements in the paper (e.g., Appendix A) show that $||q||_2^2 \approx 0.432$ for Qwen3-1.7B, implying $||q||_2 \approx 0.657$, and $p$ typically exhibits similar concentration. Consequently, $\frac{||p||_2}{||q||_2} \approx 1$ in general, even under high policy divergence, ensuring that GEPO’s bias remains mild and controllable—far from the catastrophic scale implied by worst-case theoretical bounds.
>
>
> ---

---

> ### Author Response · Authors · 2025-11-21
> **Response to Reviewer 8QcJ - Part2/7**
>
> ### **Sec 1.3 The following is proof process:**
>
> #### Step 1: Define Estimators
>
> Let:
> - $ p(y) = \pi_{\text{new}}(y|x) $,
> - $ q(y) = \pi_{\text{old}}(y|x) $,
> - $ A(y) $: advantage function with $ \mathbb{E}_p[A] = 0 $. Without loss of generality, we assume that $|A|\leq 1$.
>
> Define:
> - Standard IS estimator expectation:
>   $$
>   \mu_1 = \mathbb{E}_q\left[ \frac{p(y)}{q(y)} A(y) \right] = \sum_y p(y) A(y) = \mathbb{E}_p[A] = 0.
>   $$
> - GEPO estimator expectation (using $ \hat{q} = \mathbb{E}_q[q] = \sum_y q(y)^2 = ||q||_2^2 $):
>   $$
>   \mu_2 = \mathbb{E}_q\left[ \frac{p(y)}{ \mathbb{E}_q q} A(y) \right] = \frac{1}{||q||_2^2} \sum_y p(y) q(y) A(y).
>   $$
>
> Thus, the **bias** is:
> $$
> |\mu_1 - \mu_2| = |\mu_2| = \left| \frac{1}{||q||_2^2} \sum_y p(y) q(y) A(y) \right|.
> $$
>
>
> ##### Step 2: Apply Cauchy–Schwarz Inequality
>
>
> We bound the absolute value of the inner product by Cauchy–Schwarz:
> $$
> \left| \sum_y p(y) q(y) A(y) \right|
> \leq \sqrt{ \sum_y (p(y) A(y))^2 } \cdot \sqrt{ \sum_y q(y)^2 }.
> $$
>
> Since $ |A(y)| \leq 1 $, we have $ A(y)^2 \leq 1 $, so:
> $$
> \sum_y (p(y) A(y))^2 < \sum_y p(y)^2 = ||p||_2^2.
> $$
>
> The reason we use $<$ instead of $\leq$ here is that the "advantage $|A(y)| = 1$ for all posible LLM response $y$" cannot occur.
>
>
> Therefore:
> $$
> \left| \sum_y p(y) q(y) A(y) \right| < ||p||_2 \cdot ||q||_2.
> $$
>
>
>
> ##### Step 3: Combine with Denominator
>
> Now plug into the expression for $ |\mu_2| $:
> $$
> |\mu_2| = \left| \frac{1}{||q||_2^2} \sum_y p(y) q(y) A(y) \right| < \frac{ ||p||_2 \cdot ||q||_2 }{ ||q||_2^2 } = \frac{ ||p||_2 }{ ||q||_2 }.
> $$
>
>
> ##### Final Result
>
> $$
> \left| \mathbb{E}_q\left[ \frac{p}{q} A \right] - \mathbb{E}_q\left[ \frac{p}{||q||_2^2} A \right] \right| < \frac{||p||_2}{||q||_2}
> $$
>
> This upper bound quantifies the **bias** of GEPO’s estimator relative to the unbiased standard IS estimator. In large language models, both $ ||p||_2 $ and $ ||q||_2 $ are typically on the order of $ \sqrt{0.4} \approx 0.63 $ (as shown in Appendix A), so their ratio remains close to 1—indicating **mild, controllable bias**.

---

> ### Author Response · Authors · 2025-11-21
> **Response to Reviewer 8QcJ Part-3/7**
>
> ### **Sec 2 To weakness 2: External validity beyond math-reasoning**
> **Your Original Comments:**
> >"**External validity beyond math-reasoning.** Results are limited to math QA on Qwen. It would help to show GEPO under non-text or mixed-modality tasks, or code/data generation tasks where rollouts have different length/entropy profiles. Current related work cites broader systems, but experiments remain narrow."
>
> We appreciate the reviewer's suggestion to broaden our evaluation beyond mathematical reasoning. While we agree that demonstrating GEPO's effectiveness across diverse domains would be valuable, we believe our current experimental design aligns with established practices in the field and addresses the core technical contribution of our work.
>
> Several important considerations support our experimental focus:
>
> 1. **Math-reasoning represents a rigorous evaluation** for RL algorithms of LLM. These tasks feature:
>    - Long, structured, and multi-step reasoning chains that are highly sensitive to policy staleness, making them especially vulnerable to the effects of asynchrony and communication delay;
>    - Extremely sparse, binary correctness rewards, which create sharp, non-smooth optimization landscapes with little gradient signal, posing significant exploration and credit assignment challenges.
>
>    These properties make math reasoning an ideal benchmark for evaluating stability under latency, which are precisely the challenge GEPO addresses.
>
> 2. **Many LLM-RL related methods only evalaue on math-reasoning benchmark**. We only list 3 representative works accepted in 2025:
>
> | Method        | Venue       | Year | Model Size | Benchmarks                              |
> |---------------|-------------|------|----------------|---------------------------------------------------|
> | One-Shot-RLVR[1] | **NeurIPS**     | **2025** | 1.5B/3B/7B           | MATH500, AIME24 and other math benchmarks |
> | TSMC[2]         | **ICLR**        | **2025** | 7B              | MATH500, GSM8K                                       |
> | VinePPO[3]       | **ICML**        | **2025** | 1.1B/7B             | GSM8k, MATH                                       |
>
> We appreciate the reviewer’s push toward broader evaluation and agree it is an important next step, but hope they recognize that math-reasoning serves as a representative and challenging proxy for general long-horizon, sparse-reward text generation under policy lag.
>
> #### References
>
> \[1\] Wang Y, Yang Q, Zeng Z, et al. Reinforcement Learning for Reasoning in Large Language Models with One Training Example. **NeurIPS 2025**
>
> \[2\] Shengyu F, Xiang K, Shuang Ma, et al. Step-by-Step Reasoning for Math Problems via Twisted Sequential Monte Carlo. **ICLR 2025**
>
> \[3\] Kazemnejad A, Aghajohari M, Portelance E, et al. VinePPO: Refining Credit Assignment in RL Training of LLMs. **ICML 2025**
>
> ---

---

> ### Author Response · Authors · 2025-11-21
> **Response to Reviewer 8QcJ Part-4/7**
>
> ### **Sec 3 To weakness 3:Ablations & knobs**
> **Your Original Comments:**
> >"**Ablations & knobs.** The method depends on group size G and sampling strategy (top-k/p). Sensitivity plots for G, the truncation strategy, and the CPPO-KL coefficient under delay would strengthen the story. (Implementation notes mention CPPO-KL and latency simulation, but tuning studies are light.)"
>
> We thank the reviewer for this insightful suggestion. In response, we have conducted a comprehensive set of ablation studies to analyze the sensitivity of GEPO to six key hyperparameters under varying degrees of system delay: group size G , sampling strategies (including top-k , top-p , and temperature), the CPPO-KL regularization coefficient, and the latency simulation distribution (e.g., exponential vs. weiboll vs. lognormal).  **We also include the sensitivity plots for G, the truncation strategy, and the CPPO-KL coefficient in Appendix L (Figure 14).**
>
> ### **Hetero RL Sensitivity Analyses**
>
> **Default Parameters in Main Experiments**: group 8/kl_beta 0.005/delay lognormal
>
> | Hyper-parameters               | AMC2023 (Best/Last) | AIME2024 (Best/Last) | AIME2025 (Best/Last) | MATH500 (Best/Last) | Average (Best/Last) |
> |-----------------------|------------------------|-------------------------|-------------------------|------------------------|------------------------|
> | **group 2**/kl_beta 0.005 |  51.3/48.4 | 17.2/15.6 | 17.2/16.4 | 77.8/75.4 | 40.9/39.0 |
> | **group 4**/kl_beta 0.005 | **56.8/56.8** | 20.3/15.6 | **21.1/21.1** | 77.7/77.7  | **44.0**/42.8|
> | **group 8**/kl_beta 0.005 | 53.8/53.8 | **21.9/21.9**  | 18.8/18.8   | **79.6/79.6**  | 43.5/**43.5**  |
>
>
> **Conclusion**: Larger group sizes lead to more pronounced improvements in training stability.
>
>
> | Method                | AMC2023 (Best/Last) | AIME2024 (Best/Last) | AIME2025 (Best/Last) | MATH500 (Best/Last) | Average (Best/Last) |
> |-----------------------|------------------------|-------------------------|-------------------------|------------------------|------------------------|
> | group 8/**kl_beta 0.001** |48.8/33.4 |14.1/2.0 |17.6/5.5      |75.3/59.5 |38.9/25.1 |
> | group 8/**kl_beta 0.002** |52.5/40.3 |18.0/11.3 |**21.1**/15.2 |77.6/71.4 |42.3/34.6 |
> | group 8/**kl_beta 0.005** | **53.8/53.8** | **21.9/21.9** | 18.8/18.8 | **79.6/79.6** | **43.5/43.5** |
> | group 8/**kl_beta 0.006** |53.4/52.5 |20.7/18.0 |19.1/**19.1** |78.5/77.1 |43.0/41.7 |
> | group 8/**kl_beta 0.008** |47.5/49.7 |12.5/21.5 |14.5/17.6 |74.4/77.4 |37.2/41.5 |
> | group 8/**kl_beta 0.010** |54.1/49.7 |17.6/17.2 |17.2/15.6     |76.9/76.3 |41.4/39.7 |
>
> **Conclusion**: The KL divergence is the most critical hyperparameter for GEPO. Hence, we test more $\beta\_{KL}$ values.  The Above table demonstrates that the KL divergence coefficient is critical for stability in heterogeneous RL settings. With $\beta_{KL}=0.005$, GEPO maintains perfect stability (no performance degradation from best to last). When $\beta_{KL}$ is too small (0.001), the algorithm collapses dramatically (AIME2024 drops from 14.1 to 2.0). When $\beta_{KL}$ is too large (0.010), the model becomes overly conservative, limiting its peak performance.
>
> | Delay Distribution               | AMC2023 (Best/Last) | AIME2024 (Best/Last) | AIME2025 (Best/Last) | MATH500 (Best/Last) | Average (Best/Last) |
> |-----------------------|------------------------|-------------------------|-------------------------|------------------------|------------------------|
> |**weibull (95%CI/60-1920s)** |51.6/49.1 |18.4/14.8 |18.0/17.6 |77.4/77.4 |41.3/39.7 |
> |**lognormal (95%CI/60-1920s)**  | 53.8/53.8            | 21.9/21.9             | 18.8/18.8             | 79.6/79.6            | 43.5/43.5            |
> | **exponential (95%CI/60-1920s)** |53.4/47.8 |19.9/17.2 |16.8/15.2 |79.5/79.4 |42.4/39.9 |
>
> **Conclusion**: The lognormal distribution yields the best performance and stability. The Weibull and exponential distributions show slightly degraded performance, particularly on AIME2024 and AMC2023 benchmarks.
>
> ----

---

> ### Author Response · Authors · 2025-11-21
> **Response to Reviewer 8QcJ Part-5/7**
>
> ###  **Online RL Sensitivity Analyses**
>
> **Default Parameters in Main Experiments**: topP=0.95/topK=20/temp=0.6
>
> | Hyper-parameters               | AMC2023 (Best/Last) | AIME2024 (Best/Last) | AIME2025 (Best/Last) | MATH500 (Best/Last) | Average (Best/Last) |
> |-----------------------|------------------------|-------------------------|-------------------------|------------------------|------------------------|
> | **topP=0.95**/topK=20/temp=0.6 |56.9/56.9 |21.9/16.4 |20.3/14.1 |80.4/78.1 |44.9/41.4 |
> | **topP=0.98**/topK=20/temp=0.6 |55.9/55.9 |**23.8/23.8** |19.5/17.6 |80.3/80.3 |44.9/44.4 |
> | **topP=1.00**/topK=20/temp=0.6 |**59.7/59.7** |22.3/22.3 |**20.7/17.6** |**82.0/82.0** |**46.2/45.4** |
>
> **Conclusion**: Top-$p$=1.00 achieves the highest best performance (46.2) and final performance (45.4). It suggests that total probability provides a better balance between exploration and stability.
>
> | Hyper-parameters                | AMC2023 (Best/Last) | AIME2024 (Best/Last) | AIME2025 (Best/Last) | MATH500 (Best/Last) | Average (Best/Last) |
> |-----------------------|------------------------|-------------------------|-------------------------|------------------------|------------------------|
> | topP=0.95/topK=20/**temp=0.4** |55.9/53.4 |**24.6**/**21.9** |20.3/**19.5** |**80.9**/**80.3** |**45.4/43.8** |
> | topP=0.95/topK=20/**temp=0.6** |**56.9**/**56.9** |21.9/16.4 |20.3/14.1 |80.4/78.1 |44.9/41.4 |
> | topP=0.95/topK=20/**temp=0.8** |53.4/52.5 |22.3/7.8 |**21.1**/15.6 |80.6/73.5 |44.4/37.4 |
>
>
> **Conclusion**: Unlike top-P’s hard truncation, the temperature hyperparameter retains all possibilities but adjusts their relative probabilities. A low temperature (temp=0.4) sharpens the distribution, favoring high-confidence samples and reducing noisy trajectories, making it easier to converge to reliable solutions in high-difficulty tasks.
>
> | Hyper-parameters                | AMC2023 (Best/Last) | AIME2024 (Best/Last) | AIME2025 (Best/Last) | MATH500 (Best/Last) | Average (Best/Last) |
> |-----------------------|------------------------|-------------------------|-------------------------|------------------------|------------------------|
> | topP=0.95/**topK=10**/temp=0.6 |**58.1/58.1** |**22.3/18.4** |**21.1**/15.6 |**80.9**/78.9 |**45.6**/42.8 |
> | topP=0.95/**topK=20**/temp=0.6 |56.9/56.9 |21.9/16.4 |20.3/14.1 |80.4/78.1 |44.9/41.4 |
> | topP=0.95/**topK=50**/temp=0.6 |55.6/55.6 |20.3/**18.4** |18.4/**18.4** |80.6/**80.6** |43.7/**43.2** |
>
> **Conclusion**: GEPO is relatively insensitive to the top-k parameter. While top-k=10 achieves the highest best performance (45.6), top-k=50 yields the best final performance (43.2), suggesting that a larger candidate set enhances training stability without significantly compromising peak performance.
>
> Regarding the truncation strategy, we follow the standard practice in the literature, truncating sequences by maximum length when they exceed the context window and assume this is what the reviewer refers to. Since this is a common implementation detail shared by most prior works (e.g., GRPO, GSPO), we do not conduct additional ablations on alternative truncation schemes, as they are orthogonal to the core contributions of GEPO.

---

> ### Author Response · Authors · 2025-11-21
> **Response to Reviewer 8QcJ Part-6/7**
>
> ### **Sec 4 To weakness 4: Fairness to GSPO variants**
> **Your Original Comments:**
> >"**Fairness to GSPO variants.** Since GSPO is sequence-level IS with clipping, it would be good to include GSPO + stronger clipping / trust-region or defensive mixture baselines to test whether variance spikes can be similarly tamed by tuned GSPO, not just GEPO’s denominator. The paper itself sketches “defensive sampling” as future work—great idea, but makes me wonder how close-tuned GSPO would get."
>
> We sincerely thank the reviewer for this thoughtful and constructive comment. To ensure a fair comparison with GSPO, we emphasize that **all main experiments in the original submission used the same clipping hyperparameters**. Specifically, the standard PPO-style ratio clipping bounds of \([0.8, 1.2]\). Under this setting, GSPO already exhibited significant training instability (evidenced by large policy update variance and reward collapse), which led us to believe that *widening* the clipping range would likely exacerbate, rather than mitigate the issue. Hence our initial focus on GEPO’s structural advantage via its denominator based stabilization.
>
> To directly address the reviewer’s concern, we have now conducted an extensive set of follow-up experiments on **Qwen3-1.7B** (due to computational constraints) to explore whether carefully tuned GSPO variants can achieve comparable stability. Specifically, we evaluated:
>
>
> | **ID** | **Method** | **Key Idea** | **Performance vs. GSPO** | **Notes** |
> |--------|------------|--------------|--------------------------|-----------|
> | **1** | stronger clipping variants | Test tighter \([0.9, 1.15]\) ratio clipping |  \([0.9, 1.15]\) weakly increase the stability |  \([0.9, 1.15]\)  is better than \([0.8, 1.2]\) |
> | **2** | trust-region ($KL(p,q) < threshold$) + SFT on correct high-KL rollouts | Keep low-KL samples; apply SFT to *correct* high-KL samples only | **Best among GSPO variants**; reduces variance and improves final reward | Need tuning $threshold$ ; still underperforms GEPO |
> | **3** | Variant-2 + increase entropy on incorrect rollouts | Same as Variant-2, but increase entropy of *incorrect* high KL samples | No improvement over Variant-2; often worse | High KL incorrect samples appear too noisy to be useful |
> | **4** | Defensive sampling  | Use robust IS estimator:  $\frac{p}{\epsilon \cdot p.detach()+ (1-\epsilon) \cdot q}$ | Slight robustness gain at low $\epsilon$; unstable at higher delay | Matches “defensive sampling” idea in paper; limited efficacy alone |
>
>
> **Evaluation setting**: All variants tested on **Qwen3-1.7B** with identical delay simulation (same to our main experments).
>
> | Method                | AMC2023 (Best / Last) | AIME2024 (Best / Last) | AIME2025 (Best / Last) | MATH500 (Best / Last) | Average (Best / Last) |
> |-----------------------|------------------------|-------------------------|-------------------------|------------------------|------------------------|
> | GEPO| 53.8/**53.8**| **21.9/21.9**| 18.8/**18.8**| **79.6/79.6**| **43.5/43.5** |
> | GSPO-standard |**54.4**/23.8 | 17.6/1.6 | 17.6/2.7 | 78.2/55.6 | 42.0/20.9 |
> |GSPO-Variant1|53.8/40.6 |17.2/1.6 |19.5/3.1 |77.8/59.8 |42.1/26.3 |
> |GSPO-Variant2|53.8/50.9 |19.5/12.1 |**20.3**/14.5 |78.5/76.9 |43.0/38.6 |
> |GSPO-Variant3|43.8/43.8 |10.9/10.9 |13.7/13.7 |73.8/73.8 |35.5/35.5 |
> |GSPO-Variant4|54.3/42.3 |16.9/5.7 |15.3/4.4 |77.6/64.2 |41.0/28.9 |
>
>
> Our results show that **variant (2) achieves the best performance among all GSPO baselines**, significantly improving over vanilla GSPO, but still lags behind GEPO in both final reward and training stability under delay. Notably, variant (4), though intuitively appealing, fails to yield gains: high KL *incorrect* samples appear too noisy or misleading to provide useful signal, even with entropy regularization.
>
> These findings reinforce our claim that GEPO’s design stabilizes importance weights via the *group normalized denominator*, which offers a more principled and effective mechanism for variance control under asynchrony than post-hoc clipping or filtering strategies applied to standard IS (as in GSPO). We have added a brief summary of these results in the revised Appendix (Section A.4) for transparency.

---

> ### Author Response · Authors · 2025-11-21
> **Response to Reviewer 8QcJ Part-7/7**
>
> ### **Sec 4.2 Core Code for GSPO Variants 2–3**
>
> In this implementation, a mini-batch is the smallest unit.
>
> ```python
> # trust region based variants for GSPO
> ################################
> ##      Rollout Batch(mini)
> #         /       \
> #        /         \
> #    High-KL      Low-KL
> #     /   \       /   \
> #    /     \     /     \
> # right  wrong right  wrong
> #   |      |     |      |
> #  SFT  penalty   \    /
> #   or    or       \  /
> #discard discard    RL
> #################################
> ```
>
> ```python
> #### GSPO-Variant 2-4
> # Step-0 Compute KL-Divergence
> cppo_kl_local = loss_cppo_kl.item()
> # Step-1 judge high-KL or low-KL
> if cppo_kl_local > self.script_args.sync_beta and advantages.max() > 0:
>     if self.loss_type in "gspo-var2":
>         SFT_mask = torch.tensor(advantages > 0).clone().detach().to(torch.int)
>         # Step2 SFT on correct and high-KL samples
>         SFT_loss = -(learner_seq_lopp * SFT_mask).sum()/SFT_mask.sum().clamp(min=1.0)
>     # Step3 incrase entropy on incorrect and high-KL rollouts
>     if self.loss_type == "gspo-var3":
>         # SFT on correct and high-KL rollouts
>         SFT_mask = torch.tensor(advantages > 0).clone().detach().to(torch.int)
>         SFT_loss = -(learner_seq_lopp * SFT_mask).sum()/SFT_mask.sum().clamp(min=1.0)
>         # Increase Entropy on wrong and high-KL rollouts
>         PEN_mask = torch.tensor(advantages <= 0).clone().detach().to(torch.int)
>         seq_entropy_clip = seq_entropy.clamp(max=self.script_args.entropy_bar)
>         PEN_loss = 1 - (seq_entropy_clip * PEN_mask).sum()/PEN_mask.sum().clamp(min=1.0)
> else:
>     # Compute GSPO loss
>     ...
>
> ```
>
> ### **Sec 4.3 Core Code for GSPO Variants 4**
>
> Defensive sampling is very easy to implement using the formula $\frac{p}{\epsilon \cdot p.detach() + (1 - \epsilon) \cdot q}$.
>
> ```python
> # introduce constan $\epsilon$ for defensive sampling
> coef_1 = learner_seq_p / ( self.defen_epsilon * learner_seq_p.detach() + (1-self.defen_epsilon) * sampler_seq_p )
> ```
>
> - In our result, defensive sampling can improve the stability of GSPO, but fixed $\epsilon$ (such as 0.005) hurts the best performance.
>
> **The complete code for the GSPO variant can be found in** `GEPO_Code_Openreview/src/hetero_rl/learner_script_RFTv2.py`.

---

> > ### Author Response · Authors · 2025-11-25
> > **We Are Eagerly Looking Forward to Your Reply**
> >
> > Dear Reviewer 8QcJ,
> >
> > Thank you again for your thoughtful and constructive feedback, which has greatly helped us strengthen our work. We sincerely hope that our detailed responses—including new bias quantification, theoretical bounds, expanded ablation studies, and fair comparisons with enhanced GSPO variants—have addressed your concerns regarding GEPO’s bias–variance trade-off, external validity, hyperparameter sensitivity, and baseline fairness.
> >
> > We would be truly grateful if you could let us know:
> > - Have we adequately resolved the issues you raised?
> > - Are there any remaining gaps or unclear points in our rebuttal?
> > - Is there anything further we could clarify, analyze, or even experimentally verify to better support our claims?
> >
> > The rebuttal period is still open, and we remain fully committed to addressing any additional questions or suggestions you may have. We warmly welcome further dialogue and are happy to run additional experiments or provide more details if needed. Thank you once more for your time, rigor, and generosity in engaging with our work.

---

> > > ### Author Response · Authors · 2025-11-28
> > > **Time is passing, and we look forward to further discussion with you.**
> > >
> > > Dear Reviewer 8QcJ,
> > >
> > > Thank you very much for your thoughtful review and constructive feedback on our submission. We sincerely appreciate the time and effort you have dedicated to evaluating our work.
> > >
> > > **We understand that the ICLR review system no longer allows changes to scores at this stage. Nevertheless, we would be truly grateful if you could still participate in the open discussion phase.** Your insights remain extremely valuable to us, and we are eager to address any remaining concerns or questions you may have.
> > >
> > > Regardless of scoring constraints, we are fully committed to responding thoroughly to your comments, clarifying any ambiguities, and, where appropriate, providing additional analysis or experiments to strengthen the paper.
> > >
> > > Thank you again for your engagement and guidance. We look forward to continuing the discussion with you.
> > >
> > > Best regards,
> > >
> > > The Authors

---

> ### Author Response · Authors · 2025-12-02
> **An interesting finding highly aligned with the yesterday released DeepSeek V3.2 .**
>
> We are surprised to find that the GSPO-Variant2 is highly consistent with the idea behind DeepSeek V3.2[1], which was released on December 1st. Recent research on DeepSeek V3.2 involves **discarding negative samples with high KL divergence**. **GSPO-Variant2 (Best among GSPO variants) follows a similar philosophy—both approaches aim to discard ineffective negative samples.** In GSPO-Variant2,  we use supervised fine-tuning (SFT) on "high KL & positive" samples and discarding "high KL & negative" samples.  We then found that this approach performs best among all GSPO variants. **DeepSeek V3.2 adopting a similar approach further demonstrates the validity of our research on GSPO variants.**
>
> ### **References**
>
> [1] DeepSeek-V3.2 https://api-docs.deepseek.com/news/news251201

---

### Author Response · Authors · 2025-11-21
**Global Response**

We sincerely thank all reviewers for their thoughtful feedback and valuable suggestions.

Reviewers highlighted GEPO's strengths as: "**Clear systems problem + algorithmic handle**" (8QcJ), having a "**conceptually elegant and practically powerful**" approach to variance reduction (bBKw), and offering clear "**originality**," strong "**theory**," and convincing "**empirics**" (5yWm).

For the weakness, we have incorporated the following key improvements:

### **Update Manuscript**
- (1) New Appendix Section **J.1**: Add comprehensive bias analysis to quantify the bias $|\mathbb{E}_p [A] - \mathbb{E}_q [\frac{p}{\mathbb{E}_q q} A]|$ of GEPO's estimator (Response to Weakness-1 of Reviewer 8QcJ)
- (2) New Appendix Section **J.2**: Extend variance analysis from $Var(\frac{p}{\mathbb{E}_qq})$ to the more general $Var(\frac{p}{\mathbb{E}_qq} A)$ form for rigorous stability guarantees (Resonse to Weakness-4 of Reviewer aJa6)
- (3) New Appendix Section **K**: Add comparisons with asynchronous RL baselines (**Truncated IS** technique of **IMPALA/V-trace** [Espeholt et al., 2018], **CISPO** [Chen et al., 2025] and **TOPR** [Roux et al., 2025])  to better contextualize GEPO's advantages
- (4) New Appendix Section **L**: Conduct thorough hyperparameter studies examining group size, delay distributions, KL-regularization, and top-P/K/temperature sampling strategies
- (5) Update **Section 5 Related Work**:  Introduce asynchronous RL from both the framework and algorithmic perspectives.
###  **Update Code** (add new policy optimization algotrithms)
- (1) Truncated IS (1802.01561) as asynchronous baseline,
- (2) CISPO (2506.13585) as asynchronous baseline,
- (3) TOPR (2503.14286) as asynchronous baseline,
- (4) GSPO with trust region (by Reviewer 8QcJ) **where the GSPO-Variant2 is highly aligned with the Dec.1st released DeepSeek V3.2**,
- (5) GSPO with defensive sampling (by Reviewer 8QcJ),
- (6) GMPO (2507.20673), concurrent work,
- (7) VL Norm (2509.07558), concurrent work.

Detailed point-by-point responses follow in our individual reviewer replies.

---

### Meta-Review · Area_Chair_7Wcs · 2025-12-21

**Summary:**

This paper introduces GEPO, a method to stabilize decentralized RL training by reducing the variance of importance weights through group expectation weighting. The reviews are predominantly positive, leading to an acceptance decision. Reviewers consistently praise the work for its clear and impactful problem formulation, directly addressing the critical issue of policy staleness and variance explosion in heterogeneous systems. The proposed solution is seen as conceptually elegant and theoretically grounded, with compelling empirical results demonstrating superior stability and lower performance degradation compared to strong baselines like GRPO and GSPO.

While accepted, the paper has noted limitations that should be addressed in a final version. The primary concerns are the narrow experimental scope (limited to math reasoning tasks) and the need for a more thorough bias-variance trade-off analysis. Additional requested revisions include: expanding comparisons to recent asynchronous RL systems, providing sensitivity analyses for hyperparameters like group size, and clarifying the theoretical consistency when using non-normalized sampling distributions. However, the committee agrees that the paper's core novelty, strong theoretical insight, and practical solution to a well-motivated problem constitute a sufficient contribution for acceptance, with the expectation that the authors will incorporate the reviewers' constructive feedback.

**Reviewer Concerns:**

I think most of the reviewers' concerns have been addressed. Though there are still some unresolved comments, this won't impact the acceptance decision.

**Reviewer Scores:**

I think most of reviewers will raise the scores or maintain the positive scores.

---

### Decision · Program_Chairs · 2026-01-26

Accept (Poster)